# A Local Data Assimilation Method (Local DA v1.0) and its Application in a Simulated Typhoon Case

Shizhang Wang [1,2], Xiaoshi Qiao [1,2]

[1] Nanjing Joint Institute for Atmospheric Sciences, Nanjing, 210000, China

[2] Key Laboratory of Transportation Meteorology of China Meteorological Administration, Nanjing, 210000, China

*Correspondence to*: Xiaoshi Qiao (497390719@qq.com)

**Abstract.** Integrating the hybrid and multiscale analyses and the parallel computation is necessary for current data assimilation schemes. A local data assimilation method, Local DA, is designed to fulfill these needs. This algorithm follows the grid-independent framework of the local ensemble transform Kalman filter (LETKF) and is more flexible in hybrid analysis than the LETKF. Local DA employs an explicitly computed background error correlation matrix of model variables mapped to observed grid points/columns. This matrix allows Local DA to calculate static covariance with a preset correlation function. It also allows using the conjugate gradient (CG) method to solve the cost function and allows performing localization in model space, observation space, or both spaces (double-space localization). The Local DA performance is evaluated with a simulated multiscale observation network that includes sounding, wind profiler, precipitable water vapor, and radar observations. In the presence of a small-size time-lagged ensemble, Local DA can produce a small analysis error by combining multiscale hybrid covariance and double-space localization. The multiscale covariance is computed using error samples decomposed into several scales and independently assigning the localization radius for each scale. Multiscale covariance is conducive to error reduction, especially at a small scale. The results further indicate that applying the CG method for each local analysis does not result in a discontinuity issue. The wall clock time of Local DA implemented in parallel is halved as the number of cores doubles, indicating a reasonable parallel computational efficiency of Local DA.

## 1 Introduction

Data assimilation (DA), which estimates the atmospheric state by ingesting information from model predictions, observations, and background error covariances, is crucial for the success of numerical weather prediction (Bonavita et al., 2017). Therefore, many previous studies on DA have focused primarily on how to utilize observations and how to estimate background error covariances (e.g., Huang et al., 2021; Wang et al., 2021; Lei et al., 2021; Zhang et al., 2009; Brousseau et al., 2011, 2012; Wang et al., 2013a; Wang et al., 2012; Kalnay and Yang, 2008; Buehner and Shlyaeva, 2015). At present, there are two prevailing research orientations of DA: hybrid analysis, which concerns the background error covariance, and multiscale analysis, which often addresses the difference in observation scales.

The hybrid analysis aims to utilize both the ensemble and static covariances to leverage the advantages of flow-dependent error information and prevents the analysis from degrading due to a limited ensemble size (Wang et al., 2009; Etherton and Bishop, 2004). A widely used hybrid approach is to add an ensemble-associated control variable to a variational DA framework (Lorenc, 2003; Wang et al., 2008). An alternative combines the ensemble and static covariances (Hamill and Snyder, 2000). These two approaches are equivalent (Wang et al., 2007). Another hybrid method averages the analyses
yielded by the ensemble Kalman filter (EnKF) and the variational method (Bonavita et al., 2017; Penny, 2014). Recently, a hybrid scheme based on the EnKF framework was developed (Lei et al., 2021) that uses a large ensemble size (=800) to simulate the static error covariance. Nevertheless, given the variety of hybrid approaches available, how to conduct hybrid DA is still a matter of debate. In this study, a hybrid scheme is implemented following Hamill and Snyder (2000), although the proposed scheme differs regarding the details.

Multiscale DA is designed to utilize observations at different scales and performs multiscale localization in either the model or observation space. Localization is inevitable due to sampling errors (e.g., distant spurious correlations) in ensemble-based DA, including in hybrid DA (e.g., Huang et al., 2021; Wang et al., 2021). Varying the localization radius for observations according to the observation scale or density is a straightforward method; examples include the assimilation of synoptic-scale observations with a large localization radius and then performing radar DA with a small radius of influence (e.g.,
Zhang et al., 2009; Johnson et al., 2015). An alternative is to perform multiscale localization in model space, requiring the scale decomposition of the ensemble members (Buehner and Shlyaeva, 2015). The model space localization allows ingesting all observations on different scales simultaneously. Recent studies have shown that multiscale DA outperforms DA with fixed localization (Caron and Buehner, 2018; Caron et al., 2019; Huang et al., 2021).

In addition to the analysis quality of DA, computational efficiency should also be considered (Bonavita et al., 2017). A
highly parallelized DA scheme is preferable due to the continuously increasing model resolution and the number of available observations. One DA scheme that can be highly parallelized is the local ensemble transform Kalman filter (LETKF, Hunt et al., 2007), whose analysis is grid-independent.

In brief, both hybrid DA and multiscale DA are necessary, and the parallel computation efficiency of the LETKF is attractive. Thus, it is desirable to utilize all their advantages. A straightforward idea for achieving hybrid DA with the
LETKF is to use a large-size static ensemble, similar to the EnKF-based hybrid scheme proposed by Lei et al. (2021). The large ensemble (>=800) is not always available in practice because of the limited computational and storage resources. However, it is inevitable to use such an ensemble to realize the hybrid analysis in the original LETKF framework because the LETKF works in the ensemble space. In this situation, it is desirable to design a flexible DA scheme that follows the grid-independent analysis of the LETKF and can perform both hybrid and multiscale analysis with or without static
ensemble members, similar to other variational-based hybrid schemes. The scheme is named Local DA hereafter.

Compared with the LETKF, Local DA computes the linear combination of columns of a local background error correlation matrix rather than the combination of ensemble members. The local background error correlation matrix is in model space, but the model variables are interpolated to observed grid points/columns. In other words, Local DA works on unstructured

grids. This framework is friendly for assimilating integrated observations, such as precipitation water vapor (PWV), because vertical localization can be performed in model grid space. Explicitly computing the error correlation matrix requires much more memory than the LETKF but allows Local DA to calculate the static background error correlation with a preset correlation function, such as the distant correlation function. Moreover, the computational cost of the matrix is acceptable if observations are appropriately thinned.

Since the error correlation matrix is explicitly constructed, it is straightforward to realize the hybrid DA according to the idea of Hamill and Snyder (2000). This approach is often utilized with a simple model (Kleist and Ide, 2015; Penny, 2014; Etherton and Bishop, 2004; Lei et al., 2021) because it explicitly computes and directly combines the background error covariance matrices. In this study, we attempt to evaluate the hybrid idea of Hamill and Snyder (2000) in a realistic complicated scenario.

Another feature of Local DA is the ability to perform multiscale analysis in model space, observation space, or both spaces (double-space localization). In the model space, Local DA adopts a scale-aware localization approach for multiscale analysis that applies a bandpass filter to decompose samples and individually performs localization at each scale; no cross-scale covariance is considered in current Local DA. A similar idea (i.e., lacking cross-scale covariance) is the scale-dependent localization technique proposed by Buehner (2012). Although cross-scale covariance is likely to improve multiscale analysis, the relative performance depends on ensemble size (Caron et al., 2019).

Local DA can perform observation-space localization similar to LETKF, which magnifies the observation error as the distance between the observation and model variables increases. For the multiscale analysis in the observation space, the localization radius increases as the scale of observation increases. Compared with radar data, the scale of sounding data is larger so that a larger radius is assigned.

Because model space localization and observation space localization are conducted for covariances in different spaces, it is possible to perform both localizations synchronously. Although double-space localization may result in a double penalty, it would be interesting to note the localization performance. Note that the LETKF of Wang et al. (2021) can also realize double-space localization, but this application has not yet been investigated.

As the first paper to report on Local DA, this study focuses on the following main issues: i) how to locally conduct the hybrid and multiscale analysis, ii) the spatial continuity of local analysis, iii) the impact of the hybrid covariance, and multiscale localization on Local DA, and iv) the performance of Local DA on cycling DA. Since Local DA is designed to be a more flexible hybrid scheme than LETKF, we do not expect Local DA to outperform LETKF in all scenarios. The comparison of both methods only focuses on i) if they yield similar results in the case of using observation space localization and ensemble covariance only and ii) if Local DA with hybrid covariance outperforms the LETKF with a poor ensemble.

 Observing system simulation experiments (OSSEs) are adopted to avoid issues associated with the quality control of observations. The simulated multiscale observing system consists of sounding, wind profiler, PWV, and radar (radial velocity and reflectivity) observations; the scales of these observations vary from the synoptic scale to the convective scale. A simulated typhoon case is selected for the evaluation.

The remainder of this paper is organized as follows. In Sect. 2, Local DA and its associated multiscale localization technique are described, including the formula, workflow, and other details. Sect. 3 describes the numerical experiments, and Sect. 4

discusses the results. A summary and conclusions are given in Sect. 5.

## 2 Method

### 2.1 The Local DA scheme

As mentioned above, Local DA performs analysis in model space, but it needs to map model variables onto observed grid points/columns before the analysis. All DA methods conduct the mapping, but Local DA updates the mapped model

variables. Both the background model state ($\mathbf{x}^f$) and the ensemble perturbations ($\mathbf{X}$) are mapped according to $\mathbf{H}_i$, the vector of interpolation operators. The mapped model state and perturbations are denoted by $\mathbf{x}_o^f$ and $\mathbf{X}_o$, respectively, where the subscript "o" represents the observed grid points/columns. Note that Local DA only stores $\mathbf{x}_o^f$ and $\mathbf{X}_o$ for a local analysis rather than the whole forecast domain. An example of the spatial distribution of variables involved in Local DA is shown in Figure 1.

The cost function of Local DA is written as

$$J = \frac{1}{2}\mathbf{v}_o^T \mathbf{v}_o + \frac{1}{2}(\mathbf{H}_o \hat{\mathbf{X}}_o \mathbf{v}_o - \mathbf{d})^T \mathbf{R}^{-1}(\mathbf{H}_o \hat{\mathbf{X}}_o \mathbf{v}_o - \mathbf{d}) , \qquad (1)$$

where $\mathbf{v}_o$ is the control variable (or a combination of error samples), the observation error covariance is denoted by $\mathbf{R}$, which is a diagonal matrix in this study, $\mathbf{H}_o$ is the linear operator of h that converts the model variables into observation variables, and $\mathbf{d}$ is the observation innovation vector. $\hat{\mathbf{X}}_o$ ($=\alpha \mathbf{S}_o \mathbf{C}_{oo}$) represents a constructed error-sample matrix, where $\mathbf{C}_{oo}$ is the

local background error correlation matrix, $\mathbf{S}_o$ stores the standard deviations (STDs) of the model variables, and $\alpha$ is a parameter that adjusts the trace of $\mathbf{C}_{oo}$. The dimensions of vectors and matrices in equation (1) depend on the number of observations involved in a local analysis and the complexity of observation operators. We will give the dimensions and computations of the above variables in the following subsections.

Once $\mathbf{v}_o$ is obtained, the model state increment $\mathbf{x}^i$ on the model grids can be computed in terms of

$\mathbf{x}^i = \hat{\mathbf{X}}_{mo}\mathbf{v}_o ,$ \qquad\qquad\qquad (2)

where $\hat{\mathbf{X}}_{mo} = \alpha \mathbf{S}_m \mathbf{C}_{mo}$, $\mathbf{S}_m$ contains the STDs of the model variables on the model grids, and $\mathbf{C}_{mo}$ is a correlation matrix that contains the correlation coefficients between $\mathbf{X}_o$ and $\mathbf{X}$. Details regarding $\mathbf{C}_{oo}$ and $\mathbf{C}_{mo}$ will be given later. The analyzed model state $\mathbf{x}^a$ is computed in accordance with $\mathbf{x}^a = \mathbf{x}^f + \mathbf{x}^i$.

To update ensemble perturbations, the current version of Local DA adopts the stochastic method (Houtekamer and Mitchell,

1998) that treats observations as random variables. This method adds random perturbations with zero mean to $\mathbf{d}$ in Eq. (1). For an $M$-member ensemble, equations (1) and (2) are conducted $M$ times to update members with perturbed observations,

similar to the procedure of Li et al. (2012). These analyses share the same background error covariance but use different observations. The stochastic approach was reported to be less accurate than the deterministic approach (e.g., Whitaker and Hamill, 2002) because it introduces additional sampling error. At this stage, Local DA mainly concerns the deterministic analysis; further improvement of the analysis ensemble is left in future work.

Compared with the LETKF or the En4DVar of Liu et al. (2008), Local DA seeks the combination ($\mathbf{v}_0$) in model space, or more specifically, the combination of the columns of a local background error correlation matrix of model variables, rather than the combination in ensemble space. Thus how to construct $\mathbf{C}_{oo}$ and $\mathbf{C}_{mo}$ is key for Local DA. Explicitly computing $\mathbf{C}_{oo}$ raises the question of how to solve the cost function of Local DA in the case of large-size $\mathbf{C}_{oo}$. In addition, how to deal with nonlinear observation operators should be determined. The subsequent subsections present the answers to these questions.

### 2.1.1 The local background error correlation matrix

In Local DA, the actual correlation matrix $\tilde{\mathbf{C}}$ is the square of $\mathbf{C}_{oo}$ multiplied by a rescaling parameter $\alpha^2$:

$$\tilde{\mathbf{C}} = \alpha^2 \mathbf{C}_{oo} \mathbf{C}_{oo}^{\mathrm{T}}. \tag{3}$$

By using the rescaling parameter, the trace of $\tilde{\mathbf{C}}$ is equivalent to that of $\mathbf{C}_{oo}$. $\alpha$ is computed according to

$$\alpha = \sqrt{tr(\mathbf{C}_{oo}) \Big/ tr(\mathbf{C}_{oo} \mathbf{C}_{oo}^{\mathrm{T}})}, \tag{4}$$

where $tr(\ )$ denotes the calculation of the trace of a matrix. Notably, $tr(\mathbf{C}_{oo} \mathbf{C}_{oo}^{\mathrm{T}})$ is equal to the sum of squares of all elements in $\mathbf{C}_{oo}$. There is no need to compute $\mathbf{C}_{oo} \mathbf{C}_{oo}^{\mathrm{T}}$. $\tilde{\mathbf{C}}$ and $\mathbf{C}_{oo}$ are identical in terms of eigenvectors and the trace of the matrix (total variance). The eigenvalues of $\tilde{\mathbf{C}}$ are the squares of the corresponding eigenvalues of $\mathbf{C}_{oo}$ multiplied by $\alpha^2$. Therefore, $\tilde{\mathbf{C}}$ is an approximation of $\mathbf{C}_{oo}$. Storto and Andriopoulos (2021) proposed a hybrid DA scheme that also used the rescaling parameter to tune the trace of a matrix (see their Eq. (15)), but they constructed the background error covariance in a way differing from ours.

$\mathbf{C}_{oo}$ is a $K \times K$ matrix, where $K$ is the number of model variables associated with the observations to be assimilated. $K$ is computed according to

$$K = \sum_{i=1}^{N_t} [N_o(i) N_{op}(i)], \tag{5}$$

where $N_t$ is the number of observation types, such as the zonal wind from soundings and the radial velocity from radars, $N_o(i)$ is the number of observations of the $i$th type, and $N_{op}(i)$ is the number of model variables used by the observation operator of the $i$th type. For instance, if radar reflectivity is the only available observation type and there are 100 observations, $K$ is equal to 300 ($100 \times 3$) in the case of using the observation operator of Gao and Stensrud (2012) because the operator requires three

hydrometeors ($q_r$, $q_s$, and $q_g$). Now we are going to give an example of $\mathbf{C}_{oo}$. Assuming there are three available observations

(two zonal wind observations and a surface pressure observation), the background error correlation matrix is

$$
\mathbf{C}_{oo} = \begin{pmatrix} c_{u1u1} & c_{u1u2} & c_{u1ps1} \\ \\ c_{u2u1} & c_{u2u2} & c_{u2ps1} \\ \\ \\ c_{ps1u1} & c_{ps1u2} & c_{ps1ps1} \end{pmatrix},
\tag{6}
$$

where $c$ is the correlation coefficient in the space of $\mathbf{X}_o$ and the subscripts "$u1$", "$u2$", and "$ps1$" represent the two zonal

wind observations and a surface pressure observation, respectively. Correspondingly, the STD matrix $\mathbf{S}_o$ can be written as

$$
\mathbf{S}_o = \begin{pmatrix} s_{u1} & & \\ \\ & s_{u2} & \\ \\ \\ & & s_{ps1} \end{pmatrix},
\tag{7}
$$

where $s$ denotes the STDs of the model variables projected onto the observed grids. $\mathbf{S}_o$ is a $K \times K$ matrix, but a $K \times 1$ array is

sufficient to store $\mathbf{S}_o$. After $\mathbf{C}_{oo}$ and $\mathbf{S}_o$ are formed, $\mathbf{v}_o$ can be solved. In this example, $\mathbf{v}_o$ is in the following form:

$$
\mathbf{v}_o = \begin{pmatrix} v_{u1} & v_{u2} & v_{ps1} \end{pmatrix}^{\mathrm{T}},
\tag{8}
$$

where subscripts "$u1$", "$u2$", and "$ps1$" have the same meaning in Eqs. (6) and (7).

To obtain the model state increment $\mathbf{x}^i$, it is necessary to form $\mathbf{C}_{mo}$ and the corresponding $\mathbf{S}_m$. If the model variables to be

updated are the zonal wind ($v1$), potential temperature ($\theta1$), and water vapor mixing ratio ($q1$), $\mathbf{C}_{mo}$ is written as

$$
\mathbf{C}_{mo} = \begin{pmatrix} c_{\theta1u1} & c_{\theta1u2} & c_{\theta1ps1} \\ \\ c_{q1u1} & c_{q1u2} & c_{q1ps1} \\ \\ \\ c_{v1u1} & c_{v1u2} & c_{v1ps1} \end{pmatrix},
\tag{9}
$$

where subscripts "$u1$", "$u2$", and "$ps1$" are the same as those in Eqs. (6) and (7), while subscripts "$v1$", "$\theta1$", and "$q1$"

denote the model variables to be updated. $\mathbf{C}_{mo}$ comprises the error correlation coefficients between $\mathbf{X}$ and $\mathbf{X}_o$. The size of

$\mathbf{C}_{mo}$ is $N_m K$ which depends on the number ($N_m$) of model variables to be updated. However, there is no need to store full $\mathbf{C}_{mo}$

in practice because one row of $\mathbf{C}_{mo}$ is needed to update the corresponding model variable. $\mathbf{S}_m$ is the STD matrix of model

variables, containing $s_{v1}$, $s_{\theta1}$, and $s_{q1}$ in this example. For convenience, a summary of the dimensions of variables involved in Local DA is listed in Table 1.

Note that the variational DA methods and Local DA differ in the control variable transform viewpoint. The former uses the square root of the background error covariance matrix, while Local DA employs the error correlation matrix. It is based on the consideration of computational cost because it is expensive to obtain the square root of $\mathbf{C}_{oo}$ if the size of $\mathbf{C}_{oo}$ is large. Moreover, modeling the square root of the background error covariance matrix, as many variational DA methods do, is also difficult for Local DA because the irregular distribution of observations makes it infeasible to utilize a recursive filter.

Additionally, note that the size of $\mathbf{C}_{oo}$ grows rapidly as $K$ increases. However, the memory requirement is affordable since $\mathbf{C}_{oo}$ is only used for local analysis. For high-resolution observations, thinning can help reduce the cost, which is also necessary to ensure that the observation errors are uncorrelated (e.g., Hoeflinger et al., 2001). We use the same model variables for the data observing the same grid point/column. For example, the same hydrometeor variables ($q_r$, $q_s$, and $q_g$) are used to compute the radar reflectivity and differential reflectivity at the same observed grid point. In this situation, the size of $\mathbf{C}_{oo}$ does not increase with the observations. This strategy is also valid for passive microwave observations at different frequencies obtained by a satellite because they observe the same column of the atmosphere. Therefore, the size of $\mathbf{C}_{oo}$ is controllable. We use a simple thinning approach to control the matrix size in this study, as described in the Appendix.

### 2.1.2 The solution of Local DA

There are two methods to solve the gradient of Eq. (1): i) matrix decomposition and ii) an iterative algorithm. The first approach is straightforward but is time-consuming and sometimes infeasible if the $\mathbf{C}_{oo}$ size is large. Therefore, Local DA adopts an iterative algorithm, namely, the conjugate gradient (CG) method (Shewchuk, 1994). Theoretically, the CG method requires the background error covariance matrix to be positive definite. However, with the control variable transform, a positive semidefinite covariance matrix is sufficient to obtain the best linear unbiased estimate (Ménétrier and Auligné, 2015). A strictly diagonally dominant matrix with nonnegative diagonal elements is positive semidefinite.

Although a positive semidefinite covariance matrix is sufficient, using a higher-rank background error covariance matrix helps obtain a lower analysis error (Huang et al., 2019). Compared with the rank of $\mathbf{X}$, which is not higher than the ensemble size, that of $\hat{\mathbf{X}}_o$ is much higher after $\mathbf{C}_{oo}$ is localized. Our early test (not shown) indicates that $\hat{\mathbf{X}}_o$ is a full rank matrix in most cases. For rank-deficient cases, the rank of $\hat{\mathbf{X}}_o$ is often greater than 97% of the full rank value. The details of this localization will be given later.

Note that Local DA performs the CG step locally, unlike other variational-based DA methods that apply the CG method globally. Therefore, it is necessary to investigate whether the local application of the CG method causes a nonnegligible spatial discontinuity, which will be discussed in Sect. 4. For computational efficiency, the maximum number of iterations is 100. If the error tolerance $\varepsilon^2$ defined in (Shewchuk, 1994) cannot reach $1 \times 10^{-6}$ by the $100^{th}$ step, the CG method is stopped.

### 2.1.3 The observation operator

The EnKF algorithm often approximates the linear projection, $\mathbf{H}$ in Eq. (1), according to the departure of the observation priors from their ensemble mean. It is straightforward for Local DA to use the ensemble approximation approach. However, for nonlinear observation operators, there is an alternative, namely, the observation prior calculated by using the ensemble mean of the model variables. Tang et al. (2014) demonstrated that this alternative could lead to better results. Furthermore, Yang et al. (2015) examined the application of this alternative in radar DA and showed that the alternative approach produced lower analysis errors for the model variables associated with radial velocity (three wind components) and reflectivity (mixing ratios of rain, snow, and graupel). Given that remote sensing observations such as those obtained by radars and satellites are important parts of a multiscale observation network, Local DA adopts the alternative approach proposed by Tang et al. (2014).

Local DA approximates the linear projection $\tilde{\mathbf{Y}} = \mathbf{H}_\text{o}\hat{\mathbf{X}}_\text{o}$ according to

$$\tilde{\mathbf{Y}} \approx h(\mathbf{x}^\text{f} + \hat{\mathbf{X}}_\text{o}) - h(\mathbf{x}^\text{f} + \overline{\hat{\mathbf{X}}_\text{o}}) , \tag{10}$$

where $h$ is the nonlinear observation operator, $\mathbf{x}^\text{f}$ is the background model state vector, and $\overline{\hat{\mathbf{X}}_\text{o}}$ is the mean of $\hat{\mathbf{X}}_\text{o}$. Note that Eq. (10) is written for a deterministic forecast in this study. Compared with the results using the ensemble mean of observation priors, Eq. (10) reduces the analysis error of reflectivity by approximately 2 dBZ in our early test (not shown). This result is consistent with that of Yang et al. (2015).

### 2.2 Multiscale localization

To realize multiscale localization in model space, Local DA first performs scale decomposition with a bandpass filter. The decomposed perturbation, $\mathbf{X}'_\text{b}$, is

$$\mathbf{X}'_\text{b} = \left( \mathbf{X}^1_\text{b}, \mathbf{X}^2_\text{b}, \cdots, \mathbf{X}^l_\text{b}, \cdots, \mathbf{X}^{N_\text{b}}_\text{b} \right), \tag{11}$$

where the superscript "$l$" represents the $l$th scale and $N_\text{b}$ is the number of scales. After decomposition, the number of samples becomes $N_\text{b}$ times as large as the original ensemble size. As a localization approach lacking cross-scale covariance (no $\mathbf{X}^i_\text{b}\mathbf{X}^{j\text{T}}_\text{b}, i \neq j$ term in $\mathbf{X}'_\text{b}\mathbf{X}'^{\text{T}}_\text{b}$), Local DA computes the STD of the perturbation, $s$, according to

$$s(i) = \sqrt{\sum_{l=1}^{N_\text{b}} \frac{1}{N} \sum_{m=1}^{N} \left[ \mathbf{X}^l_\text{b}(i,m) \right]^2} , \tag{12}$$

where $i$ and $m$ denote the $i$th model variable and the $m$th sample, respectively, and $N$ is the sample size. Compared with the raw STD, $\sqrt{\dfrac{1}{N} \sum_{m=1}^{N} \left[ \sum_{l=1}^{N_\text{b}} \mathbf{X}^l_\text{b}(i,m) \right]^2}$, the cross influence among different scales of $\mathbf{X}'_\text{b}$ is ignored in Eq. (12). Nevertheless,

we acknowledge the importance of the cross influence of these perturbations and plan to investigate this issue with regard to Local DA in our future work.

230 The multiscale correlation coefficient $c(i,j)$ is calculated according to

$$c(i, j) = \sum_{l=1}^{N_b} \frac{\text{cov}\left[\mathbf{X}_b^l(i), \mathbf{X}_b^l(j)\right]}{s(i)s(j)}, \tag{13}$$

where $i$ and $j$ denote the $i$th and $j$th variables, respectively. For the case of $i=j$, Eq. (13) ensures $c(i,j)=1.0$.

We perform localization for each scale independently to construct the multiscale correlation matrix. In principle, our multiscale localization method trusts the correlation coefficient of each scale when the distance between two variables is 235 smaller than the lower bound of the scale. For instance, for the scale of 50 km – 100 km, Local DA starts the localization when the distance $d$ is greater than 50 km. The decorrelation coefficient $r(l,i,j)$ for the $l$th scale and $c(i,j)$ is calculated according to

$$\begin{cases} r(l,i,j) = 1.0, \ d<=d_{min}(l) \\ \\ r(l,i,j) = e^{-8\left[\frac{d-d_{min}(l)}{d_r(l)}\right]^2}, \ d>d_{min}(l) \ , \\ \\ r(l,i,j) = 0.0, \ d>d_{max}(l) \end{cases} \tag{14}$$

where $d_{min}(l)$ and $d_{max}(l)$ are the lower and upper bounds of the $l$th scale, respectively, and $d_r(l)$ is the localization radius for 240 the $l$th scale. Note that how to optimally localize the background error covariance is still an open question; rather, Eq. (14) is simply a preliminary implementation of multiscale localization for Local DA.

Substituting equations (13) and (14) into equation (6), an example of $\mathbf{C}_{oo}$ in equation (6) is written as:

$$\mathbf{C}_{oo} = \begin{pmatrix} \sum_{l=1}^{N_b} r(l,u1,u1)\dfrac{\text{cov}\left[\mathbf{X}_b^l(u1), \mathbf{X}_b^l(u1)\right]}{s(u1)s(u1)} & \cdots & \cdots \\ \sum_{l=1}^{N_b} r(l,u2,u1)\dfrac{\text{cov}\left[\mathbf{X}_b^l(u2), \mathbf{X}_b^l(u1)\right]}{s(u2)s(u1)} & \cdots & \cdots \\ \sum_{l=1}^{N_b} r(l,ps1,u1)\dfrac{\text{cov}\left[\mathbf{X}_b^l(ps1), \mathbf{X}_b^l(u1)\right]}{s(ps1)s(u1)} & \cdots & \cdots \end{pmatrix}, \tag{15}$$

where $i$ and $j$ in equation (13) are replaced by subscripts in equation (6). For brevity, only the first column of $\mathbf{C}_{oo}$ is listed. 245 Obviously, applying multiscale localization does not change the size of $\mathbf{C}_{oo}$. Correspondingly, an example of $\mathbf{C}_{mo}$ in equation (8) can be written as:

$$\mathbf{C}_{\mathrm{mo}} = \begin{pmatrix} \sum_{l=1}^{N_b} r(l, v1, u1) \dfrac{\mathrm{cov}\left[ \mathbf{X}_b^l(v1), \mathbf{X}_b^l(u1) \right]}{s(v1)s(u1)} & \dots & \dots \\[2ex] \sum_{l=1}^{N_b} r(l, \theta1, u1) \dfrac{\mathrm{cov}\left[ \mathbf{X}_b^l(\theta1), \mathbf{X}_b^l(u1) \right]}{s(\theta1)s(u1)} & \dots & \dots \\[2ex] \sum_{l=1}^{N_b} r(l, q1, u2) \dfrac{\mathrm{cov}\left[ \mathbf{X}_b^l(q1), \mathbf{X}_b^l(u1) \right]}{s(q1)s(u1)} & \dots & \dots \end{pmatrix}. \tag{16}$$

Because the multiscale localization does not change the sizes of $\mathbf{C}_{\mathrm{oo}}$ and $\mathbf{C}_{\mathrm{mo}}$, there is no modification for $\mathbf{v}_{\mathrm{o}}$, $\mathbf{x}^{\mathrm{i}}$, $\mathbf{x}^{\mathrm{f}}$, and $\mathbf{x}^{\mathrm{a}}$. The only modification to realize multiscale localization in model space is to store the error sample of each scale and compute the corresponding correlation coefficient. Therefore, realizing multiscale analysis within the Local DA framework is easy.

The multiscale localization proposed in this subsection gradually diminishes the contribution of small-scale covariance as the distance between two variables increases while retaining that of large-scale covariance until the distance is very large. Table 2 shows an example of multiscale localization. In this example, there are two arbitrary variables of which the error samples are decomposed into three scales. The values of covariance between the two variables are $C_1$, $C_2$, and $C_3$ at three scales. When the two variables are close (8 km), the localization coefficients of $C_2$ and $C_3$ are 1.0, according to the first formula in equation (14). As the distance increases to 300 km, the localization coefficients of $C_1$ and $C_2$ become nearly zero, and the total covariance is mainly attributable to $C_3$. Note that the multiscale covariance proposed in this section naturally excludes cross-scale covariance and is hard to incorporate cross-scale localization. How to determine the localization between two scales is also a question. The existing cross-scale localization (e.g., Huang et al., 2021; Wang et al., 2021) is implemented in spectral space and cannot be directly applied in equations (15) and (16). We plan to deal with the cross-scale issue in future work.

In addition to multiscale localization in the model space, Local DA can perform localization in the observation space, similar to LETKF. Observation space localization is conducted by enlarging the observation error as the distance between variables increases. The localization coefficient in the observation space is calculated according to the second formula of Eq. (14), but $d$-$d_{\min}(l)$ and $d_{\min}(l)$ are replaced by $d$ and $d_o$, respectively, where $d_o$ is the localization radius that varies among different observation types.

Because $\mathbf{C}_{\mathrm{oo}}$ and $\mathbf{R}$ are independently localized, Local DA can perform both localizations synchronously. Although performing localization in both spaces may result in a double penalty, it would be interesting to note the performance of the double-space localization approach, which has not yet been investigated. The related experiments and results are given in the following sections.

## 2.3 The hybrid covariance

The current version of Local DA calculates a simple "static" correlation matrix by using the second formula of Eq. (14), except that $d-d_{min}(l)$ and $d_{min}(l)$ are replaced by $d$ and $d_s$, respectively, where $d_s$ is a preset radius. For the $i$th and $j$th variables, the hybrid correlation coefficient $c(i,j)$ in $\mathbf{C}_{oo}$ is computed according to

275
$$c(i, j) = \gamma \sum_{l=1}^{N_b} r(l,i,j) \frac{\text{cov}\left[ \mathbf{X}_b^l(i), \mathbf{X}_b^l(j) \right]}{s(i)s(j)} + (1-\gamma)e^{-8\left[\frac{d}{d_s}\right]^2}, \tag{17}$$

where $\gamma$ is the weight of the dynamic correlation. The hybrid $c(i,j)$ in $\mathbf{C}_{mo}$ is also computed according to Eq. (17), but $\mathbf{X}_b^l(i)$ and $s(i)$ represent the variable at the model grid point. To prevent $s(i)$ and $s(j)$ in Eq. (17) from being forced to zero (which often occurs for convective-related variables such as the mixing ratios of rainwater, snow, and graupel), we add small, random perturbations with an STD of $1 \times 10^{-7}$ to the variables for which the STDs are smaller than $1 \times 10^{-7}$.

280 Note that the static part of equation (17) represents merely a distant correlation. It is valid for the univariate correlation rather than the cross-variable scenario. Therefore, the static part of equation (17) is forced to zero if the $i$th and $j$th variables are different types of variables. In other words, the cross-variable correlation is contributed only by the ensemble part. The authors acknowledge that the cross-variable correlation is important for DA, but the static cross-variable correlation must be carefully modeled, such as the correlation between wind components and geopotential height, or between the stream function

285 and potential temperature. The modeling work is in progress.

## 2.4 The workflow of Local DA

Here, we present a step by step description of how the hybrid and multiscale analyses described in the previous sections are performed for all the model variables. There is a way for Local DA to perform analysis much faster; we will discuss this method later.

290 1) Apply a bandpass filter to decompose $\mathbf{X}_b'$ into $N_b$ scales.

2) Store the background model state, decomposed samples, and observations in separate arrays denoted by $\mathbf{x}^f$, $\mathbf{X}_b'$, and $\mathbf{y}^o$, respectively.

3) For each model variable to be updated, search its ambient observations according to their scales and store these observations in array $\hat{\mathbf{y}}^o$; for example, search for sounding data within 300 km while searching for radar data within 15 km.

295 In addition, according to the observation operators of $\hat{\mathbf{y}}^o$, store the observation-associated model variables that have been projected onto observed grids/columns into arrays denoted by $\hat{\mathbf{x}}^f$ and $\hat{\mathbf{X}}_b'$, respectively.

4) Calculate the vector $\mathbf{d}$ in Eq. (1) with $\hat{\mathbf{y}}^o$ and $\hat{\mathbf{x}}^f$.

5) Use $\hat{\mathbf{X}}'_b$ to generate $\mathbf{S}_o$, $\mathbf{C}_{oo}$, $\mathbf{S}_m$, and $\mathbf{C}_{mo}$ according to Eqs. (12), (15), and (16).

6) Compute $\alpha$ for $\mathbf{C}_{oo}$ by using Eq. (4).

7) Compute $\hat{\mathbf{X}}_o = \alpha \mathbf{S}_o \mathbf{C}_{oo}$.

8) Calculate $\mathbf{Y} = \mathbf{R}^{-0.5}\mathbf{H}_o\hat{\mathbf{X}}_o$ by using Eq. (9).

9) Use the CG method to solve $(\mathbf{I} + \mathbf{Y}^T\mathbf{Y})\mathbf{v}_o = \mathbf{Y}^T\mathbf{R}^{-0.5}\mathbf{d}$ and obtain $\mathbf{v}_o$.

10) Compute the model state increment $\mathbf{x}_m$ according to Eq. (2).

In step 1), there are many ways to realize the bandpass filter. In this study, the difference between two low-pass analyses defines the bandpass field (Maddox, 1980), where the low-pass filter is the Gaussian filter. An example of a bandpass field is shown in Figure 2. For convenience, the radius of the Gaussian filter is used to represent the scale in this study. For the scale of 0 km - 20 km (Figure 2a), the small-scale feature prevails and corresponds to convection in the simulated typhoon. As the radius increases (Figure 2b), larger-scale information is extracted. A large-scale anticyclonic shear is observed when the radius is greater than 200 km (Figure 2c). The results (Figure 2d-f) also show that the contribution of the small-scale ensemble spread is often less than 10% out of the convective area, while in most areas of the forecast domain, the contribution of the large-scale (> 200 km) spread is greater than 20%.

Steps 5) to 9) contribute the most to the computational cost of Local DA. Computing $\mathbf{C}_{oo}$ requires $MK^2$ operations, which is not less than $N_o^2$, where $M$ represents the size of the ensemble, and $N_o$ denotes the number of observations to be assimilated. Step 7) requires $2K^2$ operations. To calculate step 8), $N_oK^2$ operations are needed. For each iteration step of the CG method, the number of operations is slightly larger than $2N_oK$. $N_i$ iteration steps require $2N_i N_oK$ operations.

As mentioned above, Step 9) can also be solved through eigenvalue decomposition as the LETKF does. However, $\mathbf{Y}$ in Local DA has more columns than the LETKF. In the LETKF, $\mathbf{Y}$ has $M$ columns, while the corresponding value is $K$ in Local DA. Therefore, Local DA has to deal with a $K$ by $K$ matrix, while the LETKF only needs to solve an $M$ by $M$ matrix. $M$ is often smaller than $10^2$, thus, $\mathbf{I}+\mathbf{Y}^T\mathbf{Y}$ can be handled efficiently by eigenvalue decomposition. In contrast, $K$ could be $10^3$ or higher, thus, the CG method is more suitable.

Despite the large amount mentioned above, we do not have to do that many operations in practice. For example, step 8) requires just $N_o^2$ operations if only scalar observations are available. Notably, for a 3-D domain containing $N_g$ grid points and $N_v$ variables, the total number of operations will be as $N_gN_v$ times that of one local analysis. However, it is possible to reduce the cost.

Considering that $\mathbf{S}_m$, $\mathbf{C}_{mo}$, and $\mathbf{x}_m$ can be applied to all variables influenced by $\hat{\mathbf{y}}^o$, it is not necessary to compute $\mathbf{C}_{oo}$ for each model variable. Moreover, $\mathbf{S}_m$, $\mathbf{C}_{mo}$, and $\mathbf{x}_m$ may contain variables in more than one vertical column ($N$-column analysis). The total number of operations in an $N$-column analysis is reduced to $N_g/(NN_z)$ times as one local analysis, where $N_z$ is the number of levels in one column. Due to using the same $\mathbf{C}_{oo}$ for neighboring columns, the $N$-column analysis is slightly rasterized (not shown), leading to slightly higher errors than the 1-column analysis. However, the extent of this

degeneration is acceptable as long as $N$ is not too large (<9). The wall clock time of the $N$-column analysis is close to $1/N$ of the 1-column analysis. All Local DA results are generated using a 5-column analysis in this study. A similar $N$-column analysis approach is the weighted interpolation technique in the LETKF (Yang et al., 2009), which performs LETKF analysis every 3 grid points in both the zonal and meridional directions.

## 3 Experimental design

### 3.1 The simulated typhoon

The third typhoon of the 2021 western Pacific season, In-Fa, is selected for the OSSEs performed herein. The true simulation, starting at 00 UTC on 25 July 2021 and ending at 18 UTC on 26 July 2021, simulates the stage in which In-Fa approaches China. The Weather Research and Forecast (WRF, Skamarock et al., 2018) model V3.9.1 is used for the simulation. The central latitude and longitude of the forecast domain are 30.5 ° and 122.0 °, respectively. The domain size is

201 grids ×201 grids×34 levels with a horizontal resolution of 5 km and a model top pressure of 50 hPa. The physical parameterization schemes are as follows. The WRF single-moment 6-class ice scheme (Hong and Lim, 2006) is adopted for microphysical processes. For longwave and shortwave radiation, the rapid radiative transfer model (RRTM) scheme (Mlawer et al., 1997) and the Dudhia scheme (Dudhia, 1989), respectively, are used. The Yonsei University (YSU) scheme (Hong et al., 2006) is employed for the planetary boundary layer simulation. For the cumulus parameterization, the Kain–Fritsch (new

Eta) scheme (Kain, 2004) is enabled. The unified Noah land surface model is used to simulate the land surface. We adopt the global forecast system (GFS) analysis at 00 UTC on 25 July 2021 as the initial condition of the Truth simulation.

According to Hoffman and Atlas (2016), a criterion for reasonable OSSEs is that true simulation agrees with the real atmosphere. The typhoon central pressure in the Truth simulation gradually increases from 968 hPa to 980 hPa by 18 UTC on 26 July 2021 (not shown), which is consistent with the real observation obtained from the China Meteorological

Administration (CMA), except that the observed pressure increases more rapidly, reaching 985 hPa by 18 UTC on 26 July 2021. The simulated typhoon's central location also agrees with the CMA observation. Therefore, the Truth simulation is eligible for OSSEs.

### 3.2 Multiscale observation network

The simulated multiscale observation network (Figure 3) comprises sounding, wind profiler, PWV, and radar observations.

Soundings are available at 00 UTC and 12 UTC on 26 July 2021, whereas the other types of observations are available hourly on 26 July 2021.

For each sounding, we simply extract the perturbed model variables, $u$, $v$, $\theta$, and $q_v$, every 2 model levels as the observations. The simulated soundings also record the perturbed surface pressure, $ps$. The sounding perturbations follow a Gaussian distribution with zero mean. The perturbation STDs are 0.5 m s$^{-1}$, 5 m s$^{-1}$, 0.5 K, 5×10$^{-5}$ kg kg$^{-1}$, and 10 Pa for $u$, $v$, $\theta$, $q_v$, and

*ps*, respectively. To better reflect reality, no simulated soundings are available over the ocean, and the horizontal resolution of each sounding is 100 km.

The simulated wind profiler provides data on horizontal wind components, *u* and *v*, at all model levels. The perturbations added to the wind profiler data follow a Gaussian distribution with zero mean and an STD of 0.5 m s$^{-1}$. The wind profilers, the data from which have a horizontal resolution of 50 km, provide data only on land.

The PWV observations are computed according to

$$PWV = \frac{1}{g} \int_{p1}^{p2} q_v dp , \qquad (18)$$

where *g* is the gravitational constant of acceleration and *p*1 and *p*2 represent the bottom and top of a model column, respectively. Perturbations with zero mean and an STD of 0.5 kg m$^{-2}$ are added to the PWV observations. Because the PWV is observed by satellites, this type of observation is available for the whole forecast domain, and the horizontal observation

interval is 50 km in both the *x* and *y* directions.

The radar data to be assimilated are radial velocity and reflectivity. We adopt Eq. (3) of Xiao and Sun (2007) to compute the radial velocity, but we ignore the terminal velocity in OSSEs. For reflectivity, the operator proposed by Gao and Stensrud (2012) is employed. Three radars located at approximately Shanghai (121.48 °E, 31.23 °N), Hangzhou (120.16 °E, 30.28 °N), and Ningbo (121.55 °E, 29.88 °N) are simulated with a maximum observation range of 230 km. The simulated radars work

on the volume coverage pattern (VCP) 11 mode, which has 14 elevation levels from 0.5 ° to 19.5 °. Radar data are created on volume-scan elevations, but they are on model grids in the horizontal direction, as shown in Xue et al. (2006). The radial velocity and reflectivity observation errors are 1.0 m s$^{-1}$ and 2.0 dBZ, respectively. The horizontal resolution of the radar data is identical to the model grid spacing.

In total, 2795 simulated soundings, 400 PWV data points, 5332 wind profiler observations, and 391618 radar observations

(including radial velocity and reflectivity) are utilized in this study.

**3.3 DA experiments**

In this study, two sets of experiments are designed. The first set of experiments consists of single deterministic analyses and is used to examine the impact of the hybrid covariance, the multiscale localization in model space, and the double-space localization. The other set of experiments comprises several cycling analyses, mainly focusing on the analysis balance (in

terms of surface pressure tendency) and the impact of Local DA on cycling analysis. To perform the analysis with ensemble covariance, it is necessary to generate the ensemble first. Therefore, in this subsection, we first describe the generation of the ensemble and then introduce the experimental design.

### 3.3.1 Ensemble perturbations

For the single deterministic analysis, the time-lagged approach (e.g., Branković et al., 1990) is employed to generate the ensemble perturbations, which are created by using deterministic forecasts with different initial times and varying GFS data. For example, the first sample at 00 UTC on 26 July 2021 stores the difference between two deterministic forecasts initialized at 06 UTC on 25 July 2021 and 12 UTC on 25 July 2021. To distinguish these forecasts from the forecasts of the DA experiments, the forecasts used to produce ensemble members are referred to as sample forecasts. The sample forecasts used in this study are shown in Figure 4a. Note that some sample forecasts are initialized by the 3-h or 6-h GFS forecast data

(highlighted by the thick tick marks in Figure 4). A small size ensemble is employed; it combines 6 sample forecasts according to $C(6,2) = \dfrac{6!}{2!4!}$ and thus has 15 members.

Focusing on the result of a small size ensemble is based on two concerns. First, Local DA is designed as a flexible scheme for hybrid analysis; hybrid analysis is often beneficial in the presence of a small ensemble or a poor ensemble. In the case of using a well sampled ensemble, the pure ensemble DA is preferred. Second, the available computational resources are not

always sufficient to support a large size ensemble. The authors have tested a larger ensemble with 36 members and obtained lower analysis errors than the 15-member counterpart. For brevity, the results with the 36-member ensemble are not shown. For the cycling analysis, the first analysis uses the time-lagged 15-member ensemble. In the remaining cycles, the ensemble forecast initialized from the previous analysis ensemble provides the ensemble perturbations. The analysis ensemble is created by performing Local DA 15 times with perturbed observations. The perturbations are added to Ctrl so that the

ensemble center on Ctrl. The Ctrl in the first cycle is obtained using GFS analysis at 00 UTC on 26 July 2021. Figure 4b shows the flowchart of the cycling DA.

### 3.3.2 The DA configurations

A total of 14 experiments for deterministic analyses at 00 UTC on 26 July 2021 are examined. The first three experiments investigate the influence of using the pure ensemble covariance (Ens_noFLTR), distant correlation covariance (Static_BE),

and hybrid covariance (Hybrid_noFLTR) on the Local DA analysis. The model variables to be analyzed are the three wind components ($u$, $v$, $w$), potential temperature ($\theta$), water vapor mixing ratio ($q_v$), dry-air mass in column ($mu$), and hydrometeor mixing ratios ($q_c$, $q_r$, $q_i$, $q_s$, and $q_g$). A fixed localization radius of 200 km is used for most variables. For $ps$ and hydrometeor variables ($q_c$, $q_r$, $q_i$, $q_s$, and $q_g$), the fixed influence radii are 1000 km and 20 km, respectively. These values are tuned for the case in which Typhoon In-Fa made landfall in this study and are only used for static correlation and experiments without

multiscale localization (e.g., Ens_noFLTR). The background error covariance is empirically inflated by 50%. For Hybrid_noFLTR, the weight between the dynamic and static covariances is 0.5. Then, the impact of model-space multiscale localization is evaluated through 6 experiments with/without the hybrid covariance. Ens_2band, Ens_3band, and Ens_5band use the pure ensemble covariance, but the ensemble is decomposed into

2, 3, and 5 scales, respectively. The 2-band experiment uses samples with a scale of 0 km - 200 km and a scale greater than

200 km. In this experiment, the contribution of a scale greater than 200 km is amplified because the localization coefficient is 1.0 until the distance between two grid points is greater than 200 km. For the Ens_3band, the three scales are 0 km - 50 km, 50 km - 200 km, and >200 km. The corresponding values for Ens_5band are 0 km - 20 km, 20 km - 50 km, 50 km - 100 km, 100 km - 200 km, and >200 km, respectively. Through the above three experiments, we can examine the sensitivity of Local DA to the configuration of multiscale analysis. Hybrid_2band, Hybrid_3band, and Hybrid_5band use the same

ensemble covariance as Ens_3band, and Ens_5band, respectively; the ensemble covariance and static covariance weight equally in the hybrid covariance.

The last five experiments are designed to discuss the impact of the localization space. Ens_noFLTR_OL performs localization in observation space. The horizontal radii are 360 km, 150 km, 120 km, and 15 km for sounding, wind profiler, PWV, and radar data, respectively. Notably, Ens_noFLTR_OL performs vertical localization in model space, identical to

Ens_noFLTR. Ens_LETKF uses the LETKF algorithm and the same horizontal localization radii as Ens_noFLTR_OL. The vertical radius for all observations is 5 km in Ens_LETKF, where the PWV observations are treated as being located at 4000 m for LETKF localization. Ens_noFLTR_DSL performs localization in both the model and observation space. In the model space, a fixed localization radius is used, as in Ens_noFLTR, while the localization parameters of Ens_noFLTR_OL are adopted for observation-space localization. By using 5-band samples, Ens_noFLTR_DSL becomes Ens_5band_DSL.

Adding hybrid covariance to Ens_5band _DSL yields Hybrid_5band_DSL. For convenience, all single deterministic analysis experiments are listed in Table 3, where "M", "O", and "M+O" denote model-space, observation-space, and double-space localization, respectively. The vertical localization in the observation space is disabled for all Local DA experiments.

For experiments with cycling analysis, we examine Local DA in the cases of i) using the ensemble covariance without multiscale localization and ii) using hybrid covariance and multiscale localization. The DA configuration of Ens_noFLTR is

employed for the first scenario, while that of Hybrid_5band_DSL is adopted for the second scenario. Cycling intervals of 3-h and 6-h are examined, where we mainly focus on the experiments with the 6-h interval. The experiment with a 3-h cycle interval is used to show the impact of imbalance analysis to forecast. A total of three experiments are examined, namely, Ens_noFLTR_6h, Hybrid_5band_DSL_6h, and Hybrid_5band_DSL_3h, where the suffixes represent the cycling intervals. During cycling, sounding observations are available at 00 UTC and 12 UTC, while other observation types are available

hourly. Fifteen sets of perturbed observations are created to update 15 members in cycling DA. The standard deviations of observation perturbations are identical to the observation errors mentioned in sect. 3.2. The covariance inflation factor is also 1.5 for cycling analysis.

## 4 Results and discussion

### 4.1 The convergence of minimization

We examine the minimization convergence by using the data extracted from Hybrid_5band. Figure 5 shows the number of iterations and the ratio of the final value of the cost function ($J_{final}$) to the initial value ($J_{initial}$). Fewer than 100 iterations indicate that the tolerance $\varepsilon^2$ reaches $1 \times 10^{-6}$ within 100 steps. If the minimization does not converge within 100 steps, the CG iteration is stopped by the program. The number of iterations is large near the center of the forecast domain but decreases rapidly outward. According to the distribution of observations (Figure 3), the results (Figure 5a) indicate that the

minimization converges more slowly as the number of observations to be assimilated increases.

Although the minimization fails to converge within 100 steps in the area where the observation density is high, the cost function is reduced by 70% or 80% (Figure 5b). In contrast, near the northeastern and southeastern corners of the domain, where the minimization converges within 10 steps, the final value of the cost function is greater than 70% of its initial value. However, in those areas, the initial cost function is small, implying no need for a large extent of correction. The results also

indicate that no severe discontinuity occurs in Hybrid_5band, which is desired. Similar to the LETKF, using slightly different $\mathbf{C}_{oo}$ between neighboring columns does not yield remarkably different analyses.

Further investigation (for data within the yellow rectangle plotted in Figure 5a) indicates that approximately 25% of minimizations fail to converge within 100 steps (Figure 6a), all associated with the application of radar data. Therefore, we rerun Hybrid_5band using only radar data and observe that only 4% of all minimizations require more than 100 steps to

converge. In the case of setting the maximum number of iterations to 500 for Hybrid_5band, all minimizations converge within 300 iteration steps. The results also show that assimilating only radar data produces a smaller ratio of $J_{final}$ to $J_{initial}$ than the case using all observations (Figure 6b). According to previous studies (e.g., Wang and Wang, 2017), the inefficient minimization may be caused by the assimilation of radar reflectivity due to the use of the mixing ratios as state variables. Too small hydrometeor mixing ratio values can lead to an overestimated cost function gradient. Nevertheless, despite the

slow convergence, Local DA reduces the cost function by more than 70% within 100 iteration steps in most cases (Figure 6b). Further suppressing the error may require a better background error covariance, which we plan to seek in future work.

### 4.2 The single deterministic analysis

The domain averaged root mean square root error (RMSE) is examined first. For convenience, the initial condition extracted from GFS analysis is referred to as BAK. All experiments reduce the errors in the observation space after DA, but their

differences are significant (Figure 7). The experiments (Ens_noFLTR, Ens_noFLTR_OL, Ens_LETKF, and Ens_noFLTR_DSL) without the hybrid covariance and model-space multiscale localization produce relatively higher analysis errors than other experiments for wind components, temperature, radial velocity, and reflectivity. Using distance correlation (Static_BE) results in lower errors than Ens_noFLTR for most variables, while Hybrid_noFLTR further

suppresses the errors except for reflectivity. The benefit of using hybrid covariance is consistent with many previous studies (e.g., Wang et al., 2009; Wang et al., 2013b; Tong et al., 2020).

Model-space multiscale localization (Ens_2band, Ens_3band, and Ens_5band) is conducive to error reduction. Even with 2-scale samples, Ens_2band dramatically reduces the errors of wind-related variables, compared with Ens_noFLTR. Involving more scales further improves the analysis, but the benefit is not as great as the case of comparing Ens_noFLTR with Ens_2band. Combining the hybrid covariance and model-space multiscale localization does not further narrow the gap between the analysis and observation.

Double-space localization does not necessarily ensure small analysis errors (Ens_noFLTR_DSL). However, when the localization is combined with the hybrid covariance and model-space multiscale localization (Hybrid_5band_DSL and Ens_5band_DSL), the analysis error can be substantially reduced, especially for PWV and reflectivity.

In model space, similar results can be observed (Figure 8). The hybrid covariance, model-space localization, and double-space localization are helpful for error reduction. Notably, unlike the result in the observation space, the analysis errors in some experiments are higher than that of BAK. Because the RMSE in model space counts for grid points that are not directly observed and are updated through error covariance, the error becoming higher after DA is likely due to the poor error covariance in model space.

In the following subsections, the background and analysis errors in model space are decomposed into three scales by using a Gaussian filter with radii of 50 km and 200 km, respectively, representing errors of the small scale (0 km - 50 km), middle-scale (50 km – 200 km), and large scale (>200 km). Through this decomposition, we can investigate the results in detail. The vertical velocity ($w$) and hydrometeor variables ($q_c$, $q_r$, $q_i$, $q_s$, and $q_g$) are not decomposed because their scales are often small. In addition, convective-scale DA usually computes the errors for grid points with reflectivity larger than a threshold, which is another way to investigate small-scale errors. The difference between errors in the convective area (reflectivity >10 dBZ) and the rest area is similar to that between small-scale and large-scale errors (not shown). Therefore, the errors in the convective area are not discussed in the subsequent sections.

### 4.2.1 Hybrid analysis

Figure 9 shows that the smallest scale error contributes most to the background and analysis error, while the quantities of large-scale errors are often half of their small-scale counterparts. Ens_noFLTR reduces errors at all scales for horizontal wind components, where the error reduction is relatively higher at a large scale. For $T$, $q_v$, and $ps$, Ens_noFLTR suppresses the large-scale errors but amplifies the small-scale ones. This result implies that the large-scale error covariance is likely reliable but the smaller one is not.

When the static correlation is enabled for Local DA (Static_BE and Hybrid_noFLTR), the small-scale and middle-scale errors are substantially decreased. This difference becomes much larger for $ps$ when Ens_noFLTR is compared with Static_BE, even at a large scale. The analysis errors of Static_BE and Hybrid_noFLTR are nearly identical at all scales for $u$, $v$, $T$, and $q_v$, but the reason for this phenomenon is still unknown. We plan to determine the cause in future work. Overall, the

main contribution of employing static correlation to the lower analysis errors of Static_BE and Hybrid_noFLTR is at a small scale. The result implies that constraining the small-scale ensemble correlation in a small radius may be conducive to the small analysis error, which is what the model-space multiscale localization does.

### 4.2.2 Multiscale analysis

After decomposing the ensemble samples into two parts (Ens_2band) and independently applying the localization radius for each scale, the small-scale analysis error becomes lower than that of Ens_noFLTR for all examined variables (Figure 10). Compared with Ens_2band, further decomposing the ensemble samples into more scales (Ens_3band and Ens_5band) and using smaller radii for small scales slightly reduces the analysis error for wind components and surface pressure but increases the error for $q_v$. This result confirms the assumption that restricting the impact of small-scale correlation in a small region is beneficial. The difference between Ens_3band and Ens_5band is small, indicating that three or five scales should be sufficient for the model-space multiscale localization in Local DA.

Experiments combining multiscale localization with hybrid covariance (Hybrid_2band, Hybrid_3band, and Hybrid_5band) produce lower analysis errors for most variables, compared with Ens_2band, Ens_3band, and Ens_5band. However, the improvement is not substantial. The small difference implies that we need more approaches to make further improvements. Employing double-space localization is one of the approaches, according to the result shown in Figure 8.

### 4.2.3 Double-space localization

Compared with Ens_noFLTR, Ens_noFLTR_DSL has a small but positive impact on the analysis of $u$, $v$, $T$, and $q_v$ at a small scale, while its influence on larger scale errors is negligible (Figure 11). In contrast, Ens_noFLTR_DSL substantially reduces the analysis error of $ps$ at all scales. After combining the model-space localization (Ens_5band_DSL), the analysis errors further decline at a small scale. Adding a hybrid covariance to Ens_5band_DSL (Hybrid_5band_DSL) leads to lower analysis error for most variables. The large-scale analysis error of $ps$ is increased after using hybrid covariance, implying that the large-scale error correlation related to $ps$ and computed by using ensemble samples is better than the distant correlation with a fixed influence radius. It is encouraging to see that Hybrid_5band_DSL and Ens_5band_DSL produce the analysis error of $q_v$ lower than BAK at small and middle scales, while Ens_5band and Hybrid_5band yield a higher analysis error than BAK. The result indicates the benefit of double-space localization.

To qualitatively assess the analysis error, we compute the difference in total energy (DTE, Meng and Zhang, 2007). Wang et al. (2012) used the square root of the mean DTE to evaluate the error of DA to simplify the presentation. The DTE is computed in the form of the difference between the analysis and truth. Ens_noFLTR (Figure 12 d-f) decreases the background errors (Figure 12 a-c) at 850 hPa and 500 hPa but generates many spurious increments over the ocean, increasing the error there; this problem is more pronounced at 200 hPa. Accordingly, the error after Ens_noFLTR analysis is still high. The spurious increment corresponds to the large analysis error at a small scale. In contrast, utilizing the hybrid covariance and model-space multiscale localization suppresses the small-scale spurious errors (Hybrid_5band, Figure 12 g-i)

from the lower to the upper levels. The spurious increment is further reduced in Hybrid_5band_DSL, especially at 850 hPa
and 500 hPa, indicating that the positive impact of double-space localization corresponds to less noise in the analysis. According to the above result, double-space localization may serve as a supplement to pure model-space localization which determines the level of analysis error.

### 4.2.4 The similarity between Local DA with observation space localization and the LETKF

Considering that Local DA can perform observation space localization only as in the LETKF, it is interesting to see if their
analyses are similar. Note that Ens_noFLTR_OL and Ens_LETKF merely share the same horizontal localization configuration; they differ in vertical localization. Figure 13 shows that the difference in analysis error between Ens_noFLTR_OL and Ens_LETKF is small for all variables and at all scales. Figure 14 gives an intuitive comparison between the Ens_noFLTR_OL and Ens_LETKF analyses. The overlarge negative-increment in both experiments is constrained in a much smaller area than Ens_noFLTR (marked by red rectangles in Figure 14). They also suppress the small-
scale noise in the Ens_noFLTR analysis, corresponding to the lower error in Figure 13e. Overall, in the case of using observation-space localization, Local DA can produce an analysis similar to the LETKF.

In addition, the small-scale error of $q_v$ yielded by Ens_noFLTR_OL is lower than that of Ens_noFLTR (Figure 13d). The result is similar to the difference between Ens_noFLTR_DSL and Ens_noFLTR, indicating that the improvement of Ens_noFLTR_DSL on $q_v$ analysis compared with Ens_noFLTR is mainly attributable to observation-space localization.

### 4.2.5 Error and ensemble spread

For a well-sampled ensemble, a criterion is that the spatial distribution of the ensemble spread is similar to that of RMSE. In addition, the amplitudes of the ensemble spread must be close to the RMSE. The relationship is shown in Figure 15 for the time-lagged ensemble at 00 UTC on 26 July 2021. For $u$, $v$, and $ps$, the ratio of ensemble spread to RMSE ascends as the error scale increases, indicating that the quality of the time-lagged ensemble is rational at a large scale. This relationship is
also valid for the spatial distribution (Figure 15b), but the correlation coefficient does not vary from small scale to large scale too much for most variables, except for $ps$. The correlation coefficient for $ps$ is nearly 1.0 at a large scale, while it is approximately 0.6 at a small scale. This large difference explains why the hybrid covariance and multiscale localization can substantially reduce the error at a small scale for $ps$. For $q_v$, the small-scale spread is greater than the large-scale spread; the correlation coefficients at all scales are close. This result implies that suppressing the small-scale error covariance does not
necessarily improve the analysis quality of $q_v$. Therefore, it is not irrational for Ens_5band and Hybrid_5band to produce a higher analysis error for $q_v$ than Ens_2band.

An example related to the ensemble spread and RMSE of $ps$ is shown in Figure 16. The RMSE is smooth at a small scale, and there is a maximum near the typhoon center. Although the ensemble spread also has a maximum near the typhoon center, there is a large bias concerning the location. Moreover, the ensemble spread is much noisier than the RMSE, which is a
cause of the noisy analysis shown in Figure 14b. In contrast, the large-scale ensemble spread matches the error well, which is

conducive to error reduction. Therefore, even with a large localization radius, the surface pressure analysis of Ens_noFLTR at a large scale is not much worse than that of the other experiments.

## 4.3 The cycling DA

Because ensemble DA approaches often take several cycles to obtain a reasonable analysis, it is worth seeing if
Ens_noFLTR produces a better analysis after some cycles and if Hybrid_5band_DSL maintains the advantage in cycling DA. Before looking at the RMSE evolution during cycling, the *ps* tendency is examined as it is a metric of dynamic imbalance (Zeng et al., 2021). If the unphysical *ps* tendency is large, the analysis may be degenerated, and the forecast could be unstable. Although it is better to analyze the *ps* tendency at each time step, in this study, the hourly *ps* tendency is sufficient to demonstrate the impact of imbalance analysis. The forecast from GFS analysis is referred to as BAK in this subsection.

### 4.3.1 The tendency of *ps*

The *ps* tendency in the truth simulation is selected as a criterion as it is assumed to be in balance status after a 24-h forecast. The balanced tendency is approximately 20 Pa h$^{-1}$ (Figure 17), which is reached by BAK in 3 h. After the first DA cycle, the *ps* tendency becomes much larger than that of BAK, no matter the DA configuration. The large *ps* tendency after the first DA cycle is not surprising because the landing typhoon is not fully observed by the simulated observation network,
especially for the wind field, causing an imbalance between the corrected part and the rest of the analyzed typhoon. A similar phenomenon was discussed by Wang et al. (2012) in a simulated supercell case. They concluded that such an imbalance shocks the model forecast and increases the forecast error.

After a 6-h forecast, the *ps* tendencies in Hybrid_5band_DSL_6h and Ens_noFLTR_6h are close to the balance status. As expected, the *ps* tendency increases again after the second DA cycle. However, Hybrid_5band_DSL_6h produces a much
smaller *ps* tendency than Ens_noFLTR_6h, indicating that Hybrid_5band_DSL_6h has a more balanced analysis. The peaks of *ps* tendency in Hybrid_5band_DSL_6h and Ens_noFLTR_6h gradually decline as the number of cycles increases. By 18 UTC, Hybrid_5band_DSL_6h reaches the balance status while Ens_noFLTR_6h does not. The above result indicates that using the hybrid covariance and multiscale localization is beneficial for cycling DA.

Note that the advantage of Hybrid_5band_DSL_6h has a precondition that the cycling interval is sufficiently long for the
model to spin up. When the cycling interval becomes shorter (Hybrid_5band_DSL_3h), the *ps* tendency cannot be effectively suppressed as Hybrid_5band_DSL_6h does.

### 4.3.2 The performance of cycling DA

We only discuss the results of *u*, *v*, *q*$_v$, and *ps* in this subsection for brevity. For *u* and *v*, all experiments reduce the forecast error compared with BAK (Figure 18a and b). However, the error evolutions of these experiments substantially differ.
Ens_noFLTR_6h fails to decrease the forecast error after the second cycle, while Hybrid_5band_DSL_6h successively reduces the forecast and analysis error as the number of cycles increases. For Hybrid_5band_DSL_3h, an oscillation in error

evolution is observed, which is likely associated with the imbalance analysis and the insufficient cycle interval for spinup. Despite the oscillation, the forecast and analysis errors of Hybrid_5band_DSL_3h are comparable to those of Hybrid_5band_DSL_6h for wind components.

However, in regard to water vapor and surface pressure (Figure 18c and d), Hybrid_5band_DSL_6h becomes better than Hybrid_5band_DSL_3h. Hybrid_5band_DSL_6h also outperforms Ens_noFLTR_6h; the latter fails to suppress the forecast error of $q_v$ and produces a higher $ps$ error after analysis. Figure 19 shows the spatial distribution of forecast error at 18 UTC for Hybrid_5band_DSL_6h and Ens_noFLTR_6h. The area of large error in Hybrid_5band_DSL_6h is much lower than that of Ens_noFLTR_6h for both $v$ and $ps$. The large error in Ens_noFLTR_6h corresponds to a weak cyclonic rotation and weak

low pressure. The above result confirms the benefit of using the hybrid covariance and multiscale localization.

### 4.3.3 The evolution of the relationship between ensemble spread and RMSE

For Hybrid_5band_DSL_6h, the initial ensemble spread is smaller than the RMSE at all scales (Figure 20a) for both $u$ and $ps$. As the number of cycles increases, the ratio of ensemble spread to RMSE increases. By 18 UTC, the ensemble spread is comparable to or greater than the corresponding RMSE at all scales for $u$. The underestimation of RMSE by the ensemble

spread is alleviated for $ps$ (Figure 20b). For the spatial distribution, the relationship between the ensemble spread and RMSE does not vary much for $u$ at all scales (Figure 20c). In contrast, the relationship becomes better for $ps$ at a small scale (Figure 20d). Overall, the ensemble is improved in Hybrid_5band_DSL_6h.

For Ens_noFLTR_6h, the ensemble spread of $u$ and $ps$ at the small-scale remains smaller than the corresponding RMSE during the cycling DA. In contrast, the ensemble spread at the large scale dramatically increases after the second cycle. The

amplitude of the large-scale ensemble spread is even higher than that of the small-scale spread, leading to a severe overestimation of the large-scale error. Meanwhile, the correlation between ensemble spread and RMSE at the small scale is not improved during cycling. In general, the ensemble in Ens_noFLTR_6h does not become better after four cycles, which explains why Ens_noFLTR_6h produces a large analysis error.

### 4.4 The computational cost and efficiency

The computational cost and efficiency of Local DA are discussed in this subsection. All tests are conducted on a 36-core workstation with an Intel Xeon Gold 6139 CPU (the maximum frequency is set to 2.30 GHz) and 48 gigabytes of available memory. Heretofore, we have implemented the parallel Local DA with OpenMP, which is not suitable for large-scale parallel computing; however, for this study, OpenMP is sufficient. The parallel efficiency is examined first. LDA_HBC_MSL is selected as an example. Figure 21 shows the wall clock time as a function of the number of cores. The

wall clock time covers Local DA steps 3) through 9) (as described in Sect. 2d). As expected, the wall clock time is reduced by approximately 50% upon doubling the number of cores, which is valid if the number of cores is not greater than 16. In contrast, increasing the number of cores from 16 to 32 does not shorten the wall clock time; this is attributable to the fact that OpenMP is suitable only when the number of processors is small (<16) (Hoeflinger et al., 2001). Given that no messages

need to be passed between the cores for steps 3) through 9), the parallel efficiency of Local DA is likely insensitive to the number of cores. In general, the results demonstrate that Local DA can be highly parallelized.

In addition to its parallelization, the computational speed of Local DA is also investigated. Hybrid_5band takes 225 s to complete all local analyses when 16 cores are used. Note that the number of horizontal grid points within the forecast domain is 40000, and more than 200000 observations are assimilated. Given that the processors work at a frequency of 2.30 GHz, the computational speed of Local DA is acceptable. On average, nearly 70% of the computational time is used to compute $\mathbf{C}_{oo}$ and $\mathbf{C}_{mo}$; for the minimization using the CG method, the corresponding percentage is approximately 18%.

We also assess the memory consumption of Local DA. To complete Local DA steps 3) through 9), Hybrid_5band uses approximately 4 gigabytes when 16 cores are engaged to store $\mathbf{C}_{oo}$ and the associated matrices. In contrast, the LETKF uses only hundreds of megabytes. For each 5-column analysis, the $\mathbf{C}_{oo}$ size varies from 2000×2000 to 4500×4500, which is affordable. However, for a much larger size, such as 9000×9000, OpenMP is insufficient; under these circumstances, the MPI-OpenMP hybrid scheme is likely a viable solution for both the computational speed and the memory consumption, which is in progress. In addition to $\mathbf{C}_{oo}$, the model-space multiscale localization requires large memory. Memory consumption is proportional to the number of scales. For example, Ens_3band requires three times as much memory as Ens_noFLTR to store the decomposed perturbations. In general, the total computational cost of Local DA is high, but the cost of each local analysis is affordable.

## 5 Summary and conclusions

This study proposed a local data assimilation scheme (Local DA) that can utilize hybrid covariance and multiscale localization. Local DA explicitly computes a local background error correlation matrix and uses the correlation matrix to construct a local error sample matrix. The error sample matrix with proper localization allows Local DA to adopt the conjugate gradient (CG) method to solve the cost function. The constructed matrix also renders Local DA to be a flexible hybrid analysis scheme. Local DA is evaluated in a perfect model scenario that includes a simulated multiscale observation network for a typhoon case. We examined the impacts of the hybrid covariance and multiscale localization on Local DA and evaluated the performance of cycling DA. Several conclusions can be drawn from the results of the DA experiments:

i) Applying the CG method independently for each column group does not result in a severe discontinuity in the Local DA analysis;

ii) Explicitly computing the background correlation matrix projected onto observation-associated grids/columns is computationally affordable if the observations have been properly thinned;

iii) Local DA can effectively utilize the hybrid covariance to produce a better analysis than the analysis using ensemble covariance with a fixed localization radius;

iv) The model-space multiscale localization can reduce the analysis error at a small scale; combining the hybrid covariance
with the multiscale localization yields a small improvement; adding double-space localization to the combination can further reduce the analysis error;

v) Local DA requires a large amount of memory, but its computational efficiency is acceptable.

Despite the encouraging results, whether to use double-space localization should be considered case by case. In this study, the background error covariance is noisy, so double-space localization has a positive impact. With a well-sampled ensemble and a well-designed multiscale localization, there is no need to use double-space localization. In the case of applying Local DA in the four-dimensional DA scenario, double-space localization should not be used because observation-space localization does not consider the advection of error covariance.

As the first study to present Local DA, this paper focuses on its idea and basic formulation. Future efforts to enhance the algorithm will include developing an MPI-OpenMP hybrid parallel scheme, a static covariance scheme that objectively determines the error variance and scales, and a better multiscale localization scheme. Furthermore, the current version of Local DA introduces a strong shock to the model, which limits the applicability of Local DA in cycling DA. Therefore, we plan to add a cross-variable balance procedure to improve the cycling DA performance. Moreover, many parameters of Local DA have yet to be tested; hence, the sensitivity of Local DA to each of these parameters will also be discussed in a future investigation.

**Code and data availability**

The code of Local DA v1.0 and the scripts for running the experiments in this study is available at the following link:https://doi.org/10.5281/zenodo.6609906 or by contacting the corresponding author via e-mail. The GFS data are available at https://www.ncdc.noaa.gov/data-access/model-data/model-datasets/global-forcast-system-gfs.

**Author contributions**

Shizhang Wang performed the coding and designed the data assimilation experiments. Xiaoshi Qiao analyzed the experimental results. Both authors contributed to the writing of the paper.

**Competing interests**

The authors declare that they have no conflicts of interest.

## Acknowledgments

This work is jointly sponsored by the National Science and Technology Major Project of the Ministry of Science and Technology of China (2021YFC3000901, 2021YFC3000902), the National Natural Science Foundation of China (41875129, 41505090, and 42105006), and the Basic Research Fund of CAMS (2021R001, 2021Y006).

## Appendix

This section provides an example of the procedure used to thin the observations (as mentioned in Sect. 2b). The observations are thinned horizontally, whereas thinning does not occur in the vertical direction. First, we set several rings with different radii at the center point or column of the model variables to be updated. For the 5-column analysis, the center coordinates of the variable-radius rings are the mean latitude and mean longitude of the 5 columns. The radius of the outer ring is the observation search radius mentioned in Sect. 2d (e.g., 300 km for sounding data and 15 km for radar data). From small to large, the radii of the rings are denoted $rr_1$, $rr_2$,..., $rr_{Nr}$, where Nr is the number of rings. We successively search the observations from the inner ring to the outer ring. Within the smallest ring, all ambient observations are selected; this is equivalent to no thinning. For the observations located between two rings (between $rr_i$ and $rr_{i-1}$), we select one observation for each quadrant of the space between the two rings. There are four quadrants: the upper-right, lower-right, lower-left and upper-left quadrants (numbered I, II, III, and IV, respectively). A schematic plot is shown in Figure A1. If no observation is available in the smallest ring, the second ring is treated as the first ring.

Because no thinning occurs in the smallest ring, in a 1-column analysis, we still utilize all observations throughout the forecast domain when Local DA is conducted at a single point. In the 5-column analysis, the thinning approach discards some observations and slightly increases the analysis error relative to the 1-column analysis. Our early test (not shown) indicates that Local DA becomes very time-consuming when the thinning process is disabled, as expected. Moreover, the resulting analysis error increases because the assumption of observation errors being uncorrelated is not valid, which is not desired.

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

**Tables**

**Table 1 The dimensions of variables in Local DA**

| | Variable space | Variable type | dimension | |
|---|---|---|---|---|
| $\mathbf{x}^{\mathrm{f}}$ | Model space | Model variable | $N_{\mathrm{m}} \times 1$ | 840 |
| $\mathbf{X}$ | Model space | Model variable | $N_{\mathrm{m}} \times M$ | |
| $\mathbf{x}_{\mathrm{o}} = \mathbf{H}_{\mathrm{i}}\mathbf{x}^{\mathrm{f}}$ | Observed grids/columns | Model variable | $K \times 1$ | |
| $\mathbf{X}_{\mathrm{o}} = \mathbf{H}_{\mathrm{i}}\mathbf{X}$ | Observed grids/columns | Model variable | $K \times M$ | 845 |
| $\mathbf{C}_{\mathrm{oo}}$ | Observed grids/columns | Model variable | $K \times K$ | |
| $\mathbf{v}_{\mathrm{o}}$ | Observed grids/columns | Model variable | $K \times 1$ | |
| $\mathbf{S}_{\mathrm{o}}$ | Observed grids/columns | Model variable | $K \times 1$ | 850 |
| $\mathbf{C}_{\mathrm{mo}}$ | Cross space | Model variable | $N_{\mathrm{m}} \times K$ | |
| $\mathbf{S}_{\mathrm{m}}$ | Model grid space | Model variable | $N_{\mathrm{m}} \times 1$ | |
| $\mathbf{d}$ | Observation space | Observation variable | $N_{\mathrm{o}} \times 1$ | 855 |

$M$ denotes the ensemble size, $N_{\mathrm{m}}$ is the total number of analysis variables, and $K$ is proportional to the number of observations ($N_{\mathrm{o}}$)


**Table 2 Examples of applying the model-space multiscale localization**

| Case | Distance between two variables | Variable name | Scale 0 km -20 km | Scale 20 km -200 km | Scale >200km | Multiscale covariance |
|---|---|---|---|---|---|---|
| 1 | 8 km | Localization coefficient | 0.5 | 1 | 1 | |
| | | Localized Covariance | $0.5C_1$ | $C_2$ | $C_3$ | $0.5C_1+C_2+C_3$ |
| 2 | 80 km | Localization coefficient | 0.01 | 0.5 | 1 | |
| | | Localized Covariance | $0.01C_1$ | $C_2$ | $C_3$ | $0.01C_1+0.5C_2+C_3$ |
| 3 | 300 km | Localization coefficient | 0.0 | 0.05 | 0.5 | |
| | | Localized Covariance | 0 | $0.05C_2$ | $0.5C_3$ | $0.05C_2+0.5C_3$ |

$C_1$, $C_2$, and $C_3$ represent the covariance of the small scale (0 km -20 km), middle scale (20 km -200 km), and large scale (>200 km), respectively.

**Table 3 DA experimental configurations.**

| Experiment names | DA scheme | Static covariance | Dynamic covariance | Localization space | Multiscale localization |
|---|---|---|---|---|---|
| Ens_noFLTR | Local DA | No | Yes | M | No |
| Static_BE | Local DA | Yes | No | M | No |
| Hybrid_noFLTR | Local DA | Yes | Yes | M | No |
| Ens_2band | Local DA | No | Yes | M | Yes |
| Ens_3band | Local DA | No | Yes | M | Yes |
| Ens_5band | Local DA | No | Yes | M | Yes |
| Hybrid_2band | Local DA | Yes | Yes | M | Yes |
| Hybrid_3band | Local DA | Yes | Yes | M | Yes |
| Hybrid_5band | Local DA | Yes | Yes | M | Yes |
| Ens_noFLTR_OL | Local DA | No | Yes | O | Yes |
| Ens_LETKF | LETKF | No | Yes | O | Yes |
| Ens_noFLTR_DSL | Local DA | No | Yes | M+O | Yes |
| Hybrid_5band_DSL | Local DA | Yes | Yes | M+O | Yes |
| Ens_5band_DSL | Local DA | No | Yes | M+O | Yes |

**Figures**

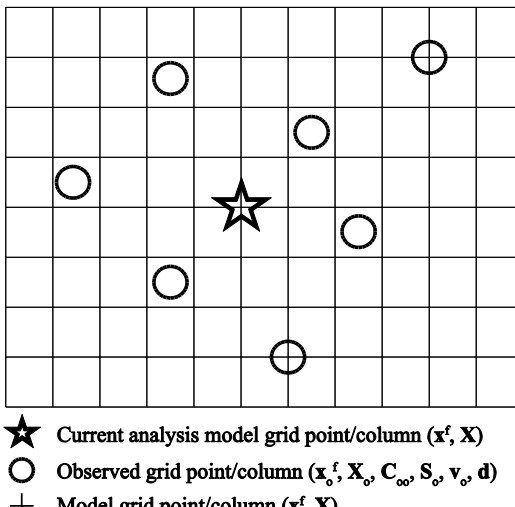

★    Current analysis model grid point/column ($\mathbf{x}^f$, $\mathbf{X}$)

◯    Observed grid point/column ($\mathbf{x}_o^{\,f}$, $\mathbf{X}_o$, $\mathbf{C}_{oo}$, $\mathbf{S}_o$, $\mathbf{v}_o$, $\mathbf{d}$)

＋    Model grid point/column ($\mathbf{x}^f$, $\mathbf{X}$)

**Figure 1 The spatial distribution of different kinds of variables in Local DA**

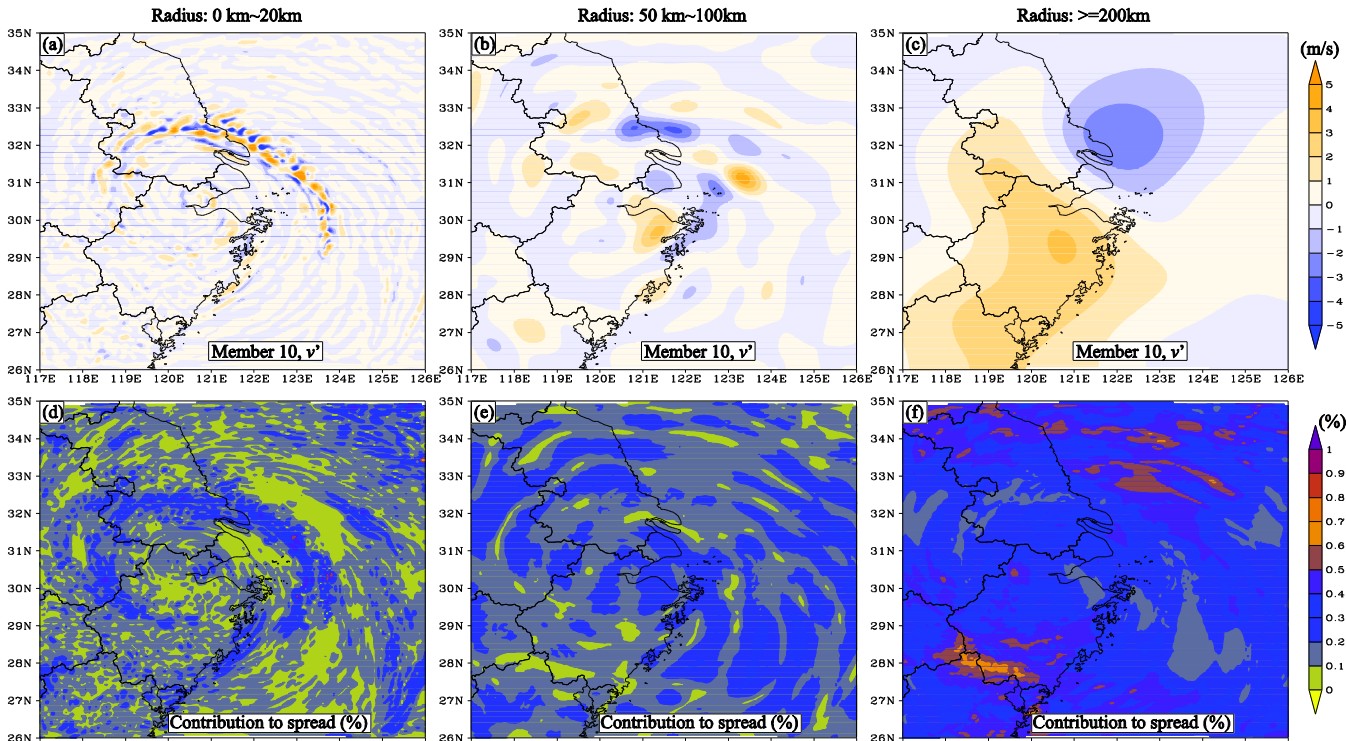

**Figure 2** An example of scale decomposition for scales of (a,d) 0 km - 20 km, (b,e) 50 km – 100 km, and (c,f) greater than 200 km. The upper panels show the decomposed *v* perturbation (m s$^{-1}$), while the lower panels show the contribution of each scale to the ensemble spread in terms of percentage.


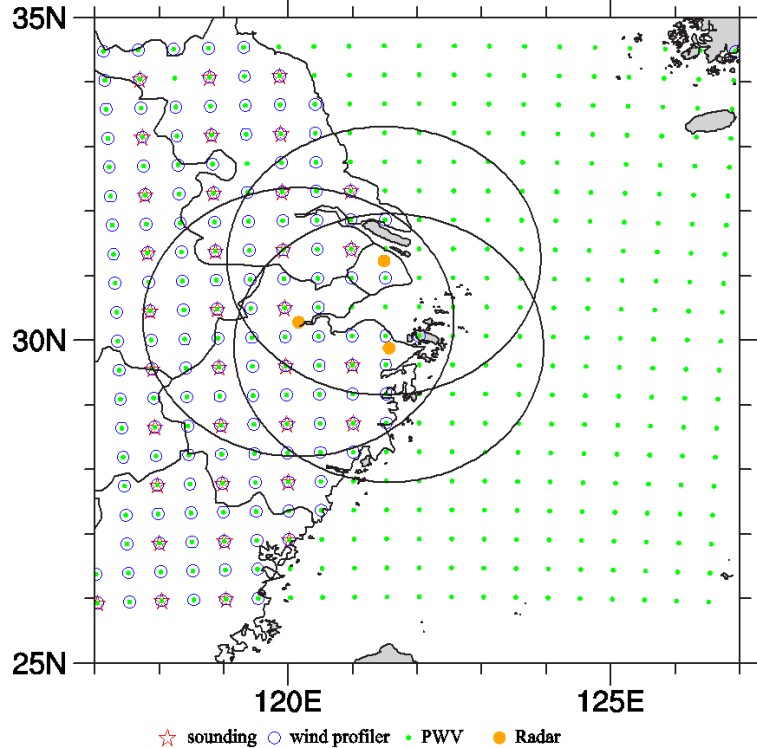

**Figure 3 The distribution of simulated observations, where the black rings denote the maximum observation ranges of radars.**


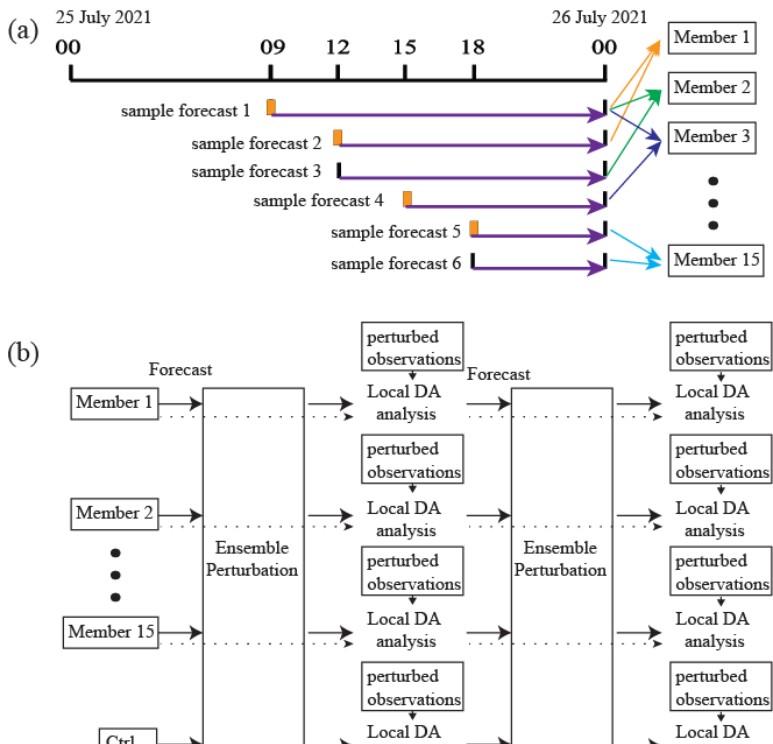

**Figure 4 (a) The flow chart of the time-lagged ensemble generation, where the thick blue arrows represent the sample forecasts used by the 15-member ensemble. The sample forecasts initialized using the GFS forecast data are highlighted with orange tick marks. Sample forecasts used to form a member are denoted by colored thin arrows. (b) The flow chart of cycling DA. Each member assimilates the observations containing a different set of perturbations.**


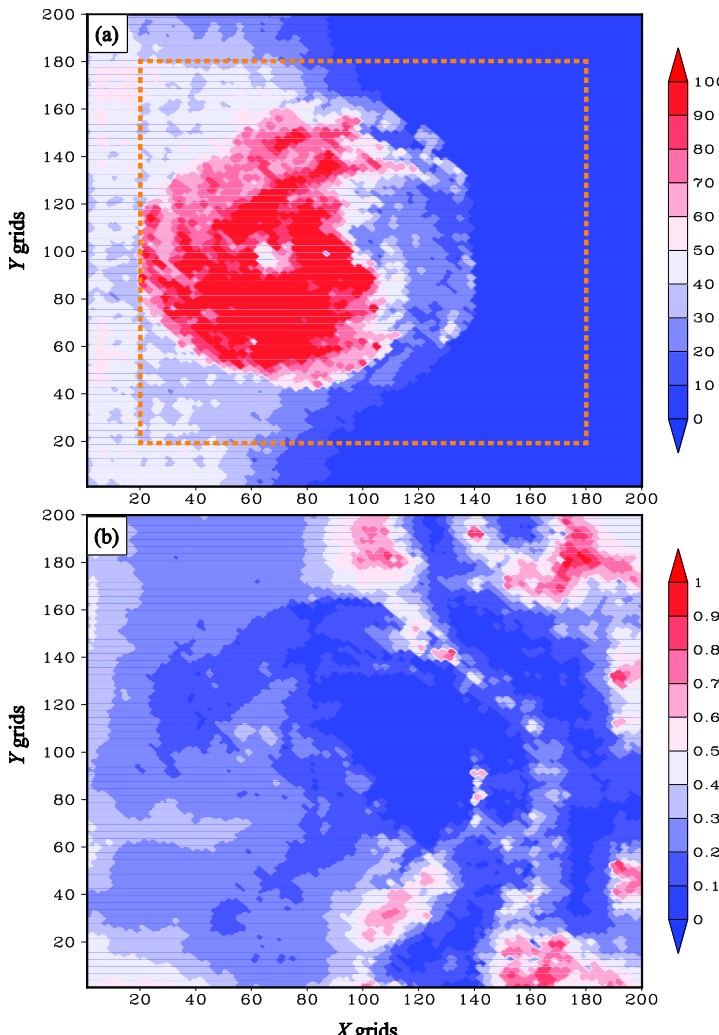

**Figure 5 The spatial distributions of (a) the number of iterations and (b) the ratio of the final value of the cost function to the initial value.**

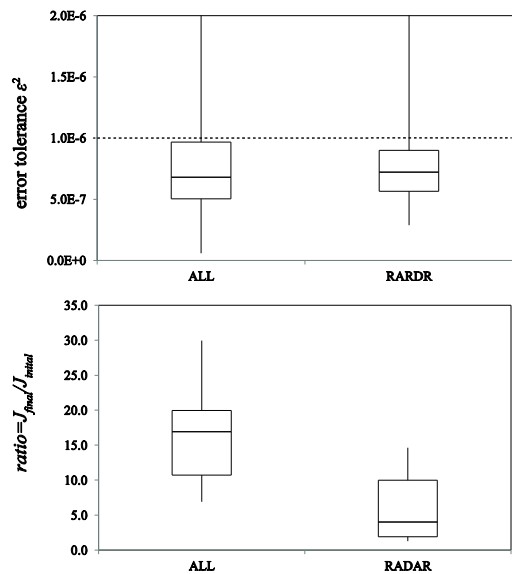

**Figure 6 Boxplots of (a) $\varepsilon^2$ and (b) the ratio of the final $J$ to the initial $J$ in the dashed rectangle area shown in Figure 3, where "ALL" denotes the DA using all observations and "RADAR" corresponds to the DA using radar data only. The upper and lower bounds of the boxes are the 75th and 25th percentiles, respectively. The middle line indicates the median.**



| | U(SND) | V(SND) | T | QV | PS | U(PRO) | V(PRO) | PWV | RV | RF |
|---|---|---|---|---|---|---|---|---|---|---|
| BAK | 3.39 | 3.35 | 1.37 | 3.50E-04 | 70.19 | 3.16 | 3.42 | 2.03 | 4.43 | 25.70 |
| Ens_noFLTR | 1.49 | 1.60 | 1.11 | 1.90E-04 | 35.38 | 1.61 | 1.71 | 1.09 | 2.46 | 17.78 |
| Static_BE | 1.16 | 1.35 | 0.86 | 1.30E-04 | 39.51 | 1.48 | 1.55 | 0.97 | 2.03 | 14.78 |
| Hybrid_noFLTR | 0.97 | 1.15 | 0.84 | 1.10E-04 | 27.70 | 1.25 | 1.29 | 0.87 | 1.95 | 15.68 |
| Ens_2band | 0.72 | 0.79 | 0.87 | 1.00E-04 | 28.74 | 0.94 | 0.90 | 0.73 | 1.83 | 14.94 |
| Ens_3band | 0.63 | 0.66 | 0.80 | 8.00E-05 | 27.66 | 0.81 | 0.83 | 0.73 | 1.67 | 14.68 |
| Ens_5band | 0.64 | 0.67 | 0.81 | 8.00E-05 | 29.96 | 0.79 | 0.79 | 0.68 | 1.65 | 14.39 |
| Hybrid_2band | 0.74 | 0.86 | 0.74 | 9.00E-05 | 26.16 | 1.09 | 1.03 | 1.07 | 1.77 | 14.22 |
| Hybrid_3band | 0.71 | 0.79 | 0.73 | 8.00E-05 | 31.33 | 1.01 | 1.03 | 1.05 | 1.74 | 14.72 |
| Hybrid_5band | 0.80 | 0.86 | 0.79 | 9.00E-05 | 33.05 | 1.04 | 1.05 | 0.95 | 1.77 | 14.71 |
| Ens_noFLTR_OL | 1.58 | 1.73 | 1.10 | 1.90E-04 | 24.76 | 1.71 | 1.77 | 1.00 | 2.67 | 15.30 |
| Ens_LETKF | 1.82 | 2.04 | 1.04 | 2.10E-04 | 40.17 | 2.22 | 2.18 | 1.17 | 2.82 | 16.81 |
| Ens_noFLTR_DSL | 1.40 | 1.50 | 1.09 | 1.80E-04 | 21.99 | 1.47 | 1.53 | 0.95 | 2.41 | 15.45 |
| Hybrid_5band_DSL | 0.66 | 0.72 | 0.72 | 8.00E-05 | 19.42 | 0.88 | 0.90 | 0.53 | 1.74 | 13.25 |
| Ens_5band_DSL | 0.60 | 0.62 | 0.77 | 8.00E-05 | 18.12 | 0.67 | 0.68 | 0.57 | 1.68 | 12.81 |
| | (m/s) | (m/s) | (K) | (kg/kg) | (Pa) | (m/s) | (m/s) | (mm) | (m/s) | (dBZ) |

**Figure 7 The RMSEs in observation space for all single deterministic analyses, where BAK represents the background error, SND denotes the sounding observation, and PRO corresponds to profile observation. The values of 1 and 15 in the legend represent the smallest and the largest error among all experiments, respectively.**


| | U | V | T | QV | W | QC | QR | QI | QS | QG | PS |
|---|---|---|---|---|---|---|---|---|---|---|---|
| BAK | 3.24 | 3.19 | 1.32 | 5.25E-04 | 0.14 | 5.71E-05 | 8.87E-05 | 2.54E-05 | 8.46E-05 | 4.47E-05 | 82.07 |
| Ens_noFLTR | 2.98 | 2.90 | 1.47 | 5.85E-04 | 0.20 | 6.72E-05 | 8.31E-05 | 2.70E-05 | 8.32E-05 | 4.49E-05 | 103.62 |
| Static_BE | 2.39 | 2.57 | 1.23 | 5.72E-04 | 0.19 | 5.71E-05 | 7.87E-05 | 2.54E-05 | 8.19E-05 | 4.44E-05 | 55.77 |
| Hybrid_noFLTR | 2.40 | 2.49 | 1.28 | 5.75E-04 | 0.19 | 5.50E-05 | 7.98E-05 | 2.43E-05 | 8.24E-05 | 4.63E-05 | 80.76 |
| Ens_2band | 2.49 | 2.56 | 1.36 | 5.19E-04 | 0.16 | 5.62E-05 | 7.98E-05 | 2.41E-05 | 8.07E-05 | 4.30E-05 | 65.92 |
| Ens_3band | 2.36 | 2.47 | 1.35 | 5.50E-04 | 0.15 | 5.35E-05 | 7.94E-05 | 2.36E-05 | 7.97E-05 | 4.31E-05 | 58.69 |
| Ens_5band | 2.35 | 2.44 | 1.35 | 5.77E-04 | 0.15 | 5.29E-05 | 7.80E-05 | 2.33E-05 | 7.96E-05 | 4.30E-05 | 58.24 |
| Hybrid_2band | 2.31 | 2.39 | 1.29 | 6.44E-04 | 0.18 | 5.30E-05 | 7.63E-05 | 2.39E-05 | 8.08E-05 | 4.31E-05 | 55.40 |
| Hybrid_3band | 2.25 | 2.35 | 1.27 | 6.33E-04 | 0.18 | 5.26E-05 | 7.60E-05 | 2.32E-05 | 8.08E-05 | 4.35E-05 | 51.48 |
| Hybrid_5band | 2.27 | 2.34 | 1.28 | 6.14E-04 | 0.18 | 5.21E-05 | 7.59E-05 | 2.31E-05 | 8.07E-05 | 4.31E-05 | 48.18 |
| Ens_noFLTR_OL | 2.83 | 2.84 | 1.39 | 5.44E-04 | 0.18 | 6.08E-05 | 8.11E-05 | 2.56E-05 | 8.24E-05 | 4.41E-05 | 53.00 |
| Ens_LETKF | 2.85 | 2.80 | 1.41 | 5.16E-04 | 0.18 | 6.24E-05 | 9.00E-05 | 2.76E-05 | 9.30E-05 | 5.81E-05 | 54.53 |
| Ens_noFLTR_DSL | 2.81 | 2.82 | 1.42 | 5.42E-04 | 0.19 | 6.53E-05 | 8.06E-05 | 2.66E-05 | 8.17E-05 | 4.39E-05 | 67.51 |
| Hybrid_5band_DSL | 2.15 | 2.26 | 1.21 | 4.83E-04 | 0.16 | 5.23E-05 | 7.54E-05 | 2.35E-05 | 8.01E-05 | 4.25E-05 | 49.72 |
| Ens_5band_DSL | 2.26 | 2.40 | 1.29 | 5.08E-04 | 0.15 | 5.31E-05 | 7.66E-05 | 2.35E-05 | 7.94E-05 | 4.26E-05 | 52.78 |
| | (m/s) | (m/s) | (K) | (kg/kg) | (m/s) | (kg/kg) | (kg/kg) | (kg/kg) | (kg/kg) | (kg/kg) | (Pa) |

Legend: 1, 2, 3, 4, 5, 6, 7, 8, 9, 10, 12, 13, 14, 15

**Figure 8 As in Figure 7, but for the RMSEs in model space.**

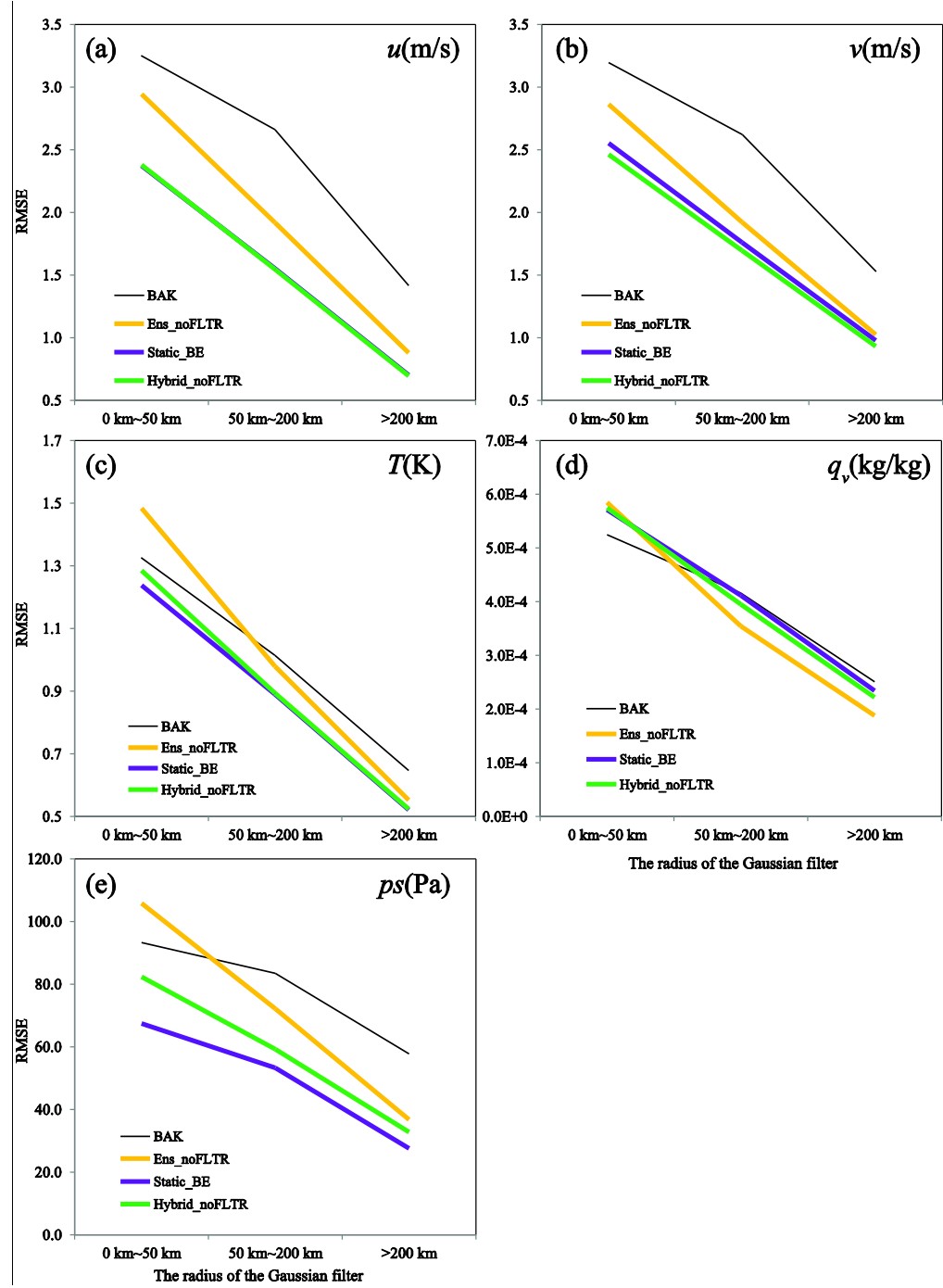


**Figure 9 The analysis error decomposed into scales of 0 km - 50 km, 50 km – 200 km, and greater than 200 km (shown on the x-axis), where BAK represents the initial condition before DA.**

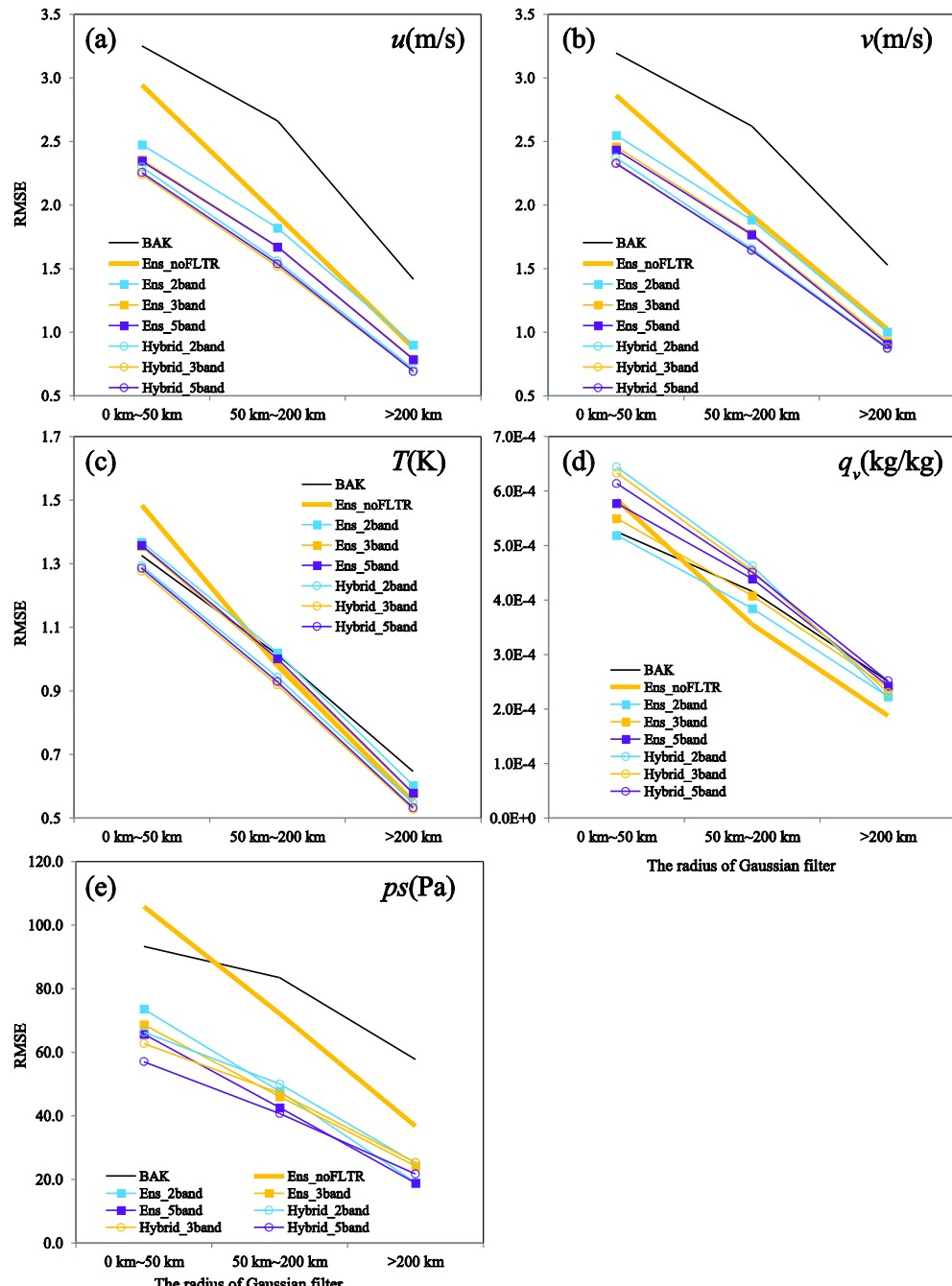


**Figure 10 As in Figure 9, but for Ens_2band, Ens_3band, Ens_5band, Hybrid_2band, Hybrid_3band, and Hybrid_5band, where BAK and Ens_noFLTR are duplicated for comparison.**

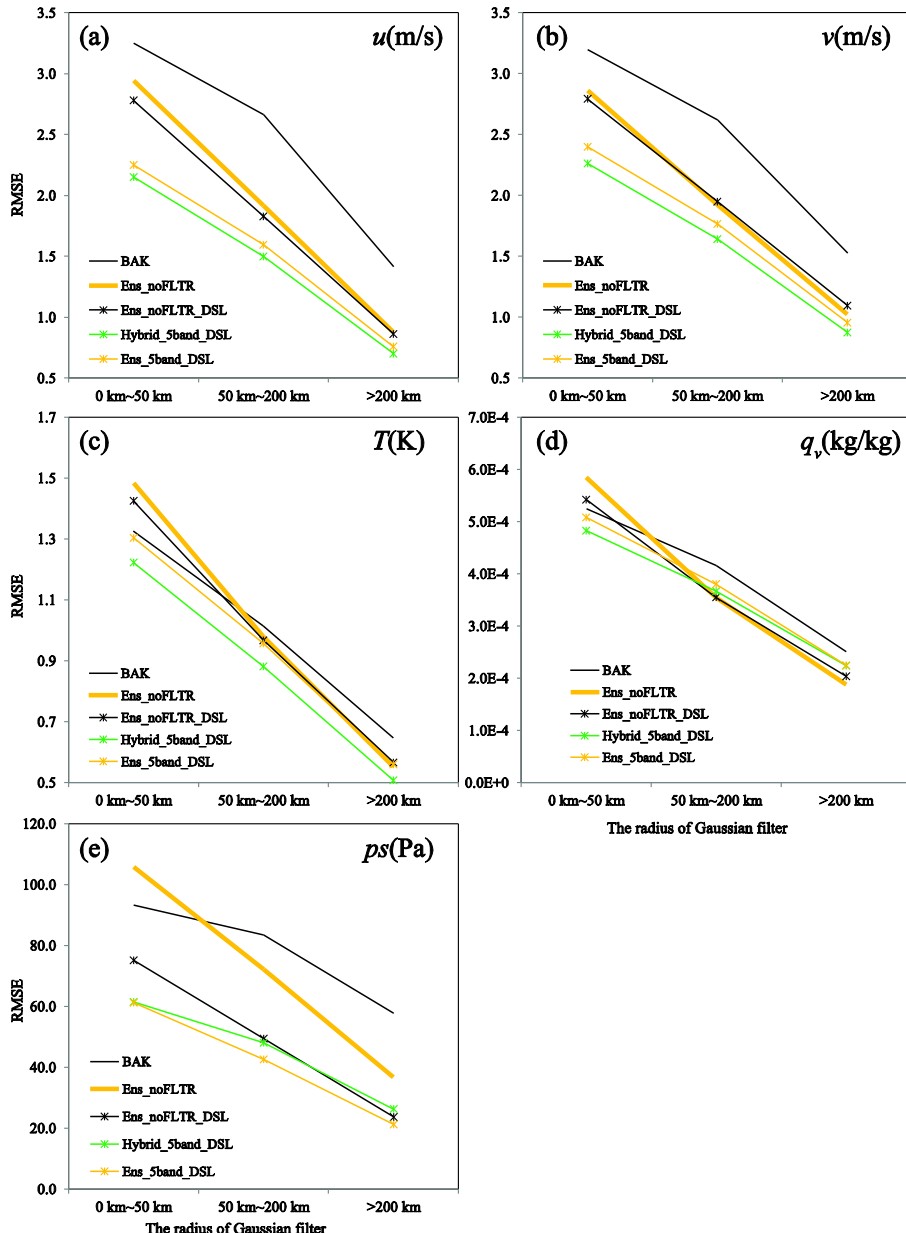

Figure 11 As in Figure 9, but for Ens_noFLTR_DSL, Hybrid_5band_DSL, and Ens_5band_DSL, where BAK and Ens_noFLTR are duplicated for comparison.

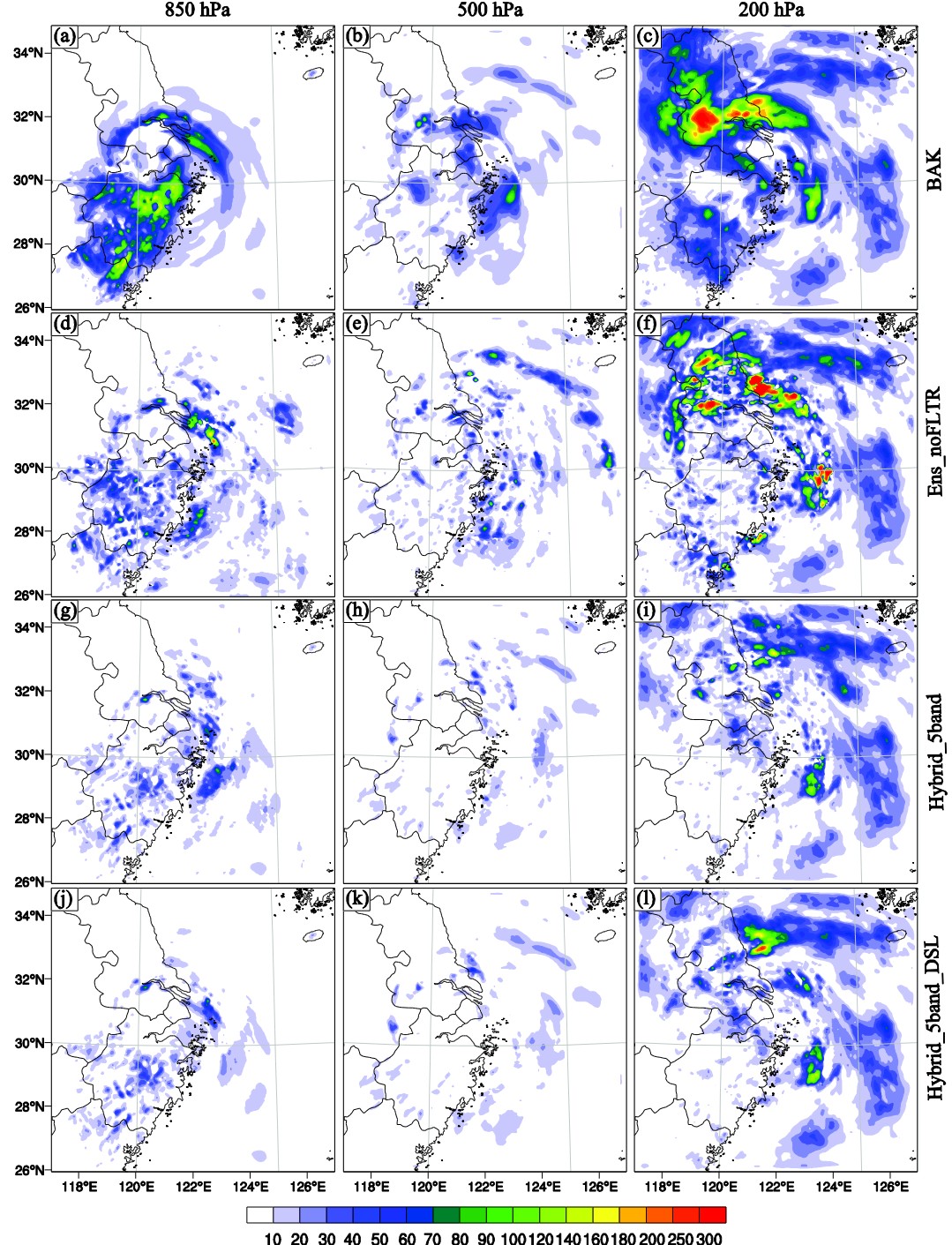


**Figure 12 The DTE at 850 hPa (left column), 500 hPa (middle column), and 200 hPa (right column) for (a-c) BAK, (d-f) Ens_noFLTR, (g-i) Hybrid_5band, and (j-l) Hybrid_5band_DSL.**

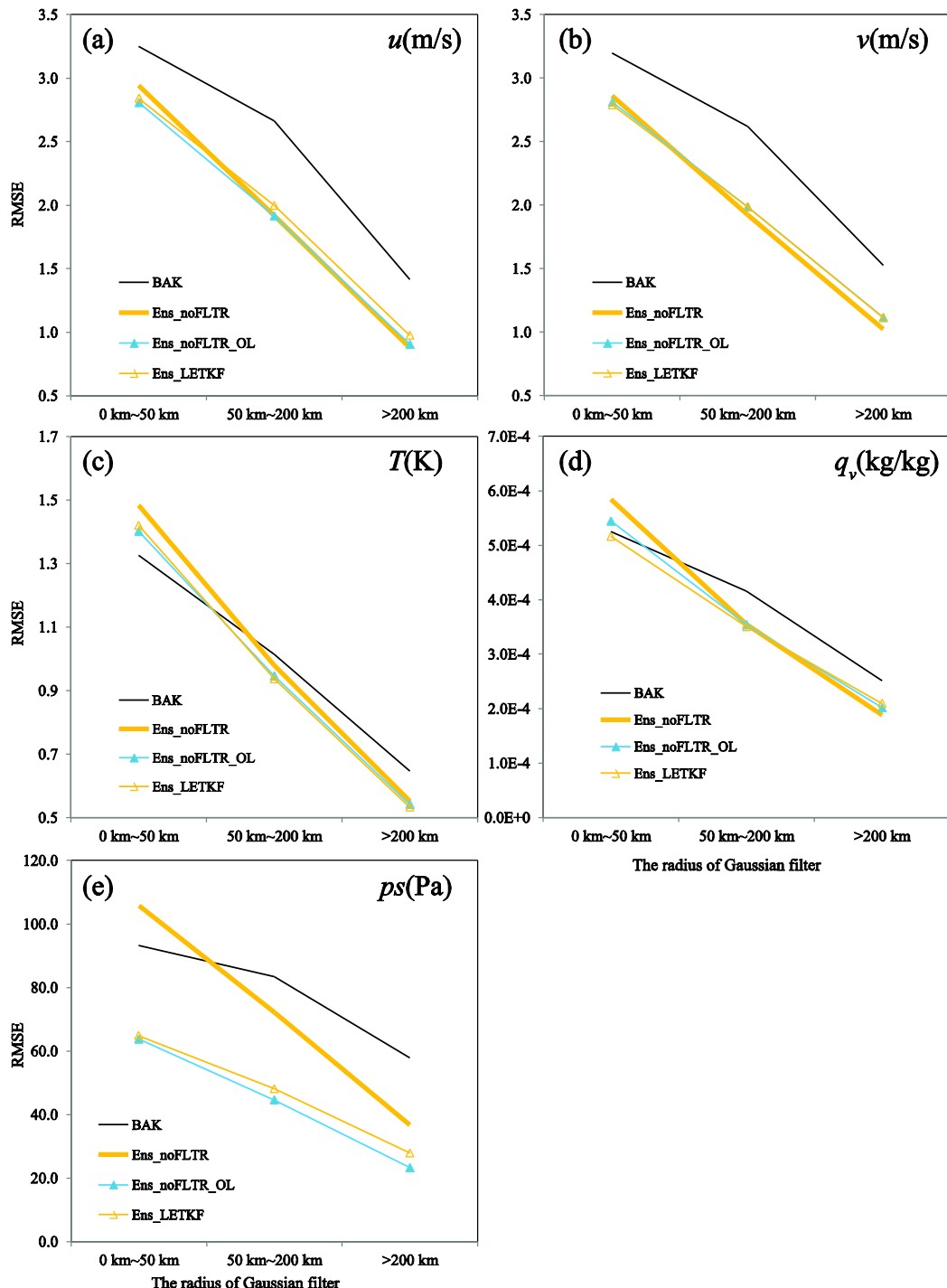

**Figure 13 As in Figure 9, but for Ens_noFLTR_OL and Ens_LETKF, where BAK and Ens_noFLTR are duplicated for comparison.**


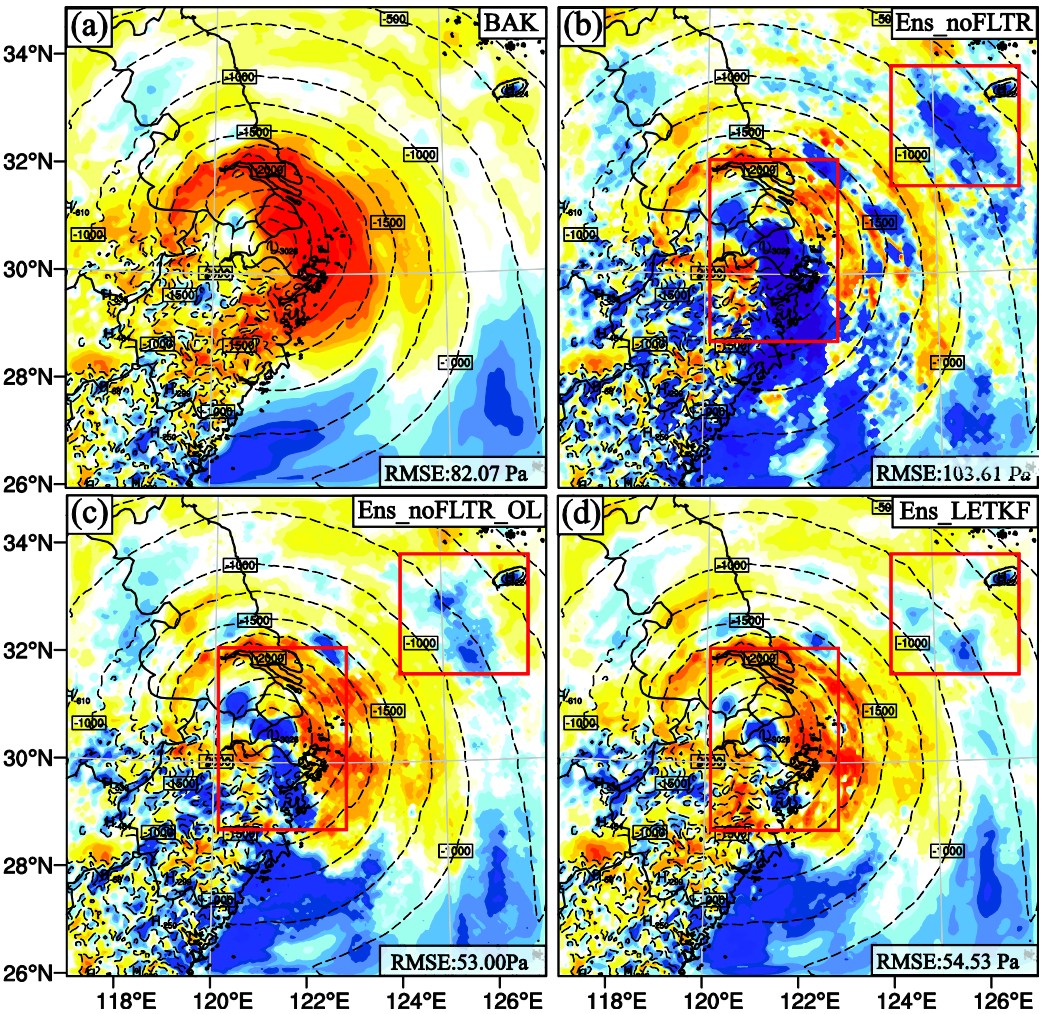

Figure 14 The difference in the dry-air mass in column (*mu*) between the truth (contours) and analysis (shading) for (a) BAK, (b) Ens_noFLTR_OL, and (c) Ens_LETKF, where rectangles highlight the areas where Ens_noFLTR_OL and Ens_LETKF analyses are similar

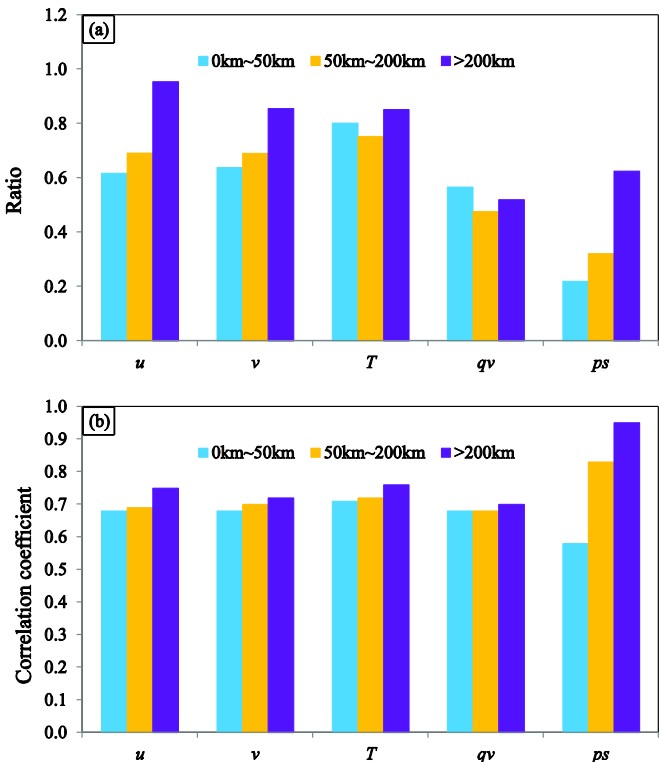

**Figure 15 (a) The ratio of ensemble spread to RMSE at 00 UTC on 26 July 2021 and (b) the spatial correlation coefficient between ensemble spread and RMSE for scales of 0 km - 50 km, 50 km – 200 km, and greater than 200 km.**

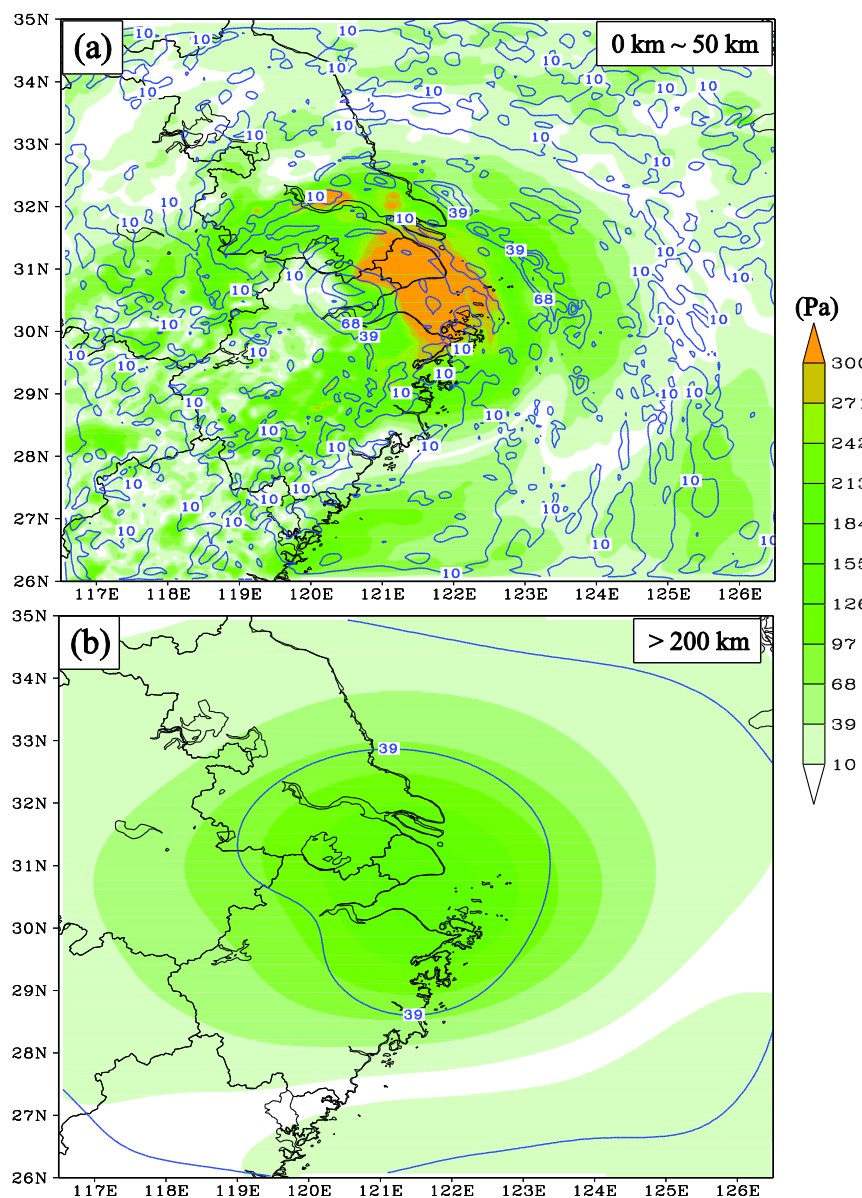


**Figure 16 The RMSE (shaded) and ensemble spread (contours) of *ps* decomposed into scales of (a) 0 km-50 km and (b) greater than 200 km**


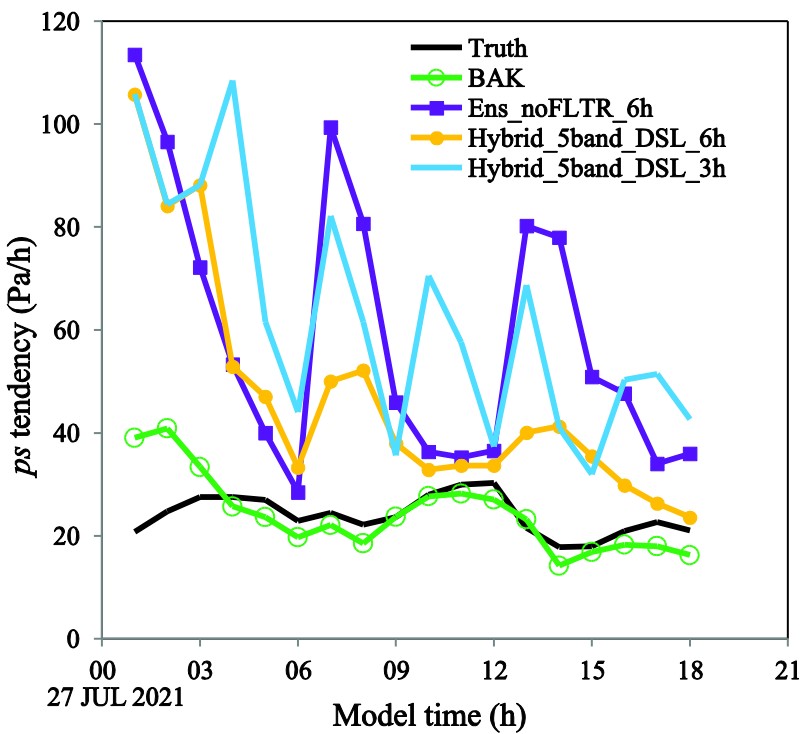

**Figure 17 The tendency of surface pressure (Pa h$^{-1}$) for Truth (black), BAK (green), Ens_noFLTR (blue), Hybrid_5band_DSL_6h** (orange), and Hybrid_5band_DSL_3h (light blue)


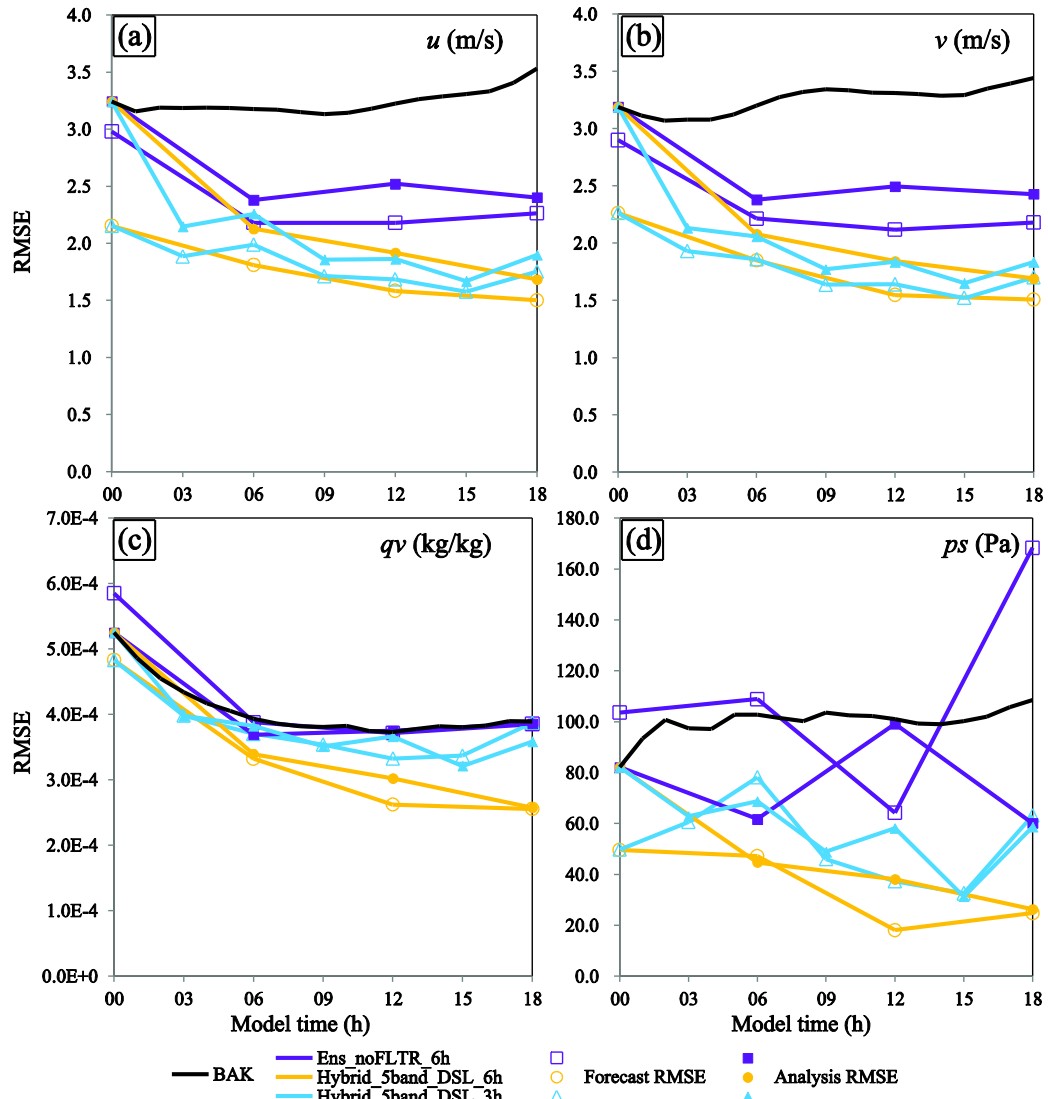

**Figure 18 The evolution of RMSE for BAK (black), Ens_noFLTR (blue), Hybrid_5band_DSL_6h (orange), and Hybrid_5band_DSL_3h (light blue), where the solid markers denote the forecast error while the hollow markers represent the analysis error**

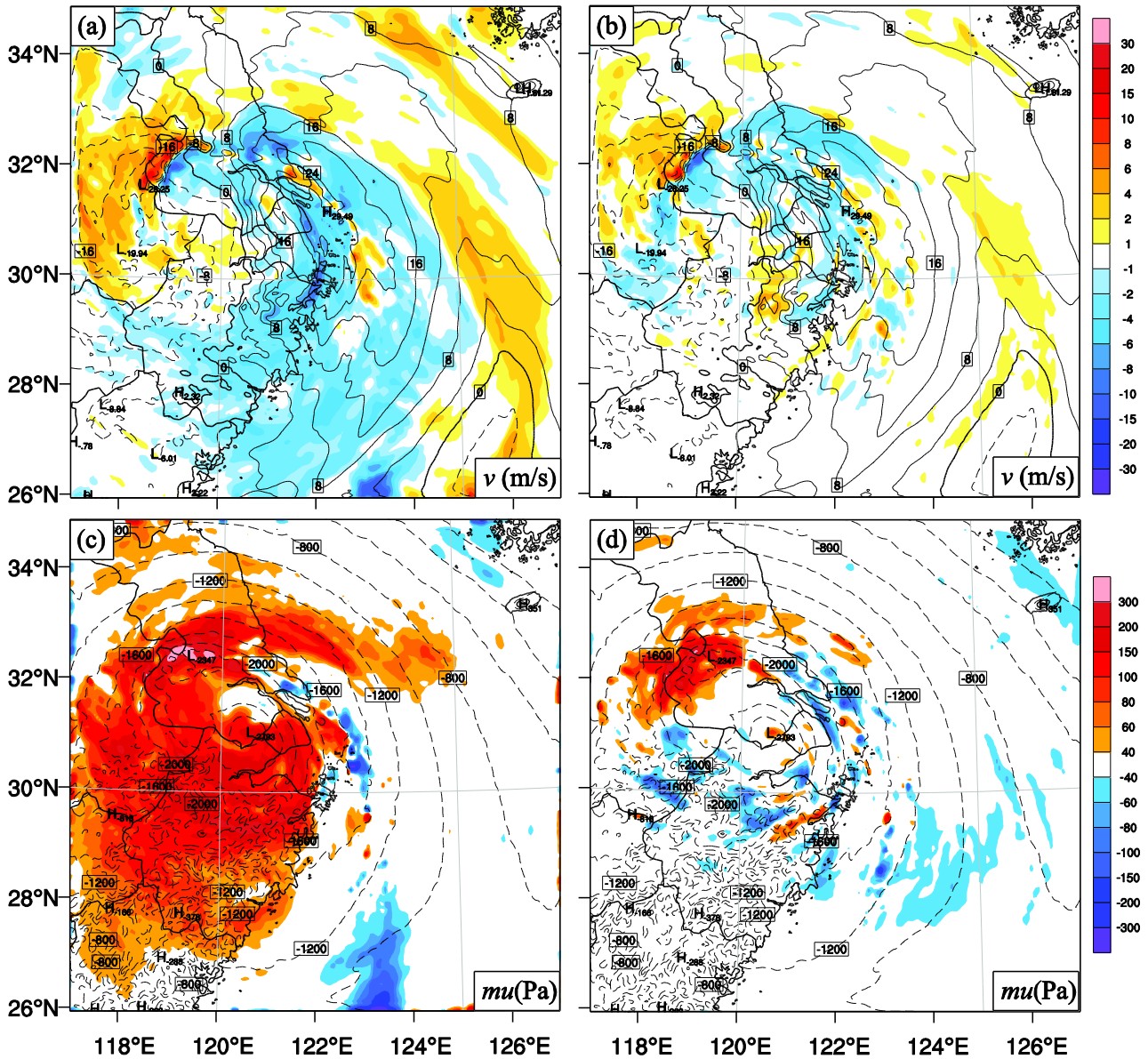

**Figure 19 The difference in (a, b) meridional wind and (c, d) the dry-air mass in column (*mu*) between the truth (contours) and forecast (shading) at 18 UTC 26 July 2021 (the last analysis cycle) for (a, c) Ens_noFLTR_6h and (b, d) Hybrid_5band_DSL_6h.**

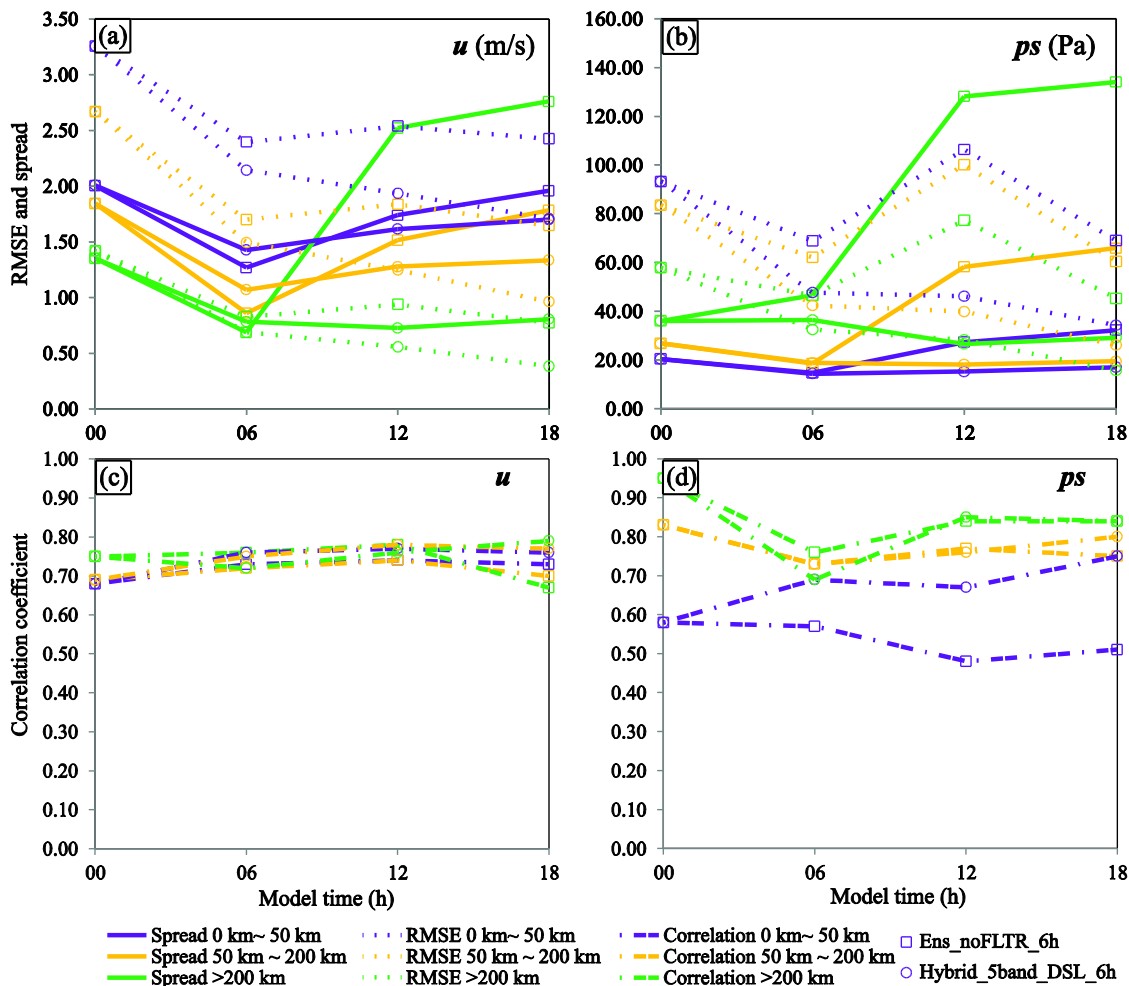

Figure 20 The ensemble spread (solid lines), RMSE (dotted lines), and correlation coefficient between spread and RMSE (dotted dash lines) in three scales for Ens_noFLTR (rectangle markers) and Hybrid_5band_DSL_6h (circle markers), where scales of 0 km-50 km, 50 km -200 km, and > 200 km are denoted by blue, orange, and light blue, respectively.


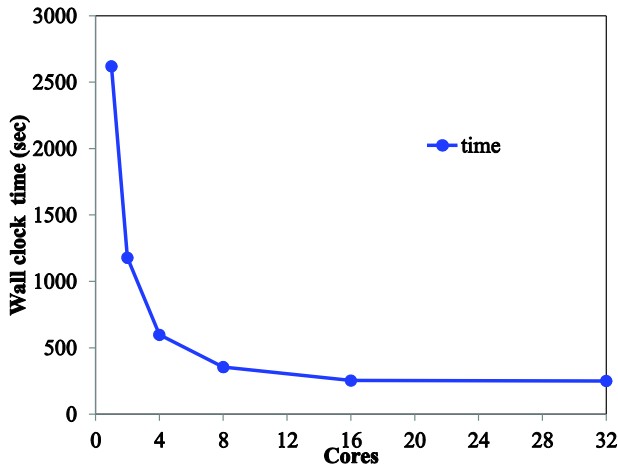

**Figure 21 The wall clock time as a function of the number of cores used in the parallel test**

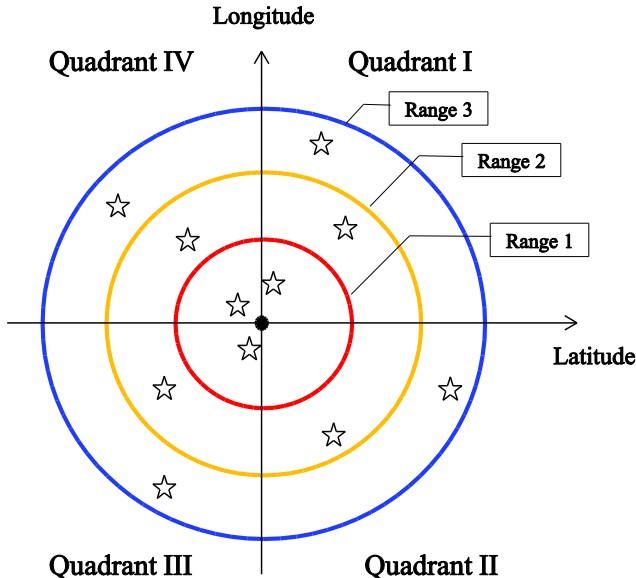

**Figure A1 A schematic of the observation searching approach used in Local DA, where stars represent the selected observations near the grid point (dark solid dot) to be analyzed.**

