# Peer review of "A Local Data Assimilation Method (Local DA v1.0) and its Application in a Simulated Typhoon Case"

_Geoscientific Model Development, 2022_

## Referee Comment (RC2)

**General comments**

The article introduces an interesting hybrid data assimilation scheme called Local DA that utilizes model space, observation space and multi-scale localization. It is believed that great efforts have been made to develop and implement this algorithm. However, the article is not well written and it is difficult to follow the algorithm of Local DA, and its advantage over the other existing methods are not convincing. I have concerns as follows:

1. Overall, the language is not concise and not professional. For instance, in Line 70-71, "observation-associated grids (for scalar observations) and/or columns (for observations that measure an integrated quantity of the atmosphere, such as precipitable water vapor (PWV))", the phrase "observation-associated grids or columns" is strange, can it be simply "observation space"? in Line 99-100 "To avoid issues associated with the quality control of observations when evaluating the performance of Local DA, we adopt observing system simulation experiments (OSSEs)", "when evaluating the performance of Local DA" can be removed. In Line 86-87, "To simplify this study, we leave this issue to be addressed in future work", it is not common to bring up the future work in the introduction. In my opinion, the language of article needs to be greatly improved.

2. It seems that Local DA utilizes model space, observation space and multi-scale localization. Is the model space localization different from the multi-scale localization? As I understand, the multi-scale localization is done in spectrum space (which wavelengths?), does it work independently from the other localizations?

3. Can authors provide the dimensions for each components of Local DA (e.g., $C_{oo}$, $S_o$ and etc.)? In addition, as in Hunt (2007), an analytical solution for Eq(1) can be derived. It is argued that the analytical solution is not feasible if the size of $C_{oo}$ is large? Considering that 100 iterations are required to converge, the iterative method is computationally expensive. Can authors provide theoretical computational expense in case of the analytical solution?

4. It seems that the deterministic run is updated. How is it initialized and how is the Kalman gain of the deterministic run calculated? How are ensemble members updated (equations for this)?

5. I do not fully understand why the main discussion focuses on results of the single-cycle data assimilation. It is well-known that some cycles are usually required to spin up the system. It is argued that the Local DA suffers from imbalance problem. However, most of data assimilation algorithms have more or less the same problem. Is it possible to shift the focus of the discussion to the cycling data assimilation?

6. Evaluation of results: 1) I understand DTE as a metric for the error growth rate in forecasts, is it appropriate to use it to validate the analysis error? 2) In Figure 3, is the analysis or background spread/rmse shown here? Why are the RMSEs of two data assimilation experiments ignored? 3) It would be also interesting to see the verification only for the typhoon region (e.g., accounting for grid points $\geq$ 10 dBZ). 4) The shock is often mentioned in the article, can authors provide a metric for imbalance? Combining it with the RMSEs may show more insights of the results. 5) I suggest that a metric with scale skills (e.g., Fractions Skill Score) can be used especially for the reflectivity forecast verification.

---

## Author Response (AR1)

We thank the reviewers for their careful review of our paper and the helpful comments that improved the manuscript. We give the response in blue and cite the revised text in orange. The original comments are reproduced.

We try to address every issue that reviewers concern with as properly as possible, but we believe there must be some issues that haven't been correctly revised. For any incorrectly processed issues, we appreciate further suggestions to fix them.

We make many modifications throughout the manuscript according to the comments. The experiments are redesigned, and most discussions on results are rewritten. We add the DA experiments using different combinations of scales, discuss errors decomposed into three scales (0 km-50 km, 50 km-200km, and >200km), and show cycling DA experiments that update ensemble perturbations. More details for the Local DA procedure are added, including the examples of multiscale localization and the dimensions of matrices used in Local DA.

Reviewer #1

Review of "A local data assimilation method (Local DA v1.0) and its application in a simulated typhoon case" by Wang and Qiao

This work introduces a local DA method to perform hybrid and simultaneous multiscale DA for each grid or column group individually. Both model- and observation-space localizations are implemented in this method. The OSSE with a simulated typhoon case is used to assess the method. This study examines the impacts of hybrid covariances vs. pure dynamic covariances, fixed localization vs. multiscale localization, and compares the relative effects on the reduction of analysis errors between hybrid covariances and localization. Finally, they explore the discontinuity issue by the CG method and the computational cost issue. Generally, this is an interesting work, and the Local DA method is attractive. However, I don't feel the authors presented convincing results that their new developments are actually useful, undercutting their efforts. I suggest a major revision before accepting for publication.

Major comments:

1. Section 2.2: can the authors provide on details on the realization of the multiscale localization analysis from a formula perspective? For example, what does the control variable Vo look like in the multiscale localization approach? Does it need to be extended to realize the multiscale localization approach compared to the fixed localization approach? How is the Cmo is changed? How is the increment defined in this approach? Without these details, it is hard for the readers to follow the realization of this approach.

Thanks for your comments.

Details on the realization of the multiscale localization analysis are added.

(1) Equations (15) and (16) are added to section 2.2, containing a column of components in the examples of $C_{oo}$ and $C_{mo}$. The related text is as follows.

Line 238-243

*Substituting equations (13) and (14) into equation (6), an example of $\boldsymbol{C}_{oo}$ in equation (6) is written as:*

$$
\mathbf{C}_{oo} = \begin{pmatrix}
\sum_{l=1}^{N_b} r(l, u1, u1) \dfrac{\text{cov}\left[\mathbf{X}_b^l(u1), \mathbf{X}_b^l(u1)\right]}{s(u1)s(u1)} & \cdots & \cdots \\
\sum_{l=1}^{N_b} r(l, u2, u1) \dfrac{\text{cov}\left[\mathbf{X}_b^l(u2), \mathbf{X}_b^l(u1)\right]}{s(u2)s(u1)} & \cdots & \cdots \\
\sum_{l=1}^{N_b} r(l, ps1, u1) \dfrac{\text{cov}\left[\mathbf{X}_b^l(ps1), \mathbf{X}_b^l(u1)\right]}{s(ps1)s(u1)} & \cdots & \cdots
\end{pmatrix}, \tag{15}
$$

*where i and j in equation (13) are replaced by subscripts in equation (6). For brevity, only the first column of $\boldsymbol{C}_{oo}$ is listed. Obviously, applying multiscale localization does not change the size of $\boldsymbol{C}_{oo}$. Correspondingly, an example of $\boldsymbol{C}_{mo}$ in equation (8) can be written as:*

$$
\mathbf{C}_{mo} = \begin{pmatrix}
\sum_{l=1}^{N_b} r(l, v1, u1) \dfrac{\text{cov}\left[\mathbf{X}_b^l(v1), \mathbf{X}_b^l(u1)\right]}{s(v1)s(u1)} & \cdots & \cdots \\
\sum_{l=1}^{N_b} r(l, \theta1, u1) \dfrac{\text{cov}\left[\mathbf{X}_b^l(\theta1), \mathbf{X}_b^l(u1)\right]}{s(\theta1)s(u1)} & \cdots & \cdots \\
\sum_{l=1}^{N_b} r(l, q1, u2) \dfrac{\text{cov}\left[\mathbf{X}_b^l(q1), \mathbf{X}_b^l(u1)\right]}{s(q1)s(u1)} & \cdots & \cdots
\end{pmatrix}. \tag{16}
$$

According to the above equations, the multiscale localization only adjusts the computation of $C_{oo}$ and $C_{mo}$ components and does not change the size of both matrices. Therefore, there is no change to the form of $\mathbf{v}_o$, no matter using the multiscale localization or not. We added the related text as follows.

Line 244-246

*Because the sizes of $\mathbf{C}_{oo}$ and $\mathbf{C}_{mo}$ do not change, there is no modification for $\mathbf{v}_o$, $\mathbf{x}^i$, $\mathbf{x}^f$, and $\mathbf{x}^a$. The only modification to realize multiscale localization in model space is to store the error sample of each scale and compute the corresponding correlation coefficient. Therefore, realizing multiscale analysis within the Local DA framework is easy.*

(2) A table (Table 2) is added to demonstrate how the multiscale localization works. The related text is also added.

*Table 2 Examples of applying the model-space multiscale localization*

| Case | Distance between two variables | Variable name | Scale 0 km -20 km | Scale 20 km -200 km | Scale >200km | **Multiscale covariance** |
|------|------|------|------|------|------|------|
| 1 | 8 km | Localization coefficient | 0.5 | 1 | 1 | |
| | | Localized Covariance | $0.5C_1$ | $C_2$ | $C_3$ | **$0.5C_1+C_2+C_3$** |
| 2 | 80 km | Localization coefficient | 0.01 | 0.5 | 1 | |
| | | Localized Covariance | $0.01C_1$ | $C_2$ | $C_3$ | **$0.01C_1+0.5C_2+C_3$** |
| 3 | 300 km | Localization coefficient | 0.0 | 0.05 | 0.5 | |
| | | Localized Covariance | 0 | $0.05C_2$ | $0.5C_3$ | **$0.05C_2+0.5C_3$** |

*$C_1$, $C_2$, and $C_3$ represent the covariance of the small scale (0 km -20 km), middle scale (20 km -200 km), and large scale (>200 km), respectively.*

*The multiscale localization proposed in this subsection gradually diminishes the contribution of small-scale covariance as the distance between two variables increases while retaining that of large-scale covariance until the distance is very large. Table shows an example of multiscale localization. In this example, there are two arbitrary variables of which the error samples are decomposed into three scales. The values of covariance between the two variables are $C_1$, $C_2$, and $C_3$ at three scales. When the two variables are close (8 km), the localization coefficients of $C_2$ and $C_3$ are 1.0, according to the first formula in equation (14). As the distance increases to 300 km, the localization coefficients of $C_1$ and $C_2$ become nearly zero, and the total covariance is mainly attributable to $C_3$.*

2. L235: is the static correlation always identical to all variables? I do not think it is realistic. For example, convective-scale variables usually have a smaller spatial correlation length than horizontal winds and temperature. The authors need to give some explanations on how they solve this issue in the static correlation.

Thanks for your comments.

The influence radii of the static correlation are independently set for each analysis variable, although the radii for wind component, temperature, and water vapor are identical (200 km) in this study. The radii for *ps* and hydrometeor variables are 1000 km and 20 km, respectively. These values are given empirically. In our early tests, 200 km is sufficient for most variables. For hydrometeor variables, a small radius should be given, as your comment. Assigning a large radius for *ps* prevents the analysis from severe imbalance because a large part of the typhoon on the ocean is not updated if a small influence radius of *ps* is used. The related text is given in section 3.3.2, the DA configurations, as follows.

*The model variables to be analyzed are the three wind components (u, v, w), potential temperature ($\theta$), water vapor mixing ratio ($q_v$), dry-air mass in column (mu), and hydrometeor mixing ratios ($q_c$, $q_r$, $q_i$, $q_s$, and $q_g$). A fixed localization radius of 200 km is used for most variables. For ps and hydrometeor variables ($q_c$, $q_r$, $q_i$, $q_s$, and $q_g$), the fixed influence radii are 1000 km and 20 km, respectively. These values are tuned for the case in which Typhoon In-Fa made landfall in this study and are only used for static correlation and experiments without multiscale localization (e.g., Ens_noFLTR).*

Authors acknowledge that the above configurations are not optimal and plan to improve it in future work.

3. L240: what is the purpose to force the second term on the RHS of Eq. (14) to zero? If it is desired to force cross-variable covariances to zero, could the authors comment that how to realize the update of unobserved variables in this Local DA method?

Thanks for your comments.

Equation (14) is equation (17) in the revised manuscript. The second term on the RHS of this equation is forced to zero only in the case of computing cross-variable correlation. This term is not zero for the univariate correlation because the correlation is modeled by a distance correlation function. For the cross-variable correlation, that term is zero because we don't have functions to model the relationship. We are working on the cross-variable correlation for Local DA. Authors acknowledge that some cross-variable correlations are modeled in previous studies, but implementing these correlations in the form of Local DA is not simple. The current Local DA updates the unobserved model variables via the cross-variable correlation computed using the ensemble covariance.

The related text is as follows.

Line 272-277

*Note that the static part of equation (17) represents merely a distant correlation. It is valid for the univariate correlation rather than the cross-variable scenario. Therefore, the static part of equation (17) is forced to zero if the ith and jth variables are different types of variables. In other words, the cross-variable correlation is contributed only by the ensemble part. The authors acknowledge that the cross-variable correlation is important for DA, but the static cross-variable correlation must be carefully modeled, such as the correlation between wind components and geopotential height, or between the stream function and potential temperature. The modeling work is in progress.*

4. Section 3.1: what is the performance of the simulated typhoon against the observation? It is better for the nature run in OSSE to agree with the real atmosphere within predefined limits, according to Hoffman and Atlas (2016, BAMS).

Thanks for your comments.

We compared the typhoon central-pressure (*ps*) at sea level of the Truth simulation and the observations published by the China Meteorological Administration (CMA). The result shows a consistency between the *ps* tendencies of the observation and the Truth simulation. The following figure shows the *ps* tendency from 00 UTC on 25 July to 18 UTC on 26 July for the observation (OBS) and the Truth simulation. The location of typhoon centers in the observation and the Truth simulation are also close (not shown).

[Figure]

Considering that the revised manuscript is long, the above figure is not shown in the revision. The related text is added in section 3.1 and is as follows.

Line 339-344

*According to Hoffman and Atlas (2016), a criterion for reasonable OSSEs is that the true simulation agrees with the real atmosphere. The typhoon central pressure in the Truth simulation gradually increases from 968 hPa to 980 hPa by 18 UTC on 26 July 2021 (not shown), which is consistent with the real observation obtained from the China Meteorological Administration (CMA), except that the observed pressure increases more rapidly, reaching 985 hPa by 18 UTC on 26 July 2021. The simulated typhoon's central location also agrees with the CMA observation. Therefore, the Truth simulation is eligible for OSSEs.*

5. L365: Please elaborate on how these localization lengths and wavebands are selected. Given the deficient ensemble used in this study, a broad localization may significantly degrade the analysis. For example, I could imagine that LDA_ctrl will obtain worse results as a fixed localization of 200 km is used to assimilate radar observations. You may add a subsection in the results part to discuss the selection of these parameters or the sensitivity to these parameters.

Thanks for your comments. The authors acknowledge that it is necessary to discuss the sensitivity of Local DA to multiscale localization parameters. Therefore, we add new experiments to test multiscale localization parameters. In addition, we rename all experiments to make them clearer.

The previous Table 1 is Table 3 in the revised manuscript.

*Table 3 DA experimental configurations.*

| Experiment names | DA scheme | Static covariance | Dynamic covariance | Localization space | Multiscale localization |
|---|---|---|---|---|---|
| Ens_noFLTR | Local DA | No | Yes | M | No |
| Static_BE | Local DA | Yes | No | M | No |
| Hybrid_noFLTR | Local DA | Yes | Yes | M | No |
| Ens_2band | Local DA | No | Yes | M | Yes |
| Ens_3band | Local DA | No | Yes | M | Yes |
| Ens_5band | Local DA | No | Yes | M | Yes |
| Hybrid_2band | Local DA | Yes | Yes | M | Yes |
| Hybrid_3band | Local DA | Yes | Yes | M | Yes |
| Hybrid_5band | Local DA | Yes | Yes | M | Yes |
| Ens_noFLTR_OL | Local DA | No | Yes | O | Yes |
| Ens_LETKF | LETKF | No | Yes | O | Yes |
| Ens_noFLTR_DSL | Local DA | No | Yes | M+O | Yes |
| Hybrid_5band_DSL | Local DA | Yes | Yes | M+O | Yes |
| Ens_5band_DSL | Local DA | No | Yes | M+O | Yes |

The prefix "Ens" represents using the ensemble covariance only, while "Hybrid" denotes the experiments using the hybrid covariance. Static_BE uses the distance correlation function. "noFLTR" means no bandpass filter is used. "2band", "3band", and "5band" indicate that the ensemble perturbations are decomposed into 2, 3, and 5 scales, respectively. "DSL" means double-space localization. "OL" means doing observation-space localization. In this study, the radius of a Gaussian filter is used to represent the scale.

The scales of 2 bands: 0 km -200 km, and >200 km.

The scales of 3 bands: 0 km -50 km, 50 km -200 km, and >200 km.

The scales of 5 bands: 0 km -20 km, 20 km -50 km, 50 km -100 km, 100 km -200 km, and >200 km.

The relationship between the original experiment name and the revised names

LDA_ctrl   ->  Ens_noFLTR

LDA_HBC_MSL   ->  Hybrid_5band

LDA_HBC   ->  Hybrid_noFLTR

LDA_DS   ->  Hybrid_5band_DSL

LDA_OS   ->  Ens_noFLTR_OL

LETKF_OS   ->  Ens_LETKF

Due to the new experimental design, LDA_DS_noENS and experiments with 36 members are removed. LDA_DS_noENS is replaced by Static_BE. Static_BE is used to compare with Ens_noFLTR and Hybrid_noFLTR. The results of experiments with 36 members are mentioned as "*The authors have tested a larger size ensemble with 36 members and obtained lower analysis errors than the 15-member counterpart. For brevity, the results with the 36-member ensemble are not shown.*" (Line 392-394)

The experimental design in section 3.3.2 is as follows, where the text related to the multiscale localization parameters is underlined.

Line 400-428

*A total of 14 experiments for deterministic analyses at 00 UTC on 26 July 2021 are examined. The first three experiments investigate the influence of using the pure ensemble covariance (Ens_noFLTR), distant correlation covariance (Static_BE), and hybrid covariance (Hybrid_noFLTR) on the Local DA analysis. The model variables to be analyzed are the three wind components (u, v, w), potential temperature ($\theta$), water vapor mixing ratio ($q_v$), dry-air mass in column (mu), and hydrometeor mixing ratios ($q_c$, $q_r$, $q_i$, $q_s$, and $q_g$). A fixed localization radius of 200 km is used for most variables. For ps and hydrometeor variables ($q_c$, $q_r$, $q_i$, $q_s$, and $q_g$), the fixed influence radii are 1000 km and 20 km, respectively. These values are tuned for the case in which Typhoon In-Fa made landfall in this study and are only used for static correlation and*

*experiments without multiscale localization (e.g., Ens_noFLTR). The background error covariance is empirically inflated by 50%. For Hybrid_noFLTR, the weight between the dynamic and static covariances is 0.5.*

*Then, the impact of model-space multiscale localization is evaluated through 6 experiments with/without the hybrid covariance. Ens_2band, Ens_3band, and Ens_5band use the pure ensemble covariance, but the ensemble is decomposed into 2, 3, and 5 scales, respectively. The 2-band experiment uses samples with a scale of 0 km - 200 km and a scale greater than 200 km. In this experiment, the contribution of a scale greater than 200 km is amplified because the localization coefficient is 1.0 until the distance between two grid points is greater than 200 km. For the Ens_3band, the three scales are 0 km - 50 km, 50 km - 200 km, and >200 km. The corresponding values for Ens_5band are 0 km - 20 km, 20 km - 50 km, 50 km - 100 km, 100 km - 200 km, and >200 km, respectively. Through the above three experiments, we can examine the sensitivity of Local DA to the configuration of multiscale analysis. Hybrid_2band, Hybrid_3band, and Hybrid_5band use the same ensemble covariance as Ens_3band, and Ens_5band, respectively; the ensemble covariance and static covariance weight equally in the hybrid covariance.*

*The last five experiments are designed to discuss the impact of the localization space. Ens_noFLTR_OL performs localization in observation space. The horizontal radii are 360 km, 150 km, 120 km, and 15 km for sounding, wind profiler, PWV, and radar data, respectively. Notably, Ens_noFLTR_OL performs vertical localization in model space, identical to Ens_noFLTR. Ens_LETKF uses the LETKF algorithm and the same horizontal localization radii as Ens_noFLTR_OL. The vertical radius for all observations is 5 km, where the PWV observations are supposed to be available at 4000 m for LETKF localization. Ens_noFLTR_DSL performs localization in both the model and observation space. In the model space, a fixed localization radius is used, as in Ens_noFLTR, while the localization parameters of Ens_noFLTR_OL are adopted for observation-space localization. By using 5-band samples, Ens_noFLTR_DSL becomes Ens_5band_DSL. Adding hybrid covariance to Ens_5band _DSL yields Hybrid_5band_DSL. For convenience, all single deterministic analysis experiments are listed in Table , where "M" and "O" denote the model and observation spaces, respectively.*

In section 4, we discuss the results related to the multiscale localization experiments as follows. In general, the multiscale localization reduces the analysis error at a small scale. Decomposing

error samples into three or five scales produces similar results. Combining the multiscale localization and hybrid covariance has a small positive impact on the analysis.

Line 506-517

*4.2.2 Multiscale analysis*

*After decomposing the ensemble samples into two parts (Ens_2band) and independently applying the localization radius for each scale, the small-scale analysis error becomes lower than that of Ens_noFLTR for all examined variables (Figure 10). Compared with Ens_2band, further decomposing the ensemble samples into more scales (Ens_3band and Ens_5band) and using smaller radii for small scales slightly reduces the analysis error for wind components and surface pressure but increases the error for $q_v$. This result confirms the speculation that restricting the impact of small-scale correlation in a small region is beneficial. The difference between Ens_3band and Ens_5band is small, indicating that three or five scales should be sufficient for the model-space multiscale localization in Local DA.*

*Experiments combining multiscale localization with hybrid covariance (Hybrid_2band, Hybrid_3band, and Hybrid_5band) produce lower analysis errors for most variables, compared with Ens_2band, Ens_3band, and Ens_5band. However, the improvement is not substantial. The small difference implies that we need more approaches to make further improvements. Employing double-space localization is one of the approaches, according to the result shown in Figure 8.*

In the above text, the errors are decomposed into three scales, representing errors of the small scale (0 km - 50 km), middle-scale (50 km – 200 km), and large scale (>200 km). This discussion gives more details on the Local DA analysis. Before introducing the scale decomposition, we briefly describe the impact of multiscale localization. The related text is as follows.

Line 472-476

*Model-space multiscale localization (Ens_2band, Ens_3band, and Ens_5band) is conducive to error reduction. Even with 2-scale samples, Ens_2band dramatically reduces the errors of wind-related variables, compared with Ens_noFLTR. Involving more scales further improves the analysis, but the benefit is not as great as the case of comparing Ens_noFLTR with Ens_2band. Combining the hybrid covariance and model-space multiscale localization does not further narrow the gap between the analysis and observation.*

For your comments "For example, I could imagine that LDA_ctrl will obtain worse results as a fixed localization of 200 km is used to assimilate radar observations", we did carefully analysis.

We found that the high analysis error of Ens_noFLTR (was LDA_ctrl) is attributable to the analysis at a small scale. The small-scale information is primarily observed by radar. Therefore, your comments can be partly explained by the analysis at a small scale.

Our results indicate that the small-scale ensemble covariance is irreliable; constraining the impact of the small-scale ensemble covariance reduces the analysis error, no matter the approach. Figures 7, 9, 10, and 11 in the revised manuscript show that the hybrid covariance, static correlation, and multiscale localization mainly reduce the small-scale error for wind components. The difference between the analysis error of Ens_noFLTR at a large scale and that of other experiments is relatively small.

Minor comments:

1. L160: If only observation "u1" is available, Eq. (7) seems to be incorrect. Is this a typo? Should it be "where observations "u1", "u2", and "ps" are available"?

   Thanks for your comments.

   Our original example is somehow confusing and we did some modifications to that example. The model variable 'u1' is replaced by 'v1' so that the model variables to be updated are "v1", "θ1", and "q1".

   Line 161-165

   *To obtain the model state increment $x^i$, it is necessary to form $\mathbf{C}_{mo}$ and the corresponding $S_m$. If the model variables to be updated are the zonal wind (v1), potential temperature (θ1), and water vapor mixing ratio (q1), $\mathbf{C}_{mo}$ is written as*

   $$\mathbf{C}_{mo} = \begin{pmatrix} c_{\theta 1u1} & c_{\theta 1u2} & c_{\theta 1ps1} \\ c_{q1u1} & c_{q1u2} & c_{q1ps1} \\ c_{v1u1} & c_{v1u2} & c_{v1ps1} \end{pmatrix}, \tag{9}$$

   *where subscripts "u1", "u2", and "ps1" are the same as those in Eqs. (6) and (7), while subscripts "v1", "θ1", and "q1" denote the model variables to be updated. $\mathbf{C}_{mo}$ comprises the error correlation coefficients between $X$ and $X_o$.*

2. L180: remove "s" after elements

Fixed

3. Section 2.4: the author may need to provide some results of the bandpass filter. What do the perturbations at each decomposed scale look like?

Thanks for your comments.

We added a new figure (Figure 2) and the related text for a decomposed member.

The related text is as follows. Line 296-303

*"In step 1), there are many ways to realize the bandpass filter. In this study, the difference between two low-pass analyses defines the bandpass field (Maddox, 1980), where the low-pass filter is the Gaussian filter. An example of a bandpass field is shown in Figure 2. For convenience, the radius of the Gaussian filter is used to represent the scale in this study. For the scale of 0 km - 20 km (Figure 2a), the small-scale feature prevails and corresponds to convection in the simulated typhoon. As the radius increases (Figure 2b), larger-scale information is extracted. A large-scale anticyclonic shear is observed when the radius is greater than 200 km (Figure 2c). The results (Figure 2d-f) also show that the contribution of the small-scale ensemble spread is often less than 10% out of the convective area, while in most areas of the forecast domain, the contribution of the large-scale (> 200 km) spread is greater than 20%."*

4. L390-395: The inefficient minimization may be caused by the assimilation of radar reflectivity due to the use of the mixing ratios as state variables. Too small hydrometeor mixing ratio values can lead to overestimated cost function gradient. See Sun et al. (2005), Wang and Wang (2017), and Liu et al. (2021).

Thanks for your comments.

We revised the related text as follows. Line 456-462

*"In the case of setting the maximum number of iterations to 500 for Hybrid_5band, all minimizations converge within 300 iteration steps. The results also show that assimilating only radar data produces a smaller ratio of $J_{final}$ to $J_{initial}$ than the case using all observations (Figure 6b).* **According to previous studies (e.g., Wang and Wang, 2017), the inefficient minimization**

*may be caused by the assimilation of radar reflectivity due to the use of the mixing ratios as state variables. Too small hydrometeor mixing ratio values can lead to an overestimated cost function gradient. Nevertheless, despite the slow convergence, Local DA reduces the cost function by more than 70% within 100 iteration steps in most cases (Figure 6b).*"

5. L425: Please elaborate on the comparison between LDA_OS and LETKF_OS. The current statement is difficult to follow.

Thanks for your comments.

The comparison is discussed in detail. The purpose of the comparison is to show how similar Local DA analysis with observation-space localization is to the LETKF. The analysis errors of both experiments are similar at nearly all scales. Compared with Ens_noFLTR which performs localization in model space, the spatial distribution of the analysis *mu* pattern of Ens_noFLTR_OL is more similar to that of Ens_LETKF (Figure 14).

The related text is as follows. Line 537-548

*"4.2.4 The similarity between Local DA with observation space localization and the LETKF*

*Considering that Local DA can perform observation space localization only as in the LETKF, it is interesting to see if their analyses are similar. Note that Ens_noFLTR_OL and Ens_LETKF merely share the same horizontal localization configuration; they differ in vertical localization. Figure 13 shows that the difference in analysis error between Ens_noFLTR_OL and Ens_LETKF is small for all variables and at all scales. Figure 14 gives an intuitive comparison between the Ens_noFLTR_OL and Ens_LETKF analyses. The overlarge negative-increment in both experiments is constrained in a much smaller area than Ens_noFLTR (marked by red rectangles in Figure 14). They also suppress the small-scale noise in the Ens_noFLTR analysis, corresponding to the lower error in Figure 13e. Overall, in the case of using observation-space localization, Local DA can produce an analysis similar to the LETKF.*

*In addition, the small-scale error of qv yielded by Ens_noFLTR_OL is lower than that of Ens_noFLTR (Figure 13d). The result is similar to the difference between Ens_noFLTR_DSL and Ens_noFLTR, indicating that the improvement of Ens_noFLTR_DSL on qv analysis compared with Ens_noFLTR is mainly attributable to observation-space localization."*

6. L429, L434, L475: As said in the above major comment, I am wondering if a tight localization length can reduce this noisy analysis as the ensemble is deficient.

Yes, a tight localization length can improve the noisy analysis. The noisy analysis is mainly due to the noisy small-scale covariance; shortening the influence length of small-scale covariance, or weakening the impact of small-scale covariance reduces the analysis error.

7. L505: Does the number of wavebands affect memory usage? if so, how is the memory is affected when increasing the number of wavebands? Can you discuss this as well?

Yes, the number of wavebands affects memory usage. Memory usage increases linearly as the number of wavebands increases. For example, using 5 wavebands requires 5 times as large memory as 1 waveband (or no decomposition).
We added the related text in section 4.4 as follows. Line 640-643
*In addition to $C_{oo}$, the model-space multiscale localization requires large memory. Memory consumption is proportional to the number of scales. For example, Ens_3band requires three times as much memory as Ens_noFLTR to store the decomposed perturbations.*

Reviewer #2

General comments

The article introduces an interesting hybrid data assimilation scheme called Local DA that utilizes model space, observation space and multi-scale localization. It is believed that great efforts have been made to develop and implement this algorithm. However, the article is not well written and it is difficult to follow the algorithm of Local DA, and its advantage over the other existing methods are not convincing. I have concerns as follows:

1. Overall, the language is not concise and not professional. For instance, in Line 70-71, "observation-associated grids (for scalar observations) and/or columns (for observations that measure an integrated quantity of the atmosphere, such as precipitable water vapor (PWV))", the phrase "observation-associated grids or columns" is strange, can it be simply "observation space"? in Line 99-100 "To avoid issues associated with the quality control of observations when evaluating the performance of Local DA, we adopt observing system simulation experiments (OSSEs)", "when evaluating the performance of Local DA" can be removed. In Line 86-87, "To simplify this study, we leave this issue to be addressed in future work", it is not common to bring up the future work in the introduction. In my opinion, the language of article needs to be greatly improved.

All "observation-associated grids or columns" are replaced by "observed grids/columns". Using "observation space" is not exact. PWV, for example, is a kind of 2-D observation, but it observes an air column. The cost function of Local DA is written for model variables in the air column observed by PWV, not in the space of the PWV itself. Therefore, we use "observed grids/columns" in the revision.

"when evaluating the performance of Local DA" is removed

"To simplify this study, we leave this issue to be addressed in future work" is removed.

We do a lot of modifications throughout the manuscript and try to improve the expression. The revised manuscript has been polished by a native speaker.

2. It seems that Local DA utilizes model space, observation space and multiscale localization. Is the model space localization different from the multi-scale localization? As I understand, the multi-scale localization is done in spectrum space (which wavelengths?), does it work independently from the other localizations?

Thanks for your comments.

Our original description may be confusing. Multiscale localization employs different localization radii and works in either model or observation space. We decompose the ensemble samples into several scales for the model-space multiscale localization and independently assign the influence radii for each scale. In observation space, multiscale localization independently assigns the influence radius for each kind of observation. For example, we use a large radius for the sounding observations but a small radius for radar observations. Because the model space and observation localizations influence different matrices ($\mathbf{C}_{oo}$ and $\mathbf{R}$, respectively), it is possible for Local DA to perform the double-space localization which has not yet been examined.

We have revised the related text as follows.

In section 1, introduction, Line 73-86

*Another feature of Local DA is the ability to perform multiscale analysis in model space, observation space, or both spaces (double-space localization). In the model space, Local DA adopts a scale-aware localization approach for multiscale analysis that applies a bandpass filter to decompose samples and individually performs localization at each scale; no cross-scale covariance is considered in current Local DA. A similar idea (i.e., lacking cross-scale covariance) is the scale-dependent localization technique proposed by Buehner (2012). Although cross-scale covariance is likely to improve multiscale analysis, the relative performance depends on ensemble size (Caron et al., 2019).*
*Local DA can perform observation-space localization similar to the LETKF, which magnifies the observation error as the distance between the observation and model variables increases. For the multiscale analysis in the observation space, the localization radius increases as the scale of observation increases. Compared with radar data, the scale of sounding data is larger so that a larger radius is assigned.*

*Because model space localization and observation space localization are conducted for covariances in different spaces, it is possible to perform both localizations synchronously. Although double-space localization may result in a double penalty, it would be interesting to note the localization performance. Note that the LETKF of Wang et al. (2021) can also realize double-space localization, but this application has not yet been investigated.*

In section 2.2, Equations (15) and (16) are added, containing a column of components in the examples of $C_{oo}$ and $C_{mo}$. A table (Table 2) is added to demonstrate how the multiscale localization works.

[revised manuscript text omitted]

In section 2.4, we added a new figure (Figure 2) and the related text for a decomposed member.

Line 296-303

*In step 1), there are many ways to realize the bandpass filter. In this study, the difference between two low-pass analyses defines the bandpass field (Maddox, 1980), where the low-pass filter is the Gaussian filter. An example of a bandpass field is shown in Figure 2. For convenience, the radius of the Gaussian filter is used to represent the scale in this study. For the scale of 0 km - 20 km (Figure 2a), the small-scale feature prevails and corresponds to convection in the simulated typhoon. As the radius increases (Figure 2b), larger-scale information is extracted. A large-scale anticyclonic shear is observed when the radius is greater than 200 km (Figure 2c). The results (Figure 2d-f) also show that the contribution of the small-scale ensemble spread is often less than 10% out of the convective area, while in most areas of the forecast domain, the contribution of the large-scale (> 200 km) spread is greater than 20%.*

3. Can authors provide the dimensions for each components of Local DA (e.g., Coo, So and etc.)? In addition, as in Hunt (2007), an analytical solution for Eq(1) can be derived. It is argued that the analytical solution is not feasible if the size of Coo is large? Considering that 100 iterations are required to converge, the iterative method is computationally expensive. Can authors provide theoretical computational expense in case of the analytical solution?

Thanks for your comments.

According to your comments, we do a lot of modifications. First, a table is added (new Table 1). All variables involved in Local DA are listed in that table.

*Table 1 The dimensions of variables in Local DA*

| | *Variable space* | *Variable type* | *dimension* |
|---|---|---|---|
| $x^f$ | *Model space* | *Model variable* | $N_m \times 1$ |
| $X$ | *Model space* | *Model variable* | $N_m \times M$ |

| | | | |
|---|---|---|---|
| $x_o = H_i x^f$ | *Observed grids/columns* | *Model variable* | $K \times 1$ |
| $X_o = H_i X$ | *Observed grids/columns* | *Model variable* | $K \times M$ |
| $C_{oo}$ | *Observed grids/columns* | *Model variable* | $K \times K$ |
| $v_o$ | *Observed grids/columns* | *Model variable* | $K \times 1$ |
| $S_o$ | *Observed grids/columns* | *Model variable* | $K \times 1$ |
| $C_{mo}$ | *Cross space* | *Model variable* | $N_m \times K$ |
| $S_m$ | *Model grid space* | *Model variable* | $N_m \times 1$ |
| $d$ | *Observation space* | *Observation variable* | $N_o \times 1$ |

*M denotes the ensemble size, $N_m$ is the total number of analysis variables, and K is proportional to the number of observations ($N_o$)*

An example for $\mathbf{C}_{oo}$ is given. The related text is as follows.

Line 144-151

*$C_{oo}$ is a $K \times K$ matrix, where K is the number of model variables associated with the observations to be assimilated. K is computed according to*

$$K = \sum_{i=1}^{N_t} [N_o(i) N_{op}(i)], \tag{5}$$

*where $N_t$ is the number of observation types, such as the zonal wind from soundings and the radial velocity from radars, $N_o(i)$ is the number of observations of the ith type, and $N_{op}(i)$ is the number of model variables used by the observation operator of the ith type. **For instance, if radar reflectivity is the only available observation type and there are 100 observations, K is equal to 300 ($100 \times 3$) in the case of using the observation operator of Gao and Stensrud (2012) because the operator requires three hydrometeors ($q_r$, $q_s$, and $q_g$).***

For $\mathbf{S}_o$, the related text is as follows. Line 157-158

*$S_o$ is a $K \times K$ matrix, but a $K \times 1$ array is sufficient to store $S_o$.*

For $\mathbf{C}_{mo}$ and $\mathbf{S}_m$, the related text is as follows. Line 165-1690

*$C_{mo}$ comprises the error correlation coefficients between $X$ and $X_o$. The size of $C_{mo}$ is $N_m K$ which depends on the number ($N_m$) of model variables to be updated. However, there is no need to store full $C_{mo}$ in practice because one row of $C_{mo}$ is needed to update the corresponding model variable. $S_m$ is the STD matrix of model variables, containing $s_{v1}$, $s_{\theta 1}$, and $s_{q1}$ in this example. For convenience, a summary of the dimensions of variables involved in Local DA is listed in Table 1.*

The computational cost for each local analysis depends on the number of observations and the number of model variables used to compute the observation priors. We try to show more details on computational cost, including the cost in CG.

The related text in section 2.4 is as follows. Line 304-307

*Steps 5) to 9) contribute the most to the computational cost of Local DA. Computing $C_{oo}$ requires $MK^2$ operations, which is not less than $N_o^2$, where M represents the size of the ensemble, and $N_o$ denotes the number of observations to be assimilated. Step 7) requires $2K^2$ operations. To calculate step 8), $N_o K^2$ operations are needed. For each iteration step of CG method, the number of operations is slightly larger than $2N_o K$. $N_i$ iteration steps require $2N_i N_o K$ operations.*

For your comment "theoretical computational expense in case of the analytical solution", it is difficult to show the theoretical computational cost of Local DA. But we give a comparison between Local DA and the LETKF in terms of computational cost.

In general, both Local DA and the LETKF need to solve the gradient in the form of $(\mathbf{I}+\mathbf{Y}^T\mathbf{Y})\mathbf{v}=\mathbf{Y}^T\mathbf{d}$. Therefore, the computational cost depends on the number of columns of $\mathbf{Y}$. For LETKF, the number equals the ensemble size. For Local DA, the number is proportional to the number of observations. In the case of assimilating hundreds of or thousands of observations in a local analysis, Local DA is more expensive. For such a large matrix, the CG method is more suitable. Our early tests show that Eigenvalue decomposition or SVD takes many times as long as the CG method for a large matrix.

We state this situation in the text as follows. Line 308-312

*As mentioned above, Step 9) can also be solved through eigenvalue decomposition as the LETKF does. However, $Y$ in Local DA has more columns than the LETKF. In the LETKF, $Y$ has M columns, while the corresponding value is K in Local DA. Therefore, Local DA has to deal with a K by K matrix, while the LETKF only needs to solve an M by M matrix. M is often smaller than $10^2$; thus, $I+Y^TY$ can be handled efficiently by eigenvalue decomposition. In contrast, K could be $10^3$ or higher; thus, the CG method is more suitable.*

The total computational cost of Local DA for the entire domain is also expensive, but we can dramatically reduce the cost by introducing an N-column analysis.

Line 313-325

*Despite the large amount mentioned above, we do not have to do that many operations in practice. For example, step 8) requires just $N_o^2$ operations if only scalar observations are available. Notably, for a 3-D domain containing $N_g$ grid points and $N_v$ variables, the total number of operations will be as $N_gN_v$ times that of one local analysis. However, it is possible to reduce the cost.*

*Considering that $S_m$, $C_{mo}$, and $x_m$ can be applied to all variables influenced by $\hat{y}^o$, it is not necessary to compute $C_{oo}$ for each model variable. Moreover, $S_m$, $C_{mo}$, and $x_m$ may contain variables in more than one vertical column (N-column analysis). The total number of operations in an N-column analysis is reduced to $N_g/(NN_z)$ times as one local analysis, where $N_z$ is the number of levels in one column. Due to using the same $C_{oo}$ for neighboring columns, the N-column analysis is slightly rasterized (not shown), leading to slightly higher errors than the 1-column analysis. However, the extent of this degeneration is acceptable as long as N is not too large (<9). The wall clock time of the N-column analysis is close to 1/N of the 1-column analysis. All Local DA results are generated using a 5-column analysis in this study. A similar N-column analysis approach is the weighted interpolation technique in the LETKF (Yang et al., 2009), which performs LETKF analysis every 3 grid points in both the zonal and meridional directions.*

4. It seems that the deterministic run is updated. How is it initialized and how is the Kalman gain of the deterministic run calculated? How are ensemble members updated (equations for this)?

Thanks for your comments.

In the case of using the CG method, Local DA solves the equation $(\mathbf{I}+\mathbf{Y}^T\mathbf{Y})\mathbf{v}=\mathbf{Y}^T\mathbf{d}$, where $\mathbf{Y}=\mathbf{R}^{-0.5}\mathbf{H}_o\mathbf{C}_{oo}$, rather than explicitly computes the Kalman gain. In the current version of Local DA, the ensemble members are updated by running Local DA $M$ times, where $M$ is the ensemble size. This procedure is similar to Li et al. (2012).

We add the related text in section 2.1 as follows. Line 121-127

*To update ensemble perturbations, the current version of Local DA adopts the stochastic method (Houtekamer and Mitchell, 1998) that treats observations as random variables. This method adds random perturbations with zero mean to d in Eq. (1). For an M-member ensemble, equations (1) and (2) are conducted M times to update members with perturbed observations, similar to the procedure of Li et al. (2012). These analyses share the same background error covariance but use different observations. The stochastic approach was reported to be less accurate than the deterministic approach (e.g., Whitaker and Hamill, 2002) because it introduces additional sampling error. At this stage, Local DA mainly concerns the deterministic analysis; further improvement of the analysis ensemble is left in future work.*

We also add a figure (new figure 4b). Line 394-398

*For the cycling analysis, the first analysis uses the time-lagged 15-member ensemble. In the remaining cycles, the ensemble forecast initialized from the previous analysis ensemble provides the ensemble perturbations. The analysis ensemble is created by performing Local DA 15 times with perturbed observations. The perturbations are added to Ctrl so that the ensemble center on Ctrl. The Ctrl in the first cycle is obtained using GFS analysis at 00 UTC on 26 July 2021. Figure 4b shows the flowchart of the cycling DA.*

For the first cycle or the single deterministic analysis experiments, the time-lagged ensemble is employed. New figure 4a gives more details on the time-lagged ensemble than the original figure 2.

5. I do not fully understand why the main discussion focuses on results of the single-cycle data assimilation. It is well-known that some cycles are usually required to spin up the system. It is argued that the Local DA suffers from imbalance problem. However, most of data assimilation algorithms have more or less the same problem. Is it possible to shift the focus of the discussion to the cycling data assimilation?

Thanks for your comments.

Because the manuscript is the first work on Local DA, a detailed discussion on the single deterministic analyses is necessary. We also agree with you that cycling DA is important. According to the comments from you and the other reviewer, we make many modifications to the experimental design and the result discussion. The discussion on the forecast after single deterministic DA is removed to shorten the manuscript, while the cycling DA experiments are discussed in detail.

In the original manuscript, we did not update the ensemble throughout the cycling to see how Local DA works with a poor ensemble. In this revision, we update the ensemble members and examine the DA configurations of Ens_noFLTR and Hybrid_5band_DSL in the cycling DA.

The related text is as follows

In section 3.3.1

Line 394-398

[revised manuscript text omitted]

New figures (Figures 17-20) are added for the related text.

6. Evaluation of results: 1) I understand DTE as a metric for the error growth rate in forecasts, is it appropriate to use it to validate the analysis error?

Thanks for your comments.

DTE has been applied to validate the DA analysis and forecast. Wang et al., (2012) used the square root of mean DTE to evaluate the DA error. The purpose of using DTE is to simplify the presentation because there are 11 analysis variables. They also created a hydro DTE to evaluate the hydrometeor variables.

We revised the related text is as follows. Line 528-530

*To qualitatively assess the analysis error, we compute the difference in total energy (DTE, Meng and Zhang, 2007). Wang et al. (2012) used the square root of the mean DTE to evaluate the error of DA to simplify the presentation. The DTE is computed in the form of the difference between the analysis and truth.*

2) In Figure 3, is the analysis or background spread/rmse shown here? Why are the RMSEs of two data assimilation experiments ignored?

Thanks for your comments.

The original Figure 3 showed the background RMSE and the corresponding ensemble spread used in the single deterministic analysis; all single analysis experiments use the same ensemble. This figure has been removed. New figures for comparing the RMSE and ensemble spread are Figures 15, 16, and 20, where Figures 15 and 16 are plotted for the background RMSE and the corresponding ensemble spread, while figure 20 is plotted for the cycling DA. The new figures provide detailed information on the RMSE and ensemble spread. Not only the amplitude but also the spatial distributions in terms of spatial correlation are discussed.

The related text for Figures 15 and 16 is as follows. Line 549-566

**4.2.5 Error and ensemble spread**

*For a well-sampled ensemble, a criterion is that the spatial distribution of the ensemble spread is similar to that of RMSE. In addition, the amplitudes of the ensemble spread must be close to the RMSE. The relationship is shown in Figure 15 for the time-lagged ensemble at 00 UTC on 26 July 2021. For u, v, and ps, the ratio of ensemble spread to RMSE ascends as the error scale increases, indicating that the quality of the time-lagged ensemble is rational at a large scale. This relationship is also valid for the spatial distribution (Figure 15b), but the correlation coefficient does not vary from small scale to large scale too much for most variables, except for ps. The correlation coefficient for ps is nearly 1.0 at a large scale, while it is approximately 0.6 at a small scale. This large difference explains why the hybrid covariance and multiscale localization can substantially reduce the error at a small scale for ps. For qv, the small-scale spread is greater than the large-scale spread; the correlation coefficients at all scales are close. This result implies that suppressing the small-scale error covariance does not necessarily improve the analysis quality of qv. Therefore, it is not irrational for Ens_5band and Hybrid_5band to produce a higher analysis error for qv than Ens_2band.*

*An example related to the ensemble spread and RMSE of ps is shown in Figure 16. The RMSE is smooth at a small scale, and there is a maximum near the typhoon center. Although the ensemble spread also has a maximum near the typhoon center, there is a large bias concerning the location. Moreover, the ensemble spread is much noisier than the RMSE, which is a cause of the noisy analysis shown in Figure 14b. In contrast, the large-scale ensemble spread matches the error well, which is conducive to error reduction. Therefore, even with a large localization radius, the surface pressure analysis of Ens_noFLTR at a large scale is not much worse than that of other experiments..*

The related text for figure 20 is as follows. Line 605-617

**4.3.3 The evolution of the relationship between ensemble spread and RMSE**

*For Hybrid_5band_DSL_6h, the initial ensemble spread is smaller than the RMSE at all scales (Figure 20a) for both u and ps. As the number of cycles increases, the ratio of ensemble spread to RMSE increases. By 18 UTC, the ensemble spread is comparable to or greater than the corresponding RMSE at all scales for u. The underestimation of RMSE by the ensemble spread is alleviated for ps (Figure 20b). For the spatial distribution, the relationship between ensemble spread and RMSE does not vary much for u at all scales (Figure 20c). In contrast, the relationship becomes better for ps at a small scale (Figure 20d). Overall, the ensemble is improved in Hybrid_5band_DSL_6h.*

*For Ens_noFLTR_6h, the ensemble spread of u and ps at the small-scale remains smaller than the corresponding RMSE during the cycling DA. In contrast, he ensemble spread at the large scale dramatically increases after the second cycle. The amplitude of the large-scale ensemble spread is even higher than that of the small scale spread, leading to a severe overestimation of the large-scale error. Meanwhile, the correlation between ensemble spread and RMSE at the small scale is not improved during cycling. In general, the ensemble in Ens_noFLTR_6h does not become better after four cycles, which explains why Ens_noFLTR_6h produces a large analysis error.*

3) It would be also interesting to see the verification only for the typhoon region (e.g., accounting for grid points ≥ 10 dBZ).

Thanks for your comments.

The region for grid points > 10 dBZ mainly represents the convective area of a typhoon and contains small-scale information. Usually, the RMSE in this region is larger than that for the whole domain. The following figure shows (a) the RMSE in the convective area and (b) the RMSE for the whole domain. Except for $T$, most examined variables have larger errors in the convective area. This difference between the convective area and the whole domain is also valid for the difference between the small-scale error and the large-scale error shown in Figures 9-11, and 13. No matter the area, Hybrid_5band_DSL produces the lowest analysis error; the hybrid covariance and multiscale localization are conducive to lowering the analysis error for wind components and hydrometeors. Therefore, we just mentioned the error in the convective area in section 4.2 as follows.

Line 489-492

*In addition, convective-scale DA usually computes the errors for grid points with reflectivity larger than a threshold, which is another way to investigate small-scale errors. The difference between errors in the convective area (reflectivity >10 dBZ) and the rest area is similar to that between small-scale and large-scale errors (not shown). Therefore, the errors in the convective area are not discussed in the subsequent sections.*

(a)

| | U | V | T | QVAPOR | QRAIN | QSNOW | QGRAUP |
|---|---|---|---|---|---|---|---|
| BAK | 4.87 | 4.22 | 0.72 | 5.30E-04 | 3.39E-04 | 3.25E-04 | 1.73E-04 |
| Ens_noFLTR | 3.24 | 3.39 | 0.97 | 6.20E-04 | 3.05E-04 | 3.07E-04 | 1.67E-04 |
| Static_BE | 2.67 | 3.28 | 0.68 | 5.30E-04 | 2.85E-04 | 2.98E-04 | 1.62E-04 |
| Hybrid_noFLTR | 2.55 | 2.85 | 0.76 | 5.60E-04 | 2.87E-04 | 2.98E-04 | 1.68E-04 |
| Ens_2band | 2.68 | 3.15 | 0.81 | 5.70E-04 | 2.97E-04 | 3.01E-04 | 1.61E-04 |
| Ens_3band | 2.38 | 2.88 | 0.78 | 5.50E-04 | 2.90E-04 | 2.92E-04 | 1.59E-04 |
| Ens_5band | 2.28 | 2.73 | 0.76 | 5.40E-04 | 2.85E-04 | 2.92E-04 | 1.58E-04 |
| Hybrid_2band | 2.34 | 2.85 | 0.69 | 6.20E-04 | 2.76E-04 | 2.94E-04 | 1.57E-04 |
| Hybrid_3band | 2.25 | 2.68 | 0.69 | 5.90E-04 | 2.71E-04 | 2.89E-04 | 1.56E-04 |
| Hybrid_5band | 2.31 | 2.66 | 0.70 | 6.40E-04 | 2.72E-04 | 2.90E-04 | 1.56E-04 |
| Ens_noFLTR_OL | 3.35 | 3.29 | 0.86 | 5.50E-04 | 3.05E-04 | 3.11E-04 | 1.67E-04 |
| Ens_LETKF | 3.41 | 3.36 | 0.88 | 6.10E-04 | 2.78E-04 | 2.89E-04 | 1.81E-04 |
| Ens_noFLTR_DSL | 3.13 | 3.30 | 0.95 | 5.80E-04 | 3.00E-04 | 3.05E-04 | 1.65E-04 |
| Hybrid_5band_DSL | 2.18 | 2.57 | 0.65 | 4.10E-04 | 2.77E-04 | 2.95E-04 | 1.57E-04 |
| Ens_5band_DSL | 2.25 | 2.81 | 0.75 | 4.80E-04 | 2.85E-04 | 2.96E-04 | 1.59E-04 |
| | (m/s) | (m/s) | (K) | (kg/kg) | (kg/kg) | (kg/kg) | (kg/kg) |

(b)

| | U | V | T | QVAPOR | QRAIN | QSNOW | QGRAUP |
|---|---|---|---|---|---|---|---|
| BAK | 3.25 | 3.19 | 1.33 | 5.20E-04 | 8.87E-05 | 8.46E-05 | 4.48E-05 |
| Ens_noFLTR | 2.91 | 2.83 | 1.48 | 5.80E-04 | 8.30E-05 | 8.31E-05 | 4.51E-05 |
| Static_BE | 2.35 | 2.53 | 1.24 | 5.70E-04 | 7.87E-05 | 8.19E-05 | 4.44E-05 |
| Hybrid_noFLTR | 2.36 | 2.44 | 1.28 | 5.70E-04 | 7.98E-05 | 8.24E-05 | 4.64E-05 |
| Ens_2band | 2.46 | 2.53 | 1.37 | 5.20E-04 | 7.98E-05 | 8.06E-05 | 4.31E-05 |
| Ens_3band | 2.34 | 2.44 | 1.35 | 5.50E-04 | 7.93E-05 | 7.96E-05 | 4.32E-05 |
| Ens_5band | 2.33 | 2.42 | 1.36 | 5.80E-04 | 7.80E-05 | 7.96E-05 | 4.31E-05 |
| Hybrid_2band | 2.28 | 2.35 | 1.29 | 6.40E-04 | 7.63E-05 | 8.08E-05 | 4.32E-05 |
| Hybrid_3band | 2.22 | 2.31 | 1.28 | 6.30E-04 | 7.60E-05 | 8.07E-05 | 4.36E-05 |
| Hybrid_5band | 2.24 | 2.30 | 1.29 | 6.10E-04 | 7.59E-05 | 8.06E-05 | 4.32E-05 |
| Ens_noFLTR_OL | 2.79 | 2.78 | 1.40 | 5.40E-04 | 8.11E-05 | 8.23E-05 | 4.41E-05 |
| Ens_LETKF | 2.82 | 2.77 | 1.42 | 5.20E-04 | 8.99E-05 | 9.30E-05 | 5.81E-05 |
| Ens_noFLTR_DSL | 2.75 | 2.76 | 1.43 | 5.40E-04 | 8.06E-05 | 8.17E-05 | 4.41E-05 |
| Hybrid_5band_DSL | 2.13 | 2.24 | 1.22 | 4.80E-04 | 7.54E-05 | 8.00E-05 | 4.26E-05 |
| Ens_5band_DSL | 2.23 | 2.38 | 1.30 | 5.10E-04 | 7.66E-05 | 7.93E-05 | 4.26E-05 |
| | (m/s) | (m/s) | (K) | (kg/kg) | (kg/kg) | (kg/kg) | (kg/kg) |

Colorbar scale: 1, 2, 3, 4, 5, 6, 7, 8, 9, 10, 11, 12, 13, 14, 15

4) The shock is often mentioned in the article, can authors provide a metric for imbalance? Combining it with the RMSEs may show more insights of the results.

Thanks for your comments.

Authors acknowledge that it is necessary to tell readers what the shock is. We add the related text and a figure (Figure 17) in section 4.3.1.

The related text is as follows. Line 574-590

*4.3.1 The tendency of ps*

*The ps tendency in the truth simulation is selected as a criterion as it is assumed to be in balance status after a 24-h forecast. The balanced tendency is approximately 20 Pa h$^{-1}$ (Figure 17), which is reached by BAK in 3 h. After the first DA cycle, the ps tendency becomes much larger than that of BAK, no matter the DA configuration. The large ps tendency after the first DA cycle is not surprising because the landing typhoon is not fully observed by the simulated observation network, especially for the wind field, causing an imbalance between the corrected part and the rest of the analyzed typhoon. A similar phenomenon was discussed by Wang et al. (2012) in a simulated supercell case. They concluded that such an imbalance shocks the model forecast and increases the forecast error.*

*After a 6-h forecast, the ps tendencies in Hybrid_5band_DSL_6h and Ens_noFLTR_6h are close to the balance status. As expected, the ps tendency increases again after the second DA cycle. However, Hybrid_5band_DSL_6h produces a much smaller ps tendency than Ens_noFLTR_6h, indicating that Hybrid_5band_DSL_6h has a more balanced analysis. The peaks of ps tendency in Hybrid_5band_DSL_6h and Ens_noFLTR_6h gradually decline as the number of cycles increases. By 18 UTC, Hybrid_5band_DSL_6h reaches the balance status while Ens_noFLTR_6h does not. The above result indicates that using the hybrid covariance and multiscale localization is beneficial for cycling DA.*

*Note that the advantage of Hybrid_5band_DSL_6h has a precondition that the cycling interval is sufficiently long for the model to spin up. When the cycling interval becomes shorter (Hybrid_5band_DSL_3h), the ps tendency cannot be effectively suppressed as Hybrid_5band_DSL_6h does.*

5) I suggest that a metric with scale skills (e.g., Fractions Skill Score) can be used especially for the reflectivity forecast verification.

Thanks for your comments.

Authors acknowledge that FSS is a good metric to evaluate the forecast of reflectivity. However, since we no longer discuss the forecasts initialized from a single deterministic analysis in the revised manuscript, there is no need to discuss the FSS of reflectivity forecast either.

---

## Referee Report (RR1)

**General comments**

A new algorithm called "Local DA" is introduced. Compared to the last version, the current manuscript has been considerably improved. Overall, I feel that the paper is lengthy as authors attempts to tackle several problems. But some of them remain still unclear to the end and will be further investigated in the future as authors claims. In my opinion, the most remarkable difference compared to the LETKF is that the Local DA works in columns instead of in points. Therefore, I have one concern in this regard.

1. If I understand correctly (I may get it wrong), the Local DA works in columns. Therefore, I thought there is no need for vertical localization for integrated observations (e.g., PWV). But there is vertical localization applied to PWV. Intuitively, I would have thought that the Local DA should be able to better assimilate PWV data since it does not require the vertical localization, but there is no discussion about this.

Minor points:

1. Line 55: do not understand "The LETKF, however, performs the analysis in the ensemble space, which implies that a static ensemble is necessary".

2. Section 2.1: It is suggested that dimensions be given here as well.

3. Line 170: do not understand "Note that the variational DA methods seek the combination of the columns of the square root of the background error covariance matrix, while Local DA combines the columns of the error correlation matrix" since the 3DVAR is also formulated as an optimization problem of control variables.

4. Line 210: What is $\overline{\hat{\mathbf{X}}}_o$ ?

5. Line 244: do not change $-->$ are equal

6. Line 259 : Typo: Coo

7. Line 283: $\mathbf{x}^f$ is not initial condition

8. Line 407: Is the fixed multiplication inflation 1.5 employed in all experiments?

9. Line 464: "Let us have a quick look at the results", the language is too causal.

10. Line 511: speculation $-->$ assumption

11. Line 540-541: differences ... are ...

12. Line 542: overestimated negative increment

13: Line 615: correlation $--$ > ratio

---

## Author Response (AR2)

We thank the reviewers for their careful review of our paper and the helpful comments that improved the manuscript. We give the response in blue and cite the revised text in orange. The original comments are reproduced.

Reviewer 1

I am generally satisfied with the revision of the manuscript, but I still have two comments on the double-space localization and the multiscale localization methods.

1. I understand that the Local DA algorithm is capable of double-space localization. My question is: do we really need to apply two localizations? The results of this study did not demonstrate the substantial benefits of the double-space localizations either. The authors need to comment on and discuss it.

Thanks for your comments. The authors agree that whether to use double-space localization should be explicitly discussed.

Firstly, the performance of double-space localization relies on the model-space localization in the combination. The results show (Figure 11) that double-space localization with a multiscale model-space localization (Ens_5band_DSL) outperforms the one with a fixed localization scale (Ens_noFLTR_DSL). Double-space localization can serve as a supplement to model-space localization.

Whether to use double-space localization depends on the application scenario. If Local DA is extended to a four-dimensional DA scenario, double-space localization is not recommended because observation-space localization does not consider the advection of background error covariance. In the case of a three-dimensional DA scenario, double-space localization is conducive to a small analysis error when the background error covariance is noisy because the localization helps reduce spurious increments. In contrast, if the background error covariance is good, double-space localization is not necessary.

We revised the related text as follows.

In section 4.2.3, line 543-546

"The spurious increment is further reduced in Hybrid_5band_DSL, especially at 850 hPa and 500 hPa, indicating that the positive impact of double-space localization corresponds to less noise in the analysis. According to the above result, double-space localization may serve as a supplement to pure model-space localization which determines the level of analysis error."

In section 5, line 672-676

"Despite the encouraging results, whether to use double-space localization should be considered case

by case. In this study, the background error covariance is noisy, so double-space localization has a positive impact. With a well-sampled ensemble and a well-designed multiscale localization, there is no need to use double-space localization. In the case of applying Local DA in the four-dimensional DA scenario, double-space localization should not be used because observation-space localization does not consider the advection of error covariance."

2. I am glad that the authors provided the algorithms (Eqs. [11-16]) for multiscale localization in Local DA. Those algorithms show that a multiscale localization method neglecting cross-scale covariances was proposed. However, those algorithms seem to be ad hoc for neglecting cross-scale covariances and are unable to be easily extended to include cross-scale covariances. If so, the authors need to clarify this in the manuscript.

Thanks for your comments. The authors agree that the multiscale covariance proposed in this study is difficult to extend to cross-scale covariance.

We revised the related text as follows.

In section 2.2, line 256-260

"Note that the multiscale covariance proposed in this section naturally excludes cross-scale covariance and is hard to incorporate cross-scale localization. How to determine the localization between two scales is also a question. The existing cross-scale localization (e.g., Huang et al., 2021; Wang et al., 2021) is implemented in spectral space and cannot be directly applied in equations (15) and (16). We plan to deal with the cross-scale issue in future work."

In addition, the multiscale localization method without the cross-scale covariances requires retaining the raw ensemble variances from the decomposed ensemble vectors. However, Eq. (11) shows that the decomposed scales are summed to recover the original raw ensemble perturbations. Eq. (11) should be used for the multiscale localization method with the cross-scale covariances. Authors can refer to Buehner (2012) and Huang et al. (2021) for details.

Thanks for pointing out this issue.

We revised the related text as follows.

In section 2.2, line 218-223

"To realize multiscale localization in model space, Local DA first performs scale decomposition with a bandpass filter. The decomposed perturbation, $\mathbf{X}'_b$, is

$$\mathbf{X}'_b = \left( \mathbf{X}_b^1, \mathbf{X}_b^2, \cdots, \mathbf{X}_b^l, \cdots, \mathbf{X}_b^{N_b} \right), \tag{11}$$

where the superscript "$l$" represents the $l$th scale and $N_b$ is the number of scales. After decomposition, the number of samples becomes $N_b$ times as large as the original ensemble size. As a localization approach

lacking cross-scale covariance (no $\mathbf{X}_b^i \mathbf{X}_b^{j\mathrm{T}}, i \neq j$ term in $\mathbf{X}_b' \mathbf{X}_b'^{\mathrm{T}}$), Local DA computes the STD of the perturbation, $s$, according to"

Reviewer 2

General comments A new algorithm called "Local DA" is introduced. Compared to the last version, the current manuscript has been considerably improved. Overall, I feel that the paper is lengthy as authors attempts to tackle several problems. But some of them remain still unclear to the end and will be further investigated in the future as authors claims. In my opinion, the most remarkable difference compared to the LETKF is that the Local DA works in columns instead of in points. Therefore, I have one concern in this regard.

1. If I understand correctly (I may get it wrong), the Local DA works in columns. Therefore, I thought there is no need for vertical localization for integrated observations (e.g., PWV). But there is vertical localization applied to PWV. Intuitively, I would have thought that the Local DA should be able to better assimilate PWV data since it does not require the vertical localization, but there is no discussion about this.

Thanks for your comments. Yes, you are right. The Local DA can work in columns, as well as in points. In this study, the Local DA works in 5-column mode.

The vertical localization in the observation space was disabled for all Local DA experiments. It can be seen in the code (src/da_core/da_core_enda.f90) that the vertical distance between an observation and a model grid point is set to zero (dv=0.0) for Local DA (broot_opt==1).

```
262     if(broot_opt==0) call cal_loc_coef_ll(dh,dv,radii_h_tem,radii_v_tem,envar%local_coef(iobs_sub),coef_mark)
263     if(broot_opt==1) call cal_loc_coef_ll(dh,0.0,radii_h_tem,radii_v_tem,envar%local_coef(iobs_sub),coef_mark)
```

In the previous revision, we stated that Local DA performs vertical localization in the model space ("Notably, Ens_noFLTR_OL performs vertical localization in model space, identical to Ens_noFLTR." and "Ens_noFLTR_DSL performs localization in both the model and observation space. In the model space, a fixed localization radius is used, as in Ens_noFLTR, while the localization parameters of Ens_noFLTR_OL are adopted for observation-space localization."), but we did not explicitly claim that the vertical localization in the observation space is disabled. To make the statement clearer and emphasize no vertical localization in the observation space, we revise the related text as follows.

In section 3.3.2, line 429-431. We state that the vertical localization in the observation space is set in the LETKF experiment (Ens_LETKF).

"**The vertical radius for all observations is 5 km in Ens_LETKF**, where the PWV observations are supposed to be available at 4000 m for LETKF localization."

We explicitly state that Local DA uses no vertical localization in the observation space at line 434-436

"For convenience, all single deterministic analysis experiments are listed in Table 3, where "M", "O", and "M+O" denote model-space, observation-space, and double-space localization, respectively. **The vertical localization in the observation space is disabled for all Local DA**

**experiments.**"

Minor points:

1. Line 55: do not understand "The LETKF, however, performs the analysis in the ensemble space, which implies that a static ensemble is necessary".

The related words are revised as "The large ensemble (>=800) is not always available in practice because of the limited computational and storage resources. However, it is inevitable to use such an ensemble to realize the hybrid analysis in the original LETKF framework because the LETKF works in the ensemble space." (Line 54-57)

2. Section 2.1: It is suggested that dimensions be given here as well.

The dimensions of vectors and matrices in equations (1) and (2) depend on the number of observations used in a local analysis and the complexity of observation operators. Therefore, it is better to give the dimensions and computations of the above variables in the subsequent subsections. We added some words in section 2.1 to state the above circumstance.
Line 115-117
"The dimensions of vectors and matrices in equations (1) depend on the number of observations involved in a local analysis and the complexity of observation operators. We will give the dimensions and computations of the above variables in the following subsections."

3. Line 170: do not understand "Note that the variational DA methods seek the combination of the columns of the square root of the background error covariance matrix, while Local DA combines the columns of the error correlation matrix" since the 3DVAR is also formulated as an optimization problem of control variables.

Thanks for your comments. The variational DA methods and Local DA differ in the control variable transform viewpoint. The control variable transform converts control variables to model state variables. The variational DA methods use the square root of the background error covariance matrix, while Local DA employs the error correlation matrix.
The related words are revised as "Note that the variational DA methods and Local DA differ in the control variable transform viewpoint. The former uses the square root of the background error covariance matrix, while Local DA employs the error correlation matrix." (Line 172-173)

4. Line 210: What is $\overline{\hat{\mathbf{X}}_o}$ ?

The mean of $\hat{\mathbf{X}}_o$.

The related words are revised as

"Local DA approximates the linear projection $\tilde{\mathbf{Y}} = \mathbf{H}_o \hat{\mathbf{X}}_o$ according to

$$\tilde{\mathbf{Y}} \approx h(\mathbf{x}^f + \hat{\mathbf{X}}_o) - h(\mathbf{x}^f + \overline{\hat{\mathbf{X}}_o}),\tag{10}$$

where $h$ is the nonlinear observation operator, $\mathbf{x}^f$ is the background model state vector, and $\overline{\hat{\mathbf{X}}_o}$ is the

mean of $\hat{\mathbf{X}}_o$." (Line 211-213)

5. Line 244: do not change $--$> are equal

Our original statement is somehow confusing and we have revised it.
The related words are revised as "Because the multiscale localization does not change the sizes of $\mathbf{C}_{oo}$ and $\mathbf{C}_{mo}$, there is no modification for $\mathbf{v}_o$, $\mathbf{x}^i$, $\mathbf{x}^f$, and $\mathbf{x}^a$." (Line 247)

6. Line 259 : Typo: Coo

Fixed as $\mathbf{C}_{oo}$. (Line 266)

7. Line 283: $\mathbf{x}^f$ is not initial condition

We use "background model state" in the revision. (Line 290)

8. Line 407: Is the fixed multiplication inflation 1.5 employed in all experiments?

Yes.

9. Line 464: "Let us have a quick look at the results", the language is too causal.

The related text is revised as "The domain averaged root mean square root error (RMSE) is examined first. For convenience, the initial condition extracted from GFS analysis is referred to as BAK. All experiments reduce the errors in the observation space after DA" (Line 472-473)

10. Line 511: speculation $--$> assumption

Fixed. (Line 519)

---

## Author Response (AR3)

We thank the editor for his careful review of our paper and the helpful comment that improved the manuscript. We give the response in blue and cite the revised text in orange. The original comments are reproduced.

On line 431, you state "the PWV observations are supposed to be available at 4000 m for LETKF localization." If I understand correctly, this vertically-integrated variable is treated as being a point observation located at 4000 km? Assuming this understanding is correct, I suggest a minor revision to the text to avoid potential confusion associated with the use of the phrase 'supposed to': "the PWV observations are treated as being located at 4000 m for LETKF localization."

Thanks for your comments. We have revised the text according to your suggestion.

Line 431-432

"The vertical radius for all observations is 5 km in Ens_LETKF, where the PWV observations are treated as being located at 4000 m for LETKF localization."

In addition, we corrected a typo in figure 5 where the label (c) in the lower panel should be (b).